# Generating Directed Graphs with Dual Attention and Asymmetric Encoding

**Alba Carballo-Castro,** * **Manuel Madeira, Yiming Qin, Dorina Thanou & Pascal Frossard**
LTS4, EPFL, Lausanne, Switzerland

## Abstract

Directed graphs naturally model systems with asymmetric, ordered relationships, essential to applications in biology, transportation, social networks, or visual understanding. Generating such graphs enables simulation, data augmentation and novel instance discovery; however, this task remains underexplored. We identify two key reasons: first, modeling edge directionality introduces a substantially larger dependency space, making the underlying distribution harder to learn; second, the absence of standardized benchmarks hinders rigorous evaluation. Addressing the former limitation requires more expressive models that are sensitive to directional topologies. Thus, we propose DIRECTO, the first generative model for directed graphs built upon the discrete flow matching framework. Our approach combines: (i) a dual-attention mechanism distinctly capturing incoming and outgoing dependencies, (ii) a robust, discrete generative framework, and (iii) principled positional encodings tailored to asymmetric pairwise relations. To address the second limitation and support evaluation, we introduce a novel and extensive benchmark suite covering synthetic and real-world datasets. Experiments show that our method outperforms existing directed graph generation approaches across diverse settings and competes with specialized models for particular classes, such as directed acyclic graphs. These results highlight the effectiveness and generality of our approach, establishing a solid foundation for future research in directed graph generation.

## 1 Introduction

Directed graphs (digraphs) model systems with asymmetric relationships, capturing essential structures such as flows, dependencies, and hierarchies that arise in many real-world applications. This makes digraphs particularly well-suited for problems in diverse domains including biology (Li et al., 2006; Takane et al., 2023; Wei et al., 2024), transportation (Concas et al., 2022), social dynamics (Schweimer et al., 2022), and, more recently, image and video understanding (Chang et al., 2023; Rodin et al., 2023), where structured and directional representations are critical for interpretation and reasoning. Consequently, generating digraphs is central to tasks such as simulation, data augmentation and novel instance discovery in domains with directional structure.

Graph generative models have shown strong potential in diverse applications, including drug discovery (Mercado et al., 2020; Vignac et al., 2023b), finance (Li et al., 2023), social network modeling (Tsai et al., 2023), and medicine (Nikolentzos et al., 2023). However, most generative models focus on undirected graphs (Liao et al., 2019; Martinkus et al., 2022; Vignac et al., 2023a; Siraudin et al., 2024; Xu et al., 2024; Eijkelboom et al., 2024; Qin et al., 2025a), with only limited efforts on directed settings where recent works propose auto-regressive models for DAGs (Zhang et al., 2019; Li et al., 2025) and, more recently, for general digraphs (Law et al., 2025). We identify two main factors limiting progress in this direction. First, at the modeling level, the directed setting is inherently more challenging than its undirected counterpart, as edge directionality greatly enlarges the learnable space (see Figure 1a: for graphs with 4 nodes, there are 218 possible digraphs, compared to only 11 undirected ones). We will see that simply extending undirected architectures without explicit directional components fails on this larger space of digraphs. This is further amplified in settings where domain-specific structural constraints must be respected, such as *directed acyclic graphs* (DAGs). Second, beyond these technical challenges, we identify the lack of standardized benchmarks for directed graph generation as a barrier to rigorous evaluation and fair comparison. Current DAG methods (Li et al., 2025) typically rely only on proxy metrics derived from downstream tasks, with no established benchmarks to evaluate digraph generative quality systematically.

---

*Correspondence to `alba.carballocastro@epfl.ch`

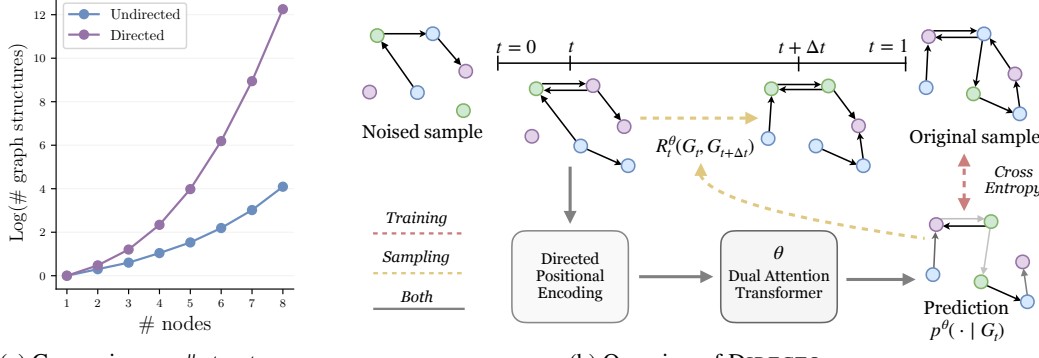

(a) Comparison on # structures.                    (b) Overview of DIRECTO.

Figure 1: (a) The learnable space increases drastically with the number of nodes for digraphs compared to undirected graphs (Harary & Palmer, 1973), highlighting challenges in extending graph generative models to directed structures. (b) Overview of our generative model for directed graphs (with node colors representing different classes). During training, the *dual attention transformer* (denoising network), parameterized by $\theta$, is enhanced with *asymmetric positional encoding*, learning to reverse predictions using cross-entropy loss. During inference, we compute the *rate matrix* $R_t^\theta(G_t, G_{t+\Delta t})$, which governs the evolution of the generative process over finite intervals $\Delta t$, based on the model prediction $p^\theta(\cdot \mid G_t)$.

To address the first challenge, we introduce DIRECTO[1], the first iterative refinement-based generative model for directed graphs of small to medium scale (from a few up to 200 nodes on average). Our approach brings together three core components of expressive graph generative modeling: (i) an expressive neural architecture, (ii) a robust generative framework, and (iii) informative input features. For the architecture, we employ a dual attention mechanism that performs cross-attention between edge features and their reversed counterparts, capturing both source-to-target and target-to-source information flow. All this is integrated within a robust discrete-state flow matching framework that enables efficient generation with state-of-the-art performance and more efficiently than diffusion-based alternatives, and which is also readily extendable to conditional generation tasks through classifier-free guidance (Ho & Salimans, 2022). Finally, the network's input features include positional encodings designed for asymmetric adjacency matrices, which we show outperform directionality-agnostic alternatives. Figure 1b gives an overview of our method and its components.

To tackle the second challenge, we propose a new benchmarking framework for rigorous evaluation of directed graph generation, spanning both synthetic and real-world datasets. The synthetic benchmarks cover different distributions, including acyclic and community-based graphs, while real-world benchmarks include a DAG dataset for *Neural Architecture Search* (NAS) (Phothilimthana et al., 2023) and a scene understanding dataset (Krishna et al., 2017). The evaluation metrics are systematic and tailored to directed graphs, including distributional distances for fidelity, and dataset-specific metrics reflecting the graph types used in different tasks. We evaluate DIRECTO on the proposed benchmarks, where it consistently outperforms prior approaches across all settings, underscoring its effectiveness on structurally diverse datasets. Extensive ablations further reveal that the dual attention mechanism is critical for modeling directional dependencies, while positional encodings play a role in enhancing overall generation quality.

Our main contributions are summarized as follows:

(i) We propose DIRECTO, the first flow-based digraph generative model, combining a direction-aware dual attention block with positional encodings that capture asymmetric structure.

(ii) We address the lack of benchmarks by releasing an evaluation suite covering synthetic graphs with relevant structural properties and real-world datasets with meaningful applications.

(iii) We conduct an extensive empirical analysis demonstrating that DIRECTO achieves state-of-the-art performance on structurally diverse synthetic and real-world graphs, while showing the need for a dedicated directed architecture.

---

[1]Code available at: https://github.com/acarballocastro/DIRECTO

Overall, by addressing the two main bottlenecks in directed graph generation, we propose an effective, new generative framework and provide standardized benchmarks, aiming to lay the groundwork for future research and to support real-world applications.

## 2 BACKGROUND: DISCRETE FLOW MATCHING FOR GRAPH GENERATION

**Notation** We denote by $G = \left( x^{(1:n:N)}, e^{(1:i \neq j:N)} \right)$ directed graphs with $N$ nodes. The set of nodes is denoted by $\{x^{(n)}\}_{n=1}^N$, and the set of directed edges by $\{e^{(i,j)}\}_{1 \leq i \neq j \leq N}$. Both nodes and edges are categorical variables, where $x^{(n)} \in \{1, \ldots, X\}$ and $e^{(i,j)} \in \{1, \ldots, E\}$. Inspired by standard practice in iterative refinement for undirected graphs (Vignac et al., 2023a; Xu et al., 2024; Siraudin et al., 2024; Qin et al., 2025a), we assume that every ordered pair of distinct nodes corresponds to a directed edge. Each edge is treated separately and always belongs to one of several possible categorical classes, including a class representing an absent edge. This formulation allows edges $(i, j)$ and $(j, i)$ to take different classes: both may be absent (no connection between nodes $i$ and $j$), only one may exist, or both may exist, even with different classes. Finally, we use a subscript $t$ to denote variables at time $t$, e.g., $x_t^{(n)}$ or $e_t^{(i,j)}$.

**Problem statement** Graph generative models aim to learn the probability distribution of a set of observed graphs, enabling the generation of new samples that preserve structural and statistical properties. *Discrete Flow Matching* (DFM)-based models (Gat et al., 2024; Campbell et al., 2024) have recently achieved state-of-the-art performance in undirected graph generation (Qin et al., 2025a). DFM belongs to a broader class of discrete state-space methods that recover the original graph distribution $p_1$ by progressively denoising samples from a pre-specified noise distribution $p_{\text{noise}}$ (Vignac et al., 2023a; Xu et al., 2024; Siraudin et al., 2024). It distinguishes itself from other methods in this class by decoupling sampling from training, allowing for post-training sampling optimization. Additionally, by operating directly in the discrete spaces, these methods naturally align with the discreteness of graphs and are well-suited to capture complex dependencies through iterative refinement. We hypothesize that this added expressivity is especially valuable for modeling the added complexity of digraphs. We now introduce the two main processes of DFM: *noising* and *denoising*.

**Noising process** The noising process runs from $t = 1$ to $t = 0$, progressively corrupting the original input graphs through a linear interpolation between the data distribution and the limit distribution $p_0 = p_{\text{noise}}$ (as detailed in Figure 1b). This interpolation is applied *independently* to each variable following discrete iterative refinement frameworks (Austin et al., 2021b; Campbell et al., 2022; Gat et al., 2024; Campbell et al., 2024), which corresponds to each node and edge in the case of graphs Qin et al. (2025a). For example, for each node $x_1^{(n)}$, its noisy distribution at step $t$ is defined as:

$$p_{t|1}^X(x_t^{(n)} \mid x_1^{(n)}) = t\,\delta(x_t^{(n)}, x_1^{(n)}) + (1-t)\,p_{\text{noise}}^X(x_t^{(n)}), \tag{1}$$

where $\delta(\cdot, \cdot)$ denotes the Kronecker delta and $p_{\text{noise}}^X$ is the limit distribution over node categories. A similar construction is used for the edges (Qin et al., 2025a).

**Denoising process** The denoising process aims to reverse the noising trajectory, running from $t = 0$ to $t = 1$. It is formulated as a *Continuous-Time Markov Chain* (CTMC), which starts sampling a noisy graph $G_0 \sim p_0$ and evolves according to:

$$p_{t+\Delta t|t}(G_{t+\Delta t} \mid G_t) = \delta(G_t, G_{t+\Delta t}) + R_t(G_t, G_{t+\Delta t})\,dt, \tag{2}$$

where $R_t$ is the CTMC rate matrix (Campbell et al., 2024). In practice, this update rule is approximated in two ways: first, by discretizing time with a finite step size $\Delta t$, in an Euler method step; and second, by approximating the ground-truth rate matrix with an estimate $R_t^\theta$ computed from the predictions of a denoising neural network $p_{1|t}^\theta(\cdot \mid G_t)$ parametrized by $\theta$. The denoising neural network, typically a denoising graph transformer, is trained to predict the clean node $p_{1|t}^{\theta,(n)}$ and edge $p_{1|t}^{\theta,(i,j)}$ distributions given a noisy graph, aggregated as:

$$p_{1|t}^\theta(\cdot | G_t) = \left( \left( p_{1|t}^{\theta,(n)}(x_1^{(n)} \mid G_t) \right)_{1 \leq n \leq N}, \left( p_{1|t}^{\theta,(i,j)}(e_1^{(i,j)} \mid G_t) \right)_{1 \leq i \neq j \leq N} \right). \tag{3}$$

Intuitively, the denoising graph transformer predicts larger probability of the clean graph $\boldsymbol{p}_{1|t}^{\theta}(\cdot \mid G_t)$ given the noisy graph $G_t$, which in turn is used to compute a higher transition rate in the rate matrix $\boldsymbol{R}_t^{\theta}$ (see Equation 43 for further clarification). This means that the network predicts the most likely transitions required to denoise the graph and recover the clean target, directly influencing the estimated rate of change.

This network is trained using a cross-entropy loss independently to each node and each edge:

$$\mathcal{L} = \mathbb{E}_{t,G_1,G_t} \left[ -\sum_n \log\left(p_{1|t}^{\theta,(n)}\left(x_1^{(n)} \mid G_t\right)\right) - \lambda \sum_{i \neq j} \log\left(p_{1|t}^{\theta,(i,j)}\left(e_1^{(i,j)} \mid G_t\right)\right) \right], \quad (4)$$

where the expectation is taken over time $t$, sampled from a predefined distribution over $[0, 1]$ (e.g., uniform); $G_1 \sim \boldsymbol{p}_1(G_1)$ is a clean graph from the dataset; and $G_t \sim \boldsymbol{p}_t(G_t|G_1)$ is its noised version at time $t$ (Qin et al., 2025a). The hyperparameter $\lambda \in \mathbb{R}^+$ controls the relative weighting between node and edge reconstruction losses. Further details are provided in Appendix I.

## 3 GENERATING DIGRAPHS WITH ASYMMETRIC ENCODING AND DUAL ATTENTION

Building on the discrete diffusion setting previously described, we now introduce DIRECTO, the first flow-based digraph generative model. We begin by outlining the overall generative framework, followed by a detailed description of the two directionality-aware components that form the core of DIRECTO: the use of directed graph positional encodings and the dual attention mechanism to enhance the edge directionality awareness of the denoising neural network in our framework.

### 3.1 DIRECTED GRAPH GENERATION VIA DISCRETE FLOW MATCHING

Building on the formulation of Section 2, we extend DFM to the directed graph setting. To address the added complexity of asymmetric adjacency matrices in digraphs (Figure 1a), we exploit a key strength of DFM: the decoupling of training and sampling. This enables post-training optimizations such as time-adaptive schedules and custom CTMC rate matrices (Qin et al., 2025a), which we can extend to directed graphs (details in Appendix C). While these strategies improve performance (see Appendix H.9), they remain insufficient to capture the structural properties of digraphs (see Section 5). We therefore introduce architectural components explicitly tailored for directionality-aware graph generation, providing the complete training and sampling algorithms in Appendix D.

### 3.2 ASYMMETRIC POSITIONAL ENCODING

Our generative framework employs a denoising GNN, which inherently struggles to capture global or higher-order patterns due to the locality of message passing (Xu et al., 2019; Morris et al., 2019). To maximize the capacity of our method, we augment our GNN with positional encodings (PEs) that inject structural information beyond local neighborhoods (Beaini et al., 2021; Bouritsas et al., 2020). Common choices for undirected graphs include Laplacian eigenvectors or shortest-path distances, effective in enriching node and edge representations (Vignac et al., 2023a; Qin et al., 2025a). However, these encodings overlook edge directionality, failing to distinguish asymmetric structural roles. To overcome this, we adopt direction-aware positional encodings (Geisler et al., 2023; Huang et al., 2025) that account for both outgoing and incoming connectivity. We append them to node and edge features, allowing our model to better capture the structural dependencies unique to digraphs.

Specifically, we experiment with three families of encodings: (i) the **Magnetic Laplacian (MagLap)** (Geisler et al., 2023), which introduces complex-valued phase shifts into the standard Laplacian to retain edge orientation; (ii) its **Multi-$q$ MagLap** extension (Huang et al., 2025), which stacks $Q$ Multiple Magnetic Laplacians with different complex potentials to provide richer representations; and (iii) **Directed Relative Random Walk Probabilities (RRWP)** (Geisler et al., 2023), which combine forward and reverse transition probabilities to capture outgoing and incoming asymmetric flows. These encodings are designed to provide direction-sensitive structural signals that better reflect the properties of directed graphs, and are described in detail in Appendix B.

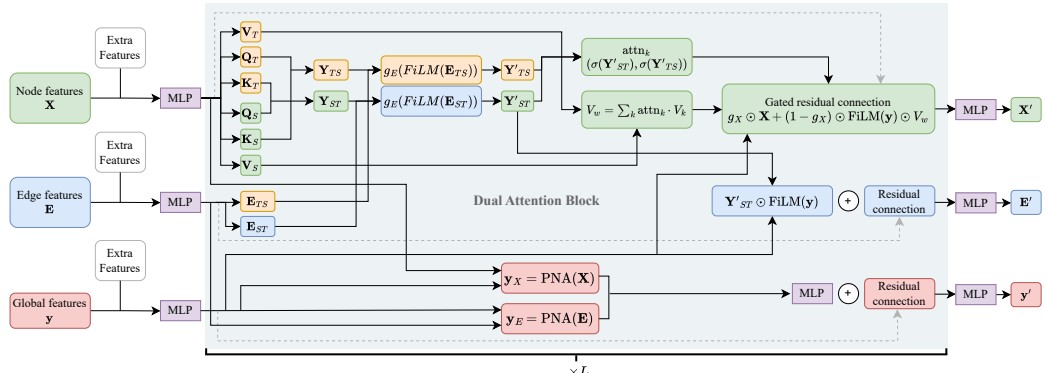

Figure 2: Network architecture of DIRECTO. We stack $L$ dual attention layers that account for both source-to-target and target-to-source information via cross-attention mechanisms. $\boldsymbol{X}$, $\boldsymbol{E}$ and $\boldsymbol{y}$ denote the stacked input node, edge, and global features. $\boldsymbol{X}'$, $\boldsymbol{E}'$ and $\boldsymbol{y}'$ are the output of the model, i.e., predicted clean node and edge distribution, and graph feature. FiLM (Perez et al., 2018) and PNA pooling layers (Corso et al., 2020) are incorporated to enable flexible modulation between node, edge, and graph-level features. Full technical details are provided in Appendix A.

We hypothesize that incorporating directionality-aware positional encodings improves digraph generative performance. To test this, we design an ablation study (see Section 5.2, with further details in Appendix H) evaluating which of the described encodings provide the most informative structural signals, while also considering their computational cost, to identify the best tradeoff between performance and efficiency.

## 3.3 GRAPH TRANSFORMER WITH DUAL ATTENTION

Graph Transformers (Dwivedi & Bresson, 2021; Rampášek et al., 2022) effectively model node interactions by leveraging the adjacency matrix to encode structural information. However, in directed graphs, it is crucial for the model to distinguish the different semantics carried by incoming and outgoing edges, which makes the standard graph transformer insufficiently expressive. Generating digraphs thus requires models that (i) capture bidirectional information flow to learn how source and target nodes influence each other distinctly, and (ii) integrate structural signals across nodes, edges, and the global graph to preserve relational dependencies and ensure coherent outputs.

To address this, our model jointly processes stacked node features $\boldsymbol{X}$, edge features $\boldsymbol{E}$, and global features $\boldsymbol{y}$ through a graph transformer composed of $L$ layers of a dual attention block (see Figure 2). This dual attention mechanism introduces two key components: (i) an attention-based aggregation scheme that captures directional information, and (ii) modulation layers to incorporate information at different levels of granularity across nodes, edges, and the global graph structure.

**Bidirectional information flow via dual attention aggregation** To model directional dependencies more effectively, we use a cross-attention mechanism between **s**ource-to-**t**arget (outgoing) edge features $\boldsymbol{E}_{\text{ST}}$ and their reversed **t**arget-to-**s**ource (incoming) counterparts $\boldsymbol{E}_{\text{TS}}$. By explicitly attending to both directions, our architecture, which builds on ideas from Wang et al. (2024), processes information bidirectionally and better captures directed relationships, improving accuracy and resulting in better generations. Concretely, we compute two directional attention maps between role-specific node projections:

$$\boldsymbol{Y}_{\text{ST}}[i,j] = \frac{\boldsymbol{Q}_{\text{S}}[i] \cdot \boldsymbol{K}_{\text{T}}[j]}{\sqrt{d_q}}, \quad \boldsymbol{Y}_{\text{TS}}[i,j] = \frac{\boldsymbol{Q}_{\text{T}}[i] \cdot \boldsymbol{K}_{\text{S}}[j]}{\sqrt{d_q}}, \tag{5}$$

where $\boldsymbol{Q}_{\text{S}}$, $\boldsymbol{Q}_{\text{T}}$, $\boldsymbol{K}_{\text{S}}$, and $\boldsymbol{K}_{\text{T}}$ are the role-specific projections for nodes that act as the **s**ource or the **t**arget of an edge, respectively, and $d_q$ is the query feature dimension. These attention weights are modulated through a FiLM layer (see Equation (8)) using edge features to enable effective reasoning, producing the updated attention maps $\boldsymbol{Y}'_{\text{ST}}$ and $\boldsymbol{Y}'_{\text{TS}}$.

To further consolidate directional information, we introduce an *attention aggregation mechanism*. Instead of treating the attentions from the two directions independently, we concatenate the modulated

attention maps and apply a single softmax operation to obtain unified attention weights:

$$\boldsymbol{A}_{\text{aggr}} = \text{softmax}(\text{concat}(\boldsymbol{Y}'_{\text{ST}}, \boldsymbol{Y}'_{\text{TS}})) \in \mathbb{R}^{n \times 2n}, \quad \boldsymbol{V}^{\top}_{\text{aggr}} = \text{concat}(\boldsymbol{V}^{\top}_{\text{S}}, \boldsymbol{V}^{\top}_{\text{T}}) \in \mathbb{R}^{2n \times h}, \quad (6)$$

where $h$ is the hidden dimension, $\boldsymbol{V}_{\text{S}}, \boldsymbol{V}_{\text{T}}$ are the value vectors for source and target nodes, and the concatenation operation is performed along the node dimensions. These unified weights are then used to compute a convex (i.e., softmax-weighted) combination of the value vectors:

$$\boldsymbol{X}' = \boldsymbol{A}_{\text{aggr}} \boldsymbol{V}_{\text{aggr}} \in \mathbb{R}^{n \times h}. \quad (7)$$

This aggregation allows the model to assign asymmetric importance to the outgoing and the incoming directions. The node features are then updated using a gated residual connection, which adaptively combines the original and updated features by learning a gate that controls how much of the new information should be integrated in the update.

**Multi-scale information modulation for graph denoising** A graph denoising model should predict clean node and edge types from noisy inputs, which requires integrating both local interactions and global structure from node, edge, and graph-level features. Building on standard graph generation architectures (Vignac et al., 2023a; Siraudin et al., 2024; Qin et al., 2025a), we incorporate *Feature-wise Linear Modulation* (FiLM) (Perez et al., 2018) and *Principal Neighbourhood Aggregation* (PNA) (Corso et al., 2020) layers to enhance multi-scale feature fusion. FiLM adaptively modulates edge features using attention signals: given the edge feature matrix $\boldsymbol{E}$, the attention matrix $\boldsymbol{E}_{\text{attn}}$, and the learnable weights $\boldsymbol{W}^1_{\text{FiLM}}, \boldsymbol{W}^2_{\text{FiLM}}$, it computes

$$\text{FiLM}(\boldsymbol{E}, \boldsymbol{E}_{\text{attn}}) = \boldsymbol{E}\boldsymbol{W}^1_{\text{FiLM}} + (\boldsymbol{E}\boldsymbol{W}^2_{\text{FiLM}}) \odot \boldsymbol{E}_{\text{attn}} + \boldsymbol{E}_{\text{attn}}, \quad (8)$$

where $\odot$ denotes element-wise multiplication. These layers are also used to integrate global graph-level signals $\boldsymbol{y}$ into the construction of edge features.

To complement this, PNA layers aggregates multi-scale neighborhood information via pooling operations: given node (or edge) features $\boldsymbol{X} \in \mathbb{R}^{n \times h}$ and a learnable weight matrix $\boldsymbol{W}_{\text{PNA}}$:

$$\text{PNA}(\boldsymbol{X}) = \text{concat}\left(\text{max}(\boldsymbol{X}), \text{min}(\boldsymbol{X}), \text{mean}(\boldsymbol{X}), \text{std}(\boldsymbol{X})\right) \boldsymbol{W}_{\text{PNA}} \in \mathbb{R}^{4h}, \quad (9)$$

with concatenation performed along the feature dimension. We use PNA layers to update global graph signals based on local node- and edge-level information at each attention pass. Together, these components enable expressive and scalable integration of local and global features, allowing the model to capture higher-order structure which is critical for accurate graph denoising.

## 4 BENCHMARKING DIRECTED GRAPH GENERATION

Despite growing interest in digraphs (Sun et al., 2024), benchmarking in this area remains under-developed (Bechler-Speicher et al., 2025). Existing approaches typically rely on downstream task performance, failing to directly evaluate generative quality. With the objective of establishing standard datasets and metrics, we propose a comprehensive benchmark that covers synthetic and real-world digraphs, along with tailored metrics that directly assess key generative qualities, including sample validity, diversity, and distributional alignment.

### 4.1 DATASETS

To enable systematic evaluation, we introduce a suite of synthetic directed datasets alongside two real-world use cases: (i) neural architecture search (NAS), where DAGs model computational pipelines, and (ii) scene graphs, where directed edges encode semantic relations in visual scenes. Further details and statistics for all datasets are available in Appendix E.

**Synthetic datasets** We sample from the directed and DAG Erdős–Rényi (binomial) model; from Price's model, a directed analogue of the Barabási–Albert model that produces DAGs; and from a directed version of the Stochastic Block Model (SBM) dataset proposed in Martinkus et al. (2022). Each dataset comprises 200 graphs, split into 128 training, 32 validation, and 40 test graphs.

**TPU Tiles**    A relevant application of DAGs is Neural Architecture Search, which aims to generate novel instances to achieve an automated design of efficient Neural Networks (Elsken et al., 2019). To support this task, the TPU Tiles dataset (Phothilimthana et al., 2023) contains computational graphs extracted from Tensor Processing Unit workloads. We split into 5,040 training, 630 validation, and 631 test graphs following the preprocessing from Li et al. (2025).

**Visual Genome**    Scene graphs represent the semantics of visual scenes, supporting tasks such as image retrieval, captioning, or visual question answering. Generating such graphs is useful for data augmentation, scenario simulation, or robustness, complementing recent scene-to-image generation work (Johnson et al., 2018; Yang et al., 2022). We use the Visual Genome dataset (Krishna et al., 2017), a widely adopted benchmark of graphs that are not acyclic but instead represent general directed structures, with nodes corresponding to objects (entities), relationships (directed interactions), or attributes (object properties). We extract 203 training, 51 validation, and 63 test graphs.

## 4.2 METRICS

To evaluate digraph generations, we build upon the evaluation metrics from the undirected graph generation literature  (Martinkus et al., 2022; Bergmeister et al., 2024; Thompson et al., 2022) and further extend the ones already present in the directed setting (Law et al., 2025).

**Validity, uniqueness, and novelty**    For synthetic datasets, *validity* is assessed via statistical tests of adherence to the generative distribution and, for DAG datasets, by ensuring acyclicity. On the TPU Tiles dataset, we measure acyclicity, while for Visual Genome we check typed structural constraints (e.g., edges go from objects to attributes and relationships, and from relationships to objects). To measure generative diversity while avoiding memorization, we compute *uniqueness* (fraction of non-isomorphic generated graphs) and *novelty* (fraction of generated graphs not in the training set). The V.U.N. ratio reports the proportion of samples that are simultaneously valid, unique, and novel.

**Structural distributional alignment**    To evaluate proximity to the original distribution, we compute the Maximum Mean Discrepancy (MMD) (Martinkus et al., 2022) between generated and test graphs across several statistics: outgoing and incoming degree distributions, directed clustering coefficients, spectral, and wavelet-based features. Spectral features are derived from the directed Laplacian (Chung, 2005), simplifying computations compared to the Magnetic Laplacian. Following standard practice, we first normalize results as the ratio from the generated samples MMD to that of the training data and then average the ratios for summarization. Values per MMD descriptor are detailed in the extended results in Appendix H..

**Joint node-edge distributional alignment**    To better capture performance on attributed graphs, we extend evaluation beyond structural statistics to joint node–edge distributions. Specifically, we measure MMD over label histograms to assess coverage of categorical proportions. We also compute triplet-based precision and recall by matching semantic tuples (e.g., (object, relation, object)) between generated and test graphs. Finally, we evaluate embedding-based distances (MMD with RBF kernel and FID) over GNN-derived representations that encode both labels and structure (Thompson et al., 2022). These metrics provide a comprehensive view of alignment with the reference distribution.

Among all metrics, we consider V.U.N. and the average ratio as the most informative for structural distributional alignment, and the RBF MMD for the joint node-label distributional alignment in real-world datasets. Additional details on the evaluation metrics are provided in Appendix F.

## 5 EXPERIMENTS

In this Section, we first evaluate the flexibility of our method to generate digraphs across synthetic and real-world datasets. Then, we analyze the role of the proposed architectural improvements on generative performance and analyze scalability and conditional generation.

**Baselines**    In the general directed setting, we compare DIRECTO against performing Maximum Likelihood Estimation (MLE) baseline of the node count, node types, and edge types, and sampling from the resulting distribution to generate new graph instances. For DAG settings, we also include

Table 1: Directed graph generation performance for different configurations of DIRECTO. Results are the mean $\pm$ standard deviation across five sampling runs. We considered $Q = 10$ for MagLap in the synthetic datasets and $Q = 5$ in the real-world ones (due to the computational cost of this positional encoding). OOT indicates that the model could not be run within a reasonable timeframe.

| Model | ER-DAG | | SBM | | TPU Tiles | | | Visual Genome | | |
|---|---|---|---|---|---|---|---|---|---|---|
| | Ratio ↓ | V.U.N. ↑ | Ratio ↓ | V.U.N. ↑ | Ratio ↓ | V.U.N. ↑ | RBF ↓ | Ratio ↓ | V.U.N. ↑ | RBF ↓ |
| *Training set* | 1.0 | 0.0 | 1.0 | 0.0 | 1.0 | 0.0 | 0.002 | 1.0 | 0.0 | 0.021 |
| MLE | 15.1 ±0.2 | 0.0 ±0.0 | 11.6 ±0.2 | 0.0 ±0.0 | 149.8 ±0.7 | 24.7 ±0.0 | 1.039 ±0.033 | 17.0 ±0.6 | 0.0 ±0.0 | 0.618 ±0.025 |
| D-VAE | 106.6 ±5.4 | 0.0 ±0.0 | - | - | OOT | OOT | OOT | - | - | - |
| LayerDAG | 4.2 ±3.2 | 21.5 ±2.7 | - | - | 413.6 ±70.1 | **98.5** ±3.0 | 1.021 ±0.023 | - | - | - |
| DiGress | 1.9 ±0.3 | 34.0 ±4.1 | 3.9 ±0.9 | 41.5 ±5.1 | 57.5 ±1.7 | 70.9 ±3.4 | 0.097 ±0.033 | 17.0 ±0.6 | 0.3 ±0.6 | 0.232 ±0.028 |
| DeFoG | 1.6 ±0.2 | 75.0 ±2.2 | 4.3 ±0.8 | 37.0 ±6.6 | 63.7 ±2.6 | 72.0 ±2.4 | 0.059 ±0.015 | 10.8 ±0.7 | 50.8 ±8.4 | 0.085 ±0.023 |
| DIRECTO-DD RRWP | 1.4 ±0.3 | 79.0 ±3.7 | 1.7 ±0.4 | 81.5 ±3.2 | 61.0 ±2.9 | 76.8 ±1.9 | 0.058 ±0.023 | 15.3 ±0.8 | 72.7 ±3.9 | 0.039 ±0.004 |
| DIRECTO-DD MagLap | 1.5 ±0.2 | 85.0 ±9.2 | **1.5** ±0.4 | 95.5 ±3.7 | 64.3 ±5.3 | 77.0 ±7.0 | 0.079 ±0.027 | 7.6 ±0.7 | 61.9 ±4.4 | 0.042 ±0.006 |
| DIRECTO RRWP | 1.7 ±0.1 | **94.0** ±1.0 | 1.8 ±0.5 | **99.5** ±1.0 | 75.4 ±8.1 | 77.0 ±2.9 | 0.044 ±0.018 | 12.8 ±0.6 | **83.8** ±4.3 | **0.038** ±0.005 |
| DIRECTO MagLap | **1.3** ±0.2 | 92.0 ±3.7 | 2.0 ±0.3 | 96.5 ±2.5 | **44.0** ±7.1 | 80.5 ±4.6 | **0.042** ±0.001 | **6.2** ±0.5 | 67.0 ±4.3 | 0.051 ±0.012 |

the only two publicly available baselines: D-VAE (Zhang et al., 2019) and LAYERDAG (Li et al., 2025). Notably, the architectural improvements in Sections 3.2 and 3.3 are agnostic to the underlying iterative refinement framework. To demonstrate versatility and enable wider experimentation, we extend DIRECTO to a discrete diffusion (Vignac et al., 2023a) backbone (DIRECTO-DD), described in Appendix I.3. We also include DEFOG (Qin et al., 2025a) and DIGRESS (Vignac et al., 2023a), discrete flow matching and diffusion methods in the undirected setting, respectively, adapting both to the directed case by removing the edge symmetrization operations. For each experiment, we highlight the **best result** and second-best method, with further experimental details in Appendix G, including Table 8 for a fairness comparison of resources and runtime of each of the models.

## 5.1 GENERATIVE PERFORMANCE EVALUATION

**Synthetic datasets** Table 1 reports the results on two synthetic datasets: ER-DAG and SBM. In both settings, DIRECTO consistently achieves the best trade-off between sample quality and validity. For ER-DAG, it achieves the highest validity score using directed RRWP as positional encoding, with a V.U.N. ratio of 94%. Notably, while LAYERDAG enforces acyclicity by design, it fails to capture the ER structure of the target graph distribution, as evidenced by its low V.U.N. score (21.5%; see Table 17 for details). In the SBM case, DIRECTO also successfully captures the distribution and consistently outperforms the baselines, achieving up to 99.5% in V.U.N. ratio and keeping low average graph statistics ratios. We provide the complete results for the distributional alignment metrics and the results for the other two synthetic datasets in Appendices H.1 and H.4.

**Real-world datasets** Table 1 shows the performance for the two real-world datasets. In TPU tiles, all DIRECTO variants significantly outperform DAG generation baselines in terms of structure-aware metrics. The ratio scores for DIRECTO are consistently and substantially lower than those observed in MLE, and especially LayerDAG, which indicates a more realistic distribution. Again, DIRECTO variants achieve higher V.U.N. than all baselines, with the exception of LAYERDAG, which benefits from enforcing acyclicity, the only validity constraint evaluated in this case. For Visual Genome, results reinforce the versatility of DIRECTO. DIRECTO RRWP achieves the highest V.U.N., indicating a superior ability to generate diverse and novel graphs while maintaining structural fidelity. Finally, we also see that DIRECTO achieves the lowest RBF MMD score for both datasets, demonstrating that the model is successful in capturing the joint node-label dependencies of the distribution in the original dataset. Complete results are available in Tables 10 and 11 in Appendix H.2.

**Conditional generation** As groundwork for directed graph generation, we prioritize strong unconditional generation, which we consider a key prerequisite for effective digraph modeling. At the same time, steering generation with graph-level properties is important for many downstream tasks. To illustrate the flexibility of our framework, we show that DIRECTO can be extended to conditional generation without major architectural changes via classifier-free guidance (Ho & Salimans, 2022; Nisonoff et al., 2025). On the TPU Tiles dataset, we condition on the execution time of the computational graph, aiming at low execution times. DIRECTO performs competitively with specialized autoregressive models like GraphRNN and LayerDAG (Appendix H.8), even surpassing the OneShotDAG variant of Li et al. (2025).

## 5.2 ABLATION STUDY

In the following, we present the results for ablations on the role of dual attention and the different positional encodings. We complement this with a scalability study and results on conditional generation. Complete results and further ablations on other model components can be found in Appendix H.

**Impact of dual attention** We now analyze the influence of the dual attention mechanism on the performance of DIRECTO. In Figure 3, we compare the V.U.N metric and the average graph statistics ratios ("Ratio") score across both synthetic datasets (ER-DAG and SBM), highlighting the impact of dual attention on generation quality and graph realism. We observe that the dual attention mechanism consistently improves generative performance, regardless of the positional encoding it is combined with. Notably, even in the absence of any positional encoding ("No PE"), dual attention still achieves non-zero V.U.N., highlighting its capacity to independently capture directionality-relevant information. See Appendix H.3 for the full ablation tables, both for DIRECTO and DIRECTO-DD.

**Impact of positional encodings** We also evaluate the sensitivity of DIRECTO to different positional encodings, including a direction-agnostic baseline with the Laplacian ("Lap") on synthetic datasets (see Appendix B for details on how it is computed using the symmetrized adjacency matrix). In Figure 3, we observe consistent trends across datasets: integrating positional encodings improves V.U.N. and Ratio, mirroring the undirected setting; moreover, direction-aware encodings outperform agnostic ones, supporting our modeling hypothesis. Among the encodings tested, Directed RRWP achieves the best overall performance on ER-DAG, and MultiMagLapPE with $Q = 10$ on SBM. Nevertheless, we remark that RRWP demonstrates clear superiority in both scalability and runtime efficiency (see Appendix G.3). Overall, while positional encodings influence performance, the dual attention mechanism remains the most critical component for capturing directional dependencies. Full tables for both DIRECTO and DIRECTO-DD can be found in Appendix H.4.

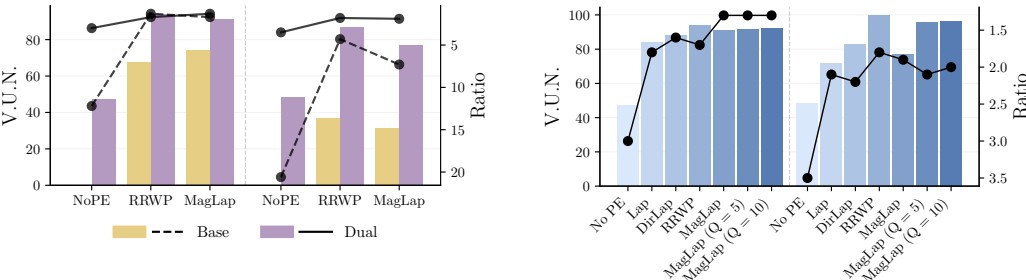

Figure 3: Ablation results for dual attention (left) and positional encodings (right). Each plot shows results on the ER-DAG (left bars/lines) and SBM (right bars/lines) datasets. Better performance corresponds to V.U.N. bars and Ratio lines appearing closer to the top of each subplot.

**Scalability study** We investigate scalability across three axes: dataset size, parameter efficiency, and graph size (Appendix H.7). Increasing the amount of training data improves performance but comes with longer training time (Table 18). For parameter efficiency, our dual attention mechanism clearly demonstrates better performance than simply enlarging the adapted undirected model, as it can be seen in Table 2. Finally, while larger graphs pose challenges, particularly for enforcing strict acyclicity, DIRECTO achieves strong generation quality (Ratio) across all graph sizes commonly used in the literature for the applications explored in this paper (Table 21).

Table 2: Directed graph generation performance for the dual attention mechanism versus doubling the depth of the standard network.

| Dataset | Model | Ratio ↓ | V.U.N. ↑ |
|---------|-------|---------|----------|
| ER-DAG | RRWP - Double | $4.9 \pm 0.2$ | $72.0 \pm 9.8$ |
| | RRWP - Dual | $\mathbf{1.7 \pm 0.1}$ | $\mathbf{94.0 \pm 1.0}$ |
| | MagLap - Double | $4.3 \pm 0.7$ | $80.0 \pm 6.3$ |
| | MagLap - Dual | $\underline{1.3 \pm 0.2}$ | $91.0 \pm 2.5$ |
| SBM | RRWP - Double | $26.3 \pm 2.5$ | $0.0 \pm 0.0$ |
| | RRWP - Dual | $\mathbf{1.8 \pm 0.5}$ | $\mathbf{99.5 \pm 1.0}$ |
| | MagLap - Double | $14.2 \pm 1.5$ | $8.0 \pm 7.5$ |
| | MagLap - Dual | $\underline{1.9 \pm 0.3}$ | $77.0 \pm 7.6$ |

## 6 RELATED WORK

**Graph generative methods**   Initial methods include *auto-regressive* models (You et al., 2018; Liao et al., 2019), which grow the graph progressively by inserting nodes and edges, but require node ordering for feasibility. On the other hand, *one-shot* graph generative models such as VAEs (Kipf & Welling, 2016; Simonovsky & Komodakis, 2018; Jin et al., 2018; Ma & Zhang, 2021; Vignac & Frossard, 2022), GANs (De Cao & Kipf, 2018; Krawczuk et al., 2021; Martinkus et al., 2022), and normalizing flows (Kaushalya et al., 2019; Wehenkel & Louppe, 2021; Lippe & Gavves, 2021; Luo et al., 2021b) can directly generate full graphs in a single forward pass, thereby eliminating the need for predefined node ordering. Graph diffusion models also belong to this class, with the initial approaches consisting of adaptations of continuous state-space discrete-time diffusion frameworks (Sohl-Dickstein et al., 2015; Ho et al., 2020; Song & Ermon, 2019) to the graph setting. These include works such as EDP-GNN (Niu et al., 2020), DGSM (Luo et al., 2021a), and GeoDiff (Xu et al., 2022). Later, the discrete state-space diffusion framework (Austin et al., 2021a) was adopted, with DiGress (Vignac et al., 2023a) and MCD (Haefeli et al., 2022) operating in discrete time. On the other hand, methods that operate in continuous time (Song et al., 2021; Campbell et al., 2022) were introduced, including GDSS (Jo et al., 2022) and GruM (Jo et al., 2024) in continuous state-spaces and DisCo (Xu et al., 2024) and Cometh (Siraudin et al., 2024) for discrete state-spaces. Recent approaches combine the sequential modeling of auto-regressive methods with the global denoising of diffusion (Zhao et al., 2024). Finally, DeFoG (Qin et al., 2025a) recently achieves state-of-the-art performance by leveraging discrete flow matching (Campbell et al., 2024; Gat et al., 2024).

**Directed graph generation**   The vast majority of works in this area focus solely on DAG generation, with research including autoencoders (Zhang et al., 2019), autoregressive methods (Li et al., 2022) and, more recently, combining diffusion in different steps of an auto-regressive process (Li et al., 2025). However, they require the input DAGs to be topologically ordered, which can result in higher computational cost. Other approaches (Asthana et al., 2024; An et al., 2024) use ideas from discrete diffusion for Neural Architecture Search (NAS), but they are tailored to the specific application and, therefore, to DAGs. Finally, Law et al. (2025) are the first to propose a general digraph generation method from an auto-regressive perspective. Unfortunately, to the best of our knowledge, its implementation is not publicly available, limiting its accessibility for study and evaluation.

## 7 CONCLUSION AND FUTURE DIRECTIONS

We propose DIRECTO, a discrete flow matching-based method for directed graph generation. It combines directionality-aware positional encodings with a dual-attention mechanism that handles both source-to-target and target-to-source dependencies, overcoming the limitations of naïve extensions from undirected models. We also introduce a dedicated benchmarking framework to address the lack of standardized evaluation. Empirical results show that DIRECTO outperforms existing baselines on synthetic and real-world datasets, successfully generating directed structures and preserving key properties, such as acyclicity, without explicit constraints, highlighting its generality and robustness.

**Limitations and future directions**   Despite these promising results, limitations remain. First, scalability can be further improved in handling larger datasets and graph sizes (a direct result of the combinatorial complexity of digraphs). Incorporating scalable strategies, such as sparsity-aware attention (Qin et al., 2025b), hierarchical (Bergmeister et al., 2024; Jang et al., 2024), or latent diffusion (Yang et al., 2024), could help extend DIRECTO at scale. Second, although our current setup includes classifier-free conditional generation, exploring alternative conditional strategies could further improve performance. Third, while DIRECTO can learn structural properties such as acyclicity implicitly, enforcing such constraints (e.g., through PRODIGY (Sharma et al., 2024) or ConStruct (Madeira et al., 2024)) could further improve controllability and domain-specific validity. Finally, although the proposed graph transformer is used here for graph-to-graph generation, it can be readily adapted to discriminative tasks such as link prediction or node classification. Given its performance in the generative setting, it would be interesting to study how the architecture transfers to these alternative setups. Exploring these directions would enhance the applicability and impact of digraph generative models and our proposed architecture, particularly on new real-world scenarios. Appendix K includes a detailed discussion of limitations and future work.

## ACKNOWLEDGEMENTS

This work was supported by the Swiss National Science Foundation (SNSF) under grant 10001445.

## REPRODUCIBILITY STATEMENT

To ensure the reproducibility of our results, we have taken the following steps. The implementation of our method is provided in a GitHub repository, including all code and scripts necessary to reproduce the experiments and details on how to run it. In addition, we detail the proposed algorithm in Appendix D, while Appendix G provides full descriptions of the baselines, training setup, and hyperparameters employed. We also report the computational resources used and the runtime required for all experiments. Together, these materials are intended to enable researchers to fully replicate and build upon our work.

## ETHICS STATEMENT

The objective of this work is to advance methodologies for the generation of directed graphs by introducing architectural mechanisms that explicitly model directionality and asymmetric relationships. Directed graphs are central to a wide range of applications, including causal inference, traffic modeling, and biological network analysis. Accordingly, improvements in directed graph generation have the potential to support progress in scientific research, decision-making systems, infrastructure design, or causal discovery.

The proposed framework strengthens the ability of generative models to handle directed graphs, and, at the moment, we do not identify any immediate societal risks associated with its deployment in its current form. Moreover, although the method is capable of generating structured and semantically meaningful graphs, the scale and complexity of these outputs remain limited, which may difficult direct applicability in high-impact domains such as clinical research or policy modeling at this stage.

## USE OF LARGE LANGUAGE MODELS

Large Language Models (LLMs) were used as a writing assistant tool in the preparation of this manuscript. Specifically, they were employed to aid in writing, checking spelling, and editing for clarity. Nonetheless, all text produced with the assistance of LLMs was carefully reviewed, verified, and revised by the authors to ensure accuracy and appropriateness. The use of LLMs was limited to these support functions, and the authors take full responsibility for the final content of the paper.

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

# Appendix

# Table of Contents

# A  DUAL ATTENTION

In this section, we provide a detailed description of the dual attention mechanism used in our model, which effectively integrates both target-to-source and source-to-target information to enhance the graph generation process.

We start with the stacked tensors of node $\boldsymbol{X} \in \mathbb{R}^{B \times N \times d_x}$, edge $\boldsymbol{E} \in \mathbb{R}^{B \times N \times N \times d_e}$ and global $\boldsymbol{y} \in \mathbb{R}^{B \times d_y}$ features, where $B$ is the batch size, $N$ is the number of nodes of the biggest graph in the batch, and $d_x$, $d_e$, and $d_y$ are the node, edge, and global feature dimensions respectively. To handle different node sizes in a batch, we use a binary node mask $\boldsymbol{M} \in \{0, 1\}^{B \times N \times 1}$ that accounts for the presence of nodes in each graph.

First, we begin by projecting the queries, keys, and values for the source and the target directions. These are then projected and reshaped into $n_h$ heads of dimension $d_q = d_x/n_h$, thus obtaining for each node:

$$\boldsymbol{Q}_\mathrm{S} = \mathrm{reshape}(\boldsymbol{M} \odot \boldsymbol{W}_\mathrm{Q}^\mathrm{S} \boldsymbol{X}), \quad \boldsymbol{K}_\mathrm{S} = \mathrm{reshape}(\boldsymbol{M} \odot \boldsymbol{W}_\mathrm{K}^\mathrm{S} \boldsymbol{X}), \quad \boldsymbol{V}_\mathrm{S} = \mathrm{reshape}(\boldsymbol{M} \odot \boldsymbol{W}_\mathrm{V}^\mathrm{S} \boldsymbol{X}), \quad (10)$$

$$\boldsymbol{Q}_\mathrm{T} = \mathrm{reshape}(\boldsymbol{M} \odot \boldsymbol{W}_\mathrm{Q}^\mathrm{T} \boldsymbol{X}), \quad \boldsymbol{K}_\mathrm{T} = \mathrm{reshape}(\boldsymbol{M} \odot \boldsymbol{W}_\mathrm{K}^\mathrm{T} \boldsymbol{X}), \quad \boldsymbol{V}_\mathrm{T} = \mathrm{reshape}(\boldsymbol{M} \odot \boldsymbol{W}_\mathrm{V}^\mathrm{T} \boldsymbol{X}), \quad (11)$$

where $\boldsymbol{W}_\mathrm{Q}^\mathrm{S}, \boldsymbol{W}_\mathrm{K}^\mathrm{S}, \boldsymbol{W}_\mathrm{V}^\mathrm{S}, \boldsymbol{W}_\mathrm{Q}^\mathrm{T}, \boldsymbol{W}_\mathrm{K}^\mathrm{T}, \boldsymbol{W}_\mathrm{V}^\mathrm{T}$ are the learned linear projections. The final shape of each of the elements after the reshaping is $B \times N \times n_h \times d_q$.

Then we compute the dual attention, both from source to target and from target to source:

$$\boldsymbol{Y}_\mathrm{ST}[i, j] = \frac{\boldsymbol{Q}_\mathrm{S}[i] \cdot \boldsymbol{K}_\mathrm{T}[j]}{\sqrt{d_q}}, \quad \boldsymbol{Y}_\mathrm{TS}[i, j] = \frac{\boldsymbol{Q}_\mathrm{T}[i] \cdot \boldsymbol{K}_\mathrm{S}[j]}{\sqrt{d_q}}. \quad (12)$$

with $i, j$ the index nodes and each $\boldsymbol{Y} \in \mathbb{R}^{B \times N \times N \times n_h \times d_q}$.

To incorporate edge features into the attention mechanism, we apply FiLM-style modulation (Perez et al., 2018) using additive and multiplicative projections $\boldsymbol{W}_\mathrm{mul}, \boldsymbol{W}_\mathrm{add}$:

$$\boldsymbol{E}_\mathrm{mul}^\mathrm{S} = \mathrm{reshape}(\boldsymbol{W}_\mathrm{mul}^\mathrm{S}(\boldsymbol{E})), \qquad \boldsymbol{E}_\mathrm{add}^\mathrm{S} = \mathrm{reshape}(\boldsymbol{W}_\mathrm{add}^\mathrm{S}(\boldsymbol{E})), \qquad (13)$$

$$\boldsymbol{E}_\mathrm{mul}^\mathrm{T} = \mathrm{reshape}(\boldsymbol{W}_\mathrm{mul}^\mathrm{T}(\boldsymbol{E}^\top)), \qquad \boldsymbol{E}_\mathrm{add}^\mathrm{T} = \mathrm{reshape}(\boldsymbol{W}_\mathrm{add}^\mathrm{T}(\boldsymbol{E}^\top)), \qquad (14)$$

where we reshape the edge features to $\boldsymbol{E} \in \mathbb{R}^{B \times N \times N \times n_h \times d_q}$. This is done so that they can be added or multiplied to the attention scores $\boldsymbol{Y}_\mathrm{ST}$ and $\boldsymbol{Y}_\mathrm{TS}$ to modify them:

$$\boldsymbol{Y}_\mathrm{ST} \leftarrow \boldsymbol{Y}_\mathrm{ST} \cdot (\boldsymbol{E}_\mathrm{mul}^\mathrm{S} + 1) + \boldsymbol{E}_\mathrm{add}^\mathrm{S}, \quad (15)$$

$$\boldsymbol{Y}_\mathrm{TS} \leftarrow \boldsymbol{Y}_\mathrm{TS} \cdot (\boldsymbol{E}_\mathrm{mul}^\mathrm{T} + 1) + \boldsymbol{E}_\mathrm{add}^\mathrm{T}. \quad (16)$$

**Edge Feature Update**  To get the final update of the edge features, the attention-weighted edge representations are flattened and modulated by global features $\boldsymbol{y}$ using FiLM with additive and multiplicative projections $\boldsymbol{W}_\mathrm{add}^E$ and $\boldsymbol{W}_\mathrm{mul}^E$:

$$\boldsymbol{E}' = \boldsymbol{W}_\mathrm{add}^y(\boldsymbol{y}) + (\boldsymbol{W}_\mathrm{mul}^y(\boldsymbol{y}) + 1) \odot \mathrm{flatten}(\boldsymbol{Y}_\mathrm{ST}), \quad (17)$$

where we flatten over the last dimension. We use $\boldsymbol{Y}_\mathrm{ST}$ as it captures the directional flow from source to target, avoiding ambiguity in the directional influence, and still effectively encoding the influence of source nodes on their targets. Then, the result is followed by an output projection and a residual connection

$$\boldsymbol{E}' = \boldsymbol{W}_\mathrm{out}^E(\boldsymbol{E}'), \quad \boldsymbol{E}' \leftarrow \boldsymbol{E} + \boldsymbol{E}', \quad (18)$$

resulting in the updated $\boldsymbol{E}' \in \mathbb{R}^{B \times N \times N \times d_e}$.

**Node Feature Update**  To perform the node feature update, first we concatenate the dual attention scores and apply softmax:

$$\boldsymbol{A}_\mathrm{aggr} = \mathrm{softmax}\left(\mathrm{concat}(\boldsymbol{Y}_\mathrm{ST}, \boldsymbol{Y}_\mathrm{TS})\right) \in \mathbb{R}^{B \times N \times 2N \times n_h \times d_e}, \quad (19)$$

$$\boldsymbol{V}_\mathrm{aggr} = \mathrm{concat}(\boldsymbol{V}_\mathrm{T}, \boldsymbol{V}_\mathrm{T}) \in \mathbb{R}^{B \times 2N \times n_h \times d_x}, \quad (20)$$

with $\mathrm{concat}()$ being the concatenation over the third dimension for $\boldsymbol{A}_\mathrm{aggr}$ and over the second dimension for $\boldsymbol{V}_\mathrm{aggr}$.

After this, the unified weights are aggregated through weighted summation and the result flattened over the second dimension to produce:

$$\boldsymbol{X}_{\text{aggr}} = \text{attn}_{\text{aggr}} \boldsymbol{V}_{\text{aggr}} \in \mathbb{R}^{B \times N \times n_h \times d_x}. \tag{21}$$

We then FiLM-modulate the attended features with the global vector $\boldsymbol{y}$ with additive and multiplicative projections $\boldsymbol{W}_{\text{add}}^X$ and $\boldsymbol{W}_{\text{mul}}^X$ to produce:

$$\boldsymbol{X}_{\text{mod}} = \boldsymbol{W}_{\text{add}}^X(\boldsymbol{y}) + (\boldsymbol{W}_{\text{mul}}^X(\boldsymbol{y}) + 1) \odot \boldsymbol{X}_{\text{aggr}}, \tag{22}$$

To allow the model to adaptively balance between preserving the original node features and incorporating the updated ones, we introduce a learnable gating mechanism $\boldsymbol{g}X \in [0, 1]$:

$$\boldsymbol{g}_X = \sigma(\boldsymbol{W}_{\text{gate}}^X \odot \text{concat}(\boldsymbol{X}, \boldsymbol{V}_{\text{aggr}})), \tag{23}$$

followed by a gated residual connection:

$$\boldsymbol{X}' = (1 + \boldsymbol{g}_X) \odot \boldsymbol{X} + (1 - \boldsymbol{g}_X) \odot \boldsymbol{X}_{\text{mod}}, \tag{24}$$

where the weights assigned to $\boldsymbol{X}$ and $\boldsymbol{X}_{\text{mod}}$ sum up to 2, ensuring that the overall magnitude remains consistent with the ungated case.

Finally, the result is followed by an output projection:

$$\boldsymbol{X}' = \boldsymbol{W}_{\text{out}}^X(\boldsymbol{X}'), \tag{25}$$

resulting in the updated $\boldsymbol{X}' \in \mathbb{R}^{B \times N \times d_x}$.

**Global Features Update**   The updated global vector is computed by aggregating from the node $\boldsymbol{X}$ and edge $\boldsymbol{E}$ projections:

$$\boldsymbol{y}' = \boldsymbol{y} + W_y(\boldsymbol{y}) + X_{\to y}(\boldsymbol{X}) + E_{\to y}(\boldsymbol{E}), \tag{26}$$

where $X_{\to y}$ and $E_{\to y}$ are PNA layers (Corso et al., 2020) that given node features $\boldsymbol{X} \in \mathbb{R}^{B \times N \times d_x}$ or edge features $\boldsymbol{E} \in \mathbb{R}^{B \times N \times N \times d_e}$ and a learnable weight matrix $\boldsymbol{W}_{\text{PNA}}$, computes

$$\text{PNA}(\boldsymbol{X}) = \text{concat}\left(\max(\boldsymbol{X}), \min(\boldsymbol{X}), \text{mean}(\boldsymbol{X}), \text{std}(\boldsymbol{X})\right) \boldsymbol{W}_{\text{PNA}} \in \mathbb{R}^{4d_x}, \tag{27}$$

$$\text{PNA}(\boldsymbol{E}) = \text{concat}\left(\max(\boldsymbol{E}), \min(\boldsymbol{E}), \text{mean}(\boldsymbol{E}), \text{std}(\boldsymbol{E})\right) \boldsymbol{W}_{\text{PNA}} \in \mathbb{R}^{4d_e}, \tag{28}$$

where the different operations are performed over the features and then concatenation is performed along the feature dimension. Both results then pass through a linear layer resulting in $X_{\to y}(\boldsymbol{X}), E_{\to y}(\boldsymbol{E}) \in \mathbb{R}^{d_y}$.

Once again, we use an output projection

$$\boldsymbol{y}' = \boldsymbol{W}_{\text{out}}^y(\boldsymbol{y}'), \tag{29}$$

which results in the updated global features $\boldsymbol{y}' \in \mathbb{R}^{B \times d_y}$.

This dual attention block is applied $L$ times within the graph transformer.

## B   POSITIONAL ENCODINGS

To effectively capture the structural properties of directed graphs, we explore a range of positional encodings (PEs) that can be incorporated into the transformer architecture. These encodings provide nodes with a sense of position and orientation within the digraph, enabling the model to better reason about directionality. Below, we detail the specific PEs evaluated in our work.

**Laplacian (Lap)**   Traditionally in the undirected case, the eigenvectors of the combinatorial Laplacian are widely used to encode graph structure, relying on the spectral decomposition $\boldsymbol{L} = \boldsymbol{\Gamma} \boldsymbol{\Lambda} \boldsymbol{\Gamma}^{-1}$ where $\boldsymbol{L}$ is the combinatorial Laplacian, $\boldsymbol{\Lambda}$ is the diagonal matrix of eigenvalues, and $\boldsymbol{\Gamma}$ is the matrix of eigenvectors. The unnormalized and normalized Laplacians are defined as

$$\boldsymbol{L}_U = \boldsymbol{D} - \boldsymbol{A}_s, \quad \boldsymbol{L}_N = \boldsymbol{I} - \boldsymbol{D}^{-1/2} \boldsymbol{A}_s \boldsymbol{D}^{-1/2}, \tag{30}$$

where $\boldsymbol{A}_s$ is the adjacency matrix (symmetrized when working with digraphs) and $\boldsymbol{D}$ is the degree matrix. However, when dealing with directed graphs, this symmetrization discards directionality information, as it enforces $\boldsymbol{A}_s =_s^\top$, but it is necessary to guarantee that $\boldsymbol{L}_U$ and $\boldsymbol{L}_N$ are symmetric and positive semi-definite.

**Directed Laplacian (DirLap)**  The directed Laplacian Chung (2005) is defined as

$$L = I - \frac{1}{2}\left(\mathbf{\Phi}^{\frac{1}{2}}P\mathbf{\Phi}^{-\frac{1}{2}} + \mathbf{\Phi}^{-\frac{1}{2}}P\mathbf{\Phi}^{\frac{1}{2}}\right),\tag{31}$$

where $I$ is the identity matrix, $P$ is the transition matrix of the directed graph, and $\mathbf{\Phi}$ is a diagonal matrix containing the Perron vector of $P$. Depending on the chosen walk type, $P$ may correspond to a standard random walk, a lazy random walk, or a random walk with teleportation (PageRank). This construction results in real eigenvalues and eigenvectors and, as a consequence, it cannot jointly encode magnitude and phase information, limiting its ability to capture directional patterns.

**Magnetic Laplacian (MagLap)**  To retain directional information while preserving desirable spectral properties, Geisler et al. (2023) propose the *Magnetic Laplacian*, which introduces a complex-valued phase encoding into the adjacency matrix. Specifically, the (unnormalized and normalized) Magnetic Laplacians are given by:

$$\boldsymbol{L}_{\text{dir},U}^{(q)} = \boldsymbol{D} - (\boldsymbol{A}_s \odot \exp(i\Theta^{(q)})), \quad \boldsymbol{L}_{\text{dir},N}^{(q)} = \boldsymbol{I} - \left((\boldsymbol{D}^{-1/2}\boldsymbol{A}_s\boldsymbol{D}^{-1/2}) \odot \exp(i\Theta^{(q)})\right),\tag{32}$$

where $\odot$ denotes the element-wise (Hadamard) product, $i$ is the imaginary unit, and the phase matrix $\Theta^{(q)}$ is defined as:

$$\Theta_{u,v}^{(q)} = 2\pi q(\boldsymbol{A}_{u,v} - \boldsymbol{A}_{v,u}).\tag{33}$$

The potential parameter $q \geq 0$ controls the strength of the phase shift. For $q = 0$, the Magnetic Laplacian reduces to the classical combinatorial Laplacian. The resulting complex-valued eigenvectors can effectively encode directionality patterns in the graph. We leverage the information by concatenating the real and complex parts of the eigenvectors to the node features, and the eigenvalues to the global graph features.

**Multi-$q$ Magnetic Laplacian**  Recently, Huang et al. (2025) extends the Magnetic Laplacian by introducing a multi-$q$ formulation, which considers as positional encoding stacking together the eigenvalues and eigenvectors of $Q$ distinct Magnetic Laplacians with potentials $q_1, \ldots, q_Q$. This approach enables the recovery of the bidirectional walk profile, a generalization of walk counting in undirected graphs that captures a broader range of directional relationships.

By incorporating information from multiple potentials, the multi-$q$ formulation enhances the representational power for directed structures. In particular, they show that the method is robust to the chosen potentials $q$ chosen. In our model, we leverage this information by concatenating the first $k$ eigenvectors (real and imaginary parts) of each of the $Q$ magnetic Laplacians to the global features, and the corresponding first $k$ eigenvalues to the node features, providing direction-aware structural signals to both levels of representation. As a drawback, this positional encoding results in higher computational costs, as it is necessary to compute several eigenvalue decompositions and, in addition, results in a higher dimensional positional encoding to be processed by the denoising network.

**Directed Relative Random Walk Probabilities (RRWP)**  For undirected graphs, RRWP encodes the likelihood of arriving from one node to another through $k$-step random walks. This is usually expressed via a transition matrix $\boldsymbol{T}^k$, with $\boldsymbol{T} = \boldsymbol{A}\boldsymbol{D}^{-1}$, that encodes the transition probabilities.

For directed graphs, it is possible to consider $\boldsymbol{T} = \boldsymbol{A}\boldsymbol{D}_{out}^{-1}$, where we take into account the degree matrix of the outgoing edges $\boldsymbol{D}_{out}^{-1}$ (Geisler et al., 2023). The $K$-step transition probabilities are then captured by powers of $\boldsymbol{T}$, producing the sequence $[\boldsymbol{I}, \boldsymbol{T}, \boldsymbol{T}^2, \ldots, \boldsymbol{T}^{K-1}]$.

Additionally, in the directed setting, Geisler et al. (2023) suggest it is also informative to model reverse random walks, starting from incoming edges. For this, we can define the reverse transition matrix $\boldsymbol{R} = \boldsymbol{A}^\top\boldsymbol{D}_{\text{in}}^{-1}$, where $\boldsymbol{D}_{\text{in}}$ is the in-degree matrix. This would represent the likelihood of arriving at a given edge starting from another through $k$-step random walks. Analogously, we compute $[\boldsymbol{I}, \boldsymbol{R}, \boldsymbol{R}^2, \ldots, \boldsymbol{R}^{K-1}]$.

To have our final positional encoding, following Geisler et al. (2023), we ensure that graphs are not nilpotent and that the probabilities encoded in the matrices sum up to 1 by adding self-loops to sink nodes during pre-processing. Finally, the full positional encoding concatenates both forward and reverse walk features for a pre-defined $K$, yielding:

$$\text{RRPW}(G) = [\boldsymbol{I}, \boldsymbol{T}, \boldsymbol{T}^2, \ldots, \boldsymbol{T}^{K-1}, \boldsymbol{I}, \boldsymbol{R}, \boldsymbol{R}^2, \ldots, \boldsymbol{R}^{K-1}].\tag{34}$$

We concatenate the PE to the edge features, and additionally concatenate the diagonal elements of each matrix to the node features to further infuse them with the information.

Additionally, to mitigate that for large $k$ the probabilities tend to converge to sink nodes, we optionally incorporate *Personalized PageRank (PPR)* features. The PPR matrix is defined in closed form as:

$$\text{PPR} = p_r(\boldsymbol{I} - (1 - p_r)\boldsymbol{T})^{-1}, \tag{35}$$

where $p_r$ denotes the restart probability.

## C  SAMPLING OPTIMIZATION IN DIRECTO

The decoupling of training and sampling in discrete flow matching enables principled, training-free modifications to the sampling procedure, offering a flexible design space to enhance generation quality across digraph distributions (Qin et al., 2025a). In this section, we detail the three core parameters that govern sampling behavior: (i) time distortion functions, (ii) target guidance intensity, and (iii) stochasticity strength. Each plays a distinct role in shaping the denoising trajectory and contributes to different aspects of generation quality and structural fidelity.

**Time Distortion Functions**  A central part for sampling optimization lies in the application of *time distortion functions*. These functions transform the time variable $t \in [0, 1]$ through a bijective, monotonic mapping $f(t)$, yielding a distorted time variable $t' = f(t)$. While used in both training and sampling, their objectives differ. In training, time distortions skew the sampling distribution over time, concentrating the model's learning capacity on specific denoising intervals. In sampling, they induce variable step sizes along the denoising trajectory, with finer resolution where structural sensitivity is highest (e.g., late-stage edits near $t = 1$).

Formally, the probability density function (PDF) of the transformed time variable $t'$ is:

$$\phi_{t'}(t') = \phi_t(t) \left| \frac{d}{dt'} f^{-1}(t') \right|, \quad \text{where } \phi_t(t) = 1 \text{ for } t \in [0, 1]. \tag{36}$$

We consider the same five representative time distortion functions from Qin et al. (2025a) that yield diverse distributions for $t'$:

- **Identity**: $f(t) = t$ — Uniform time density, baseline behavior.
- **Polydec**: $f(t) = 2t - t^2$ — Increasing density over time, emphasizing late-stage denoising.
- **Cos**: $f(t) = \frac{1 - \cos(\pi t)}{2}$ — Concentrates density at both boundaries.
- **Revcos**: $f(t) = 2t - \frac{1 - \cos(\pi t)}{2}$ — Peaks at intermediate times.
- **Polyinc**: $f(t) = t^2$ — Decreasing density over time, emphasizes early steps.

In the sampling phase, the induced variable step sizes, particularly from functions like *polydec*, allow for finer resolution near the clean data, where structural coherence is most fragile. Empirically, these functions are selected based on dataset-specific properties to improve fidelity and structural constraints without need for retraining.

**Target Guidance**  Another axis for improving sampling efficiency is the modification of the *conditional rate matrix* $R_t$ to better guide the generation trajectory toward the clean target graph $z_1$. This is achieved by incorporating an additive guidance term:

$$R_t(z_t, z_{t+\Delta t} \mid z_1) = R_t^*(z_t, z_{t+\Delta t} \mid z_1) + \omega \cdot \frac{\delta(z_{t+\Delta t}, z_1)}{\mathcal{Z}_t^{>0} \cdot p_{t|1}(z_t \mid z_1)}. \tag{37}$$

Here, $\omega \in \mathbb{R}_+$ controls the strength of the guidance, $\mathcal{Z}_t^{>0}$ is a discrete variable with $Z$ possible positive values, and $\delta(\cdot, \cdot)$ is the Kronecker delta. Intuitively, this formulation prioritizes transitions that directly align with the clean graph.

**Stochasticity Control** The final sampling-time parameter is the *stochasticity coefficient* $\eta$. This parameter scales an auxiliary rate matrix $R_t^{\text{DB}}$ that satisfies the detailed balance condition (Campbell et al., 2024) and adds it to the optimal rate matrix $R_t^*$:

$$R_t^{\eta} = R_t^* + \eta \cdot R_t^{\text{DB}}. \tag{38}$$

The role of $\eta$ is to regulate trajectory stochasticity: higher values promote broader exploration during denoising, potentially correcting suboptimal states. Setting $\eta = 0$ recovers the deterministic path prescribed by $R_t^*$, minimizing the expected number of transitions.

**Sampling Optimization** Together, these sampling parameters constitute a flexible, principled toolkit for tailoring the sampling procedure to diverse graph domains and structural requirements. In our case, we optimize these parameters at sampling time to improve the performance of DIRECTO. In particular, we individually optimize each of them for the main results, and perform the final sampling using the best performing parameters in terms of V.U.N (a detailed description is available in Appendix G.2). and ratio performance. A study of the effect of these parameters on sampling performance can be found in H.9.

## D DIRECTO TRAINING AND SAMPLING ALGORITHMS

This appendix provides detailed pseudocode for the training and sampling procedures used in the DIRECTO framework. Alg. 1 outlines the training loop, where a digraph is progressively noised and a denoising model is optimized to reconstruct the original graph.

We consider the original data distribution $\boldsymbol{p}_1$ as well as the time distribution $\mathcal{T}$ which in our case is set to be the uniform distribution. In addition, $f^\theta$ refers to the denoising network, in our case the graph transformer with dual attention.

---

**Algorithm 1** DIRECTO Training

---

1: **Input:** Graph dataset $\{G^1, \ldots, G^M\} \sim \boldsymbol{p}_1$
2: **while** $f_\theta$ not converged **do**
3:     Sample $G \sim \boldsymbol{p}_1$
4:     Sample $t \sim \mathcal{T}$
5:     Iteratively sample $G_t \sim \boldsymbol{p}_{t|1}(G_t|G)$          ▷ Noising process
6:     $h \leftarrow \text{PosEnc}(G_t)$          ▷ Positional encoding
7:     $\boldsymbol{p}_{1|t}^\theta(\cdot|G_t) \leftarrow f_\theta(G_t, h, t)$          ▷ Denoising prediction with dual attention
8:     loss $\leftarrow \text{CE}_\lambda(G, \boldsymbol{p}_{1|t}^\theta(\cdot|G_t))$
9: **end while**

---

Alg. 2 describes the generative sampling process in which new digraphs are synthesized by iteratively denoising from an initial random digraph structure. Again, we consider $\boldsymbol{p}_0 = \boldsymbol{p}_{noise}$ the predefined noise distribution, $f^\theta$ the denoising graph transformer with dual attention, $\boldsymbol{p}_{1|t}^\theta$ the denoising predictions, and $\boldsymbol{p}_{t+\Delta t|t}^\theta$ as the update rule, see Equation (42).

---

**Algorithm 2** DIRECTO Sampling

---

1: **Input:** # graphs to sample $S$
2: **for** $i = 1$ to $S$ **do**
3:     Sample $N$ from train set          ▷ # nodes
4:     Sample $G_0 \sim \boldsymbol{p}_0(G_0)$
5:     **for** $t = 0$ to $1 - \Delta t$ with step $\Delta t$ **do**
6:         $h \leftarrow \text{PosEnc}(G_t)$          ▷ Positional encoding
7:         $\boldsymbol{p}_{1|t}^\theta(\cdot|G_t) \leftarrow f_\theta(G_t, h, t)$          ▷ Denoising prediction with dual attention
8:         $G_{t+\Delta t} \sim \boldsymbol{p}_{t+\Delta t|t}^\theta(G_{t+\Delta t}|G_t)$          ▷ Update rule
9:     **end for**
10:     Store $G_1$ as $G^i$
11: **end for**

---

# E    DATASET DESCRIPTIONS

In this section, we provide further detailed information about the datasets described in Section 4 and used for the experiments in Section 5.

## E.1    SYNTHETIC DATASETS

We outline the mathematical formulation of each graph generation strategy used in our experiments. All graphs are generated as adjacency matrices $\boldsymbol{A} \in \{0,1\}^{n \times n}$ (where $\boldsymbol{A}_{ij} = 1$ denotes a directed edge from node $i$ to node $j$), and preprocessed to adapt to the structure required by the model.

For all datasets, we generate a total of 200 graphs, split as: 128 train, 32 validation, 40 test. A study on the effect of the dataset size on the quality of the generations can be found in Appendix H.7. Furthermore, the dataset statistics can be found in Table 3.

Table 3: Synthetic dataset statistics.

| Dataset | Min. nodes | Max. nodes | Avg. nodes | Min. edges | Max. edges | Avg. edges | #Train | #Val | #Test |
|---|---|---|---|---|---|---|---|---|---|
| Erdos-Renyi (ER) | 20 | 80 | 46 | 223 | 3670 | 1446 | 128 | 32 | 40 |
| SBM | 44 | 175 | 106 | 340 | 3008 | 1426 | 128 | 32 | 40 |
| ER (DAG) | 20 | 80 | 49 | 103 | 1892 | 762 | 128 | 32 | 40 |
| Price (DAG) | 64 | 64 | 64 | 132 | 246 | 197 | 128 | 32 | 40 |

**Erdős-Renyi (Erdös & Rényi, 1959)**   We generate graphs with a variable number of nodes between $n = 20$ and $n = 80$. For a number of nodes $n$, edges are sampled independently with a fixed probability $p = 0.6$. The adjacency matrix $\boldsymbol{A}$ is generated by:

$$\boldsymbol{A}_{ij} \sim \text{Bernoulli}(p), \quad \forall i \neq j$$

To generate a second dataset of DAGs, we enforce a lower-triangular structure (i.e., edges only from higher-indexed to lower-indexed nodes), which results in graphs generated with a probability $p = 0.3$.

**Price's model (de Solla Price, 1965)**   This is the directed version of the Barabási-Albert or preferential attachment model, which simulates the growth of citation networks. We build DAGs sequentially for a fixed number of nodes $N = 64$. Each new node $i$ forms $m = \log_2(N)$ edges to existing nodes $j$ with probability proportional to their degree:

$$\mathbb{P}(i \rightarrow j) \propto \deg(j).$$

Since the number of nodes is fixed to $N = 64$, mean out-degree results in $m = 6$. In practice, we implement this "bag" of nodes that replicates existing connections. For each new node $i$, we sample $m$ destination nodes from the bag. Then, for each selected node $j$, we set $\boldsymbol{A}_{ij} = 1$ and add $i$ and the selected nodes to the bag. This ensures a directed acyclic graph (DAG) due to the order of node addition.

**Stochastic Block Model (Holland et al., 1983)**   We create $K$ communities with sizes $\{n_1, n_2, \ldots, n_K\}$. We set the number of communities between $K = 2$ and $K = 5$, and the number of nodes per community between 20 and 40. The probability of an edge between any two nodes depends on their community assignments

$$\boldsymbol{A}_{ij} \sim \text{Bernoulli}(P_{z_i z_j})$$

where $z_i \in \{1, \ldots, K\}$ is the block assignment of node $i$ and $P \in [0,1]^{K \times K}$ is a matrix of intra- and inter-community connection probabilities. In particular, we use

$$P_{kk} = p_{\text{intra}} = 0.3, \quad P_{kl} = p_{\text{inter}} = 0.05 \text{ for } k \neq l$$

to generate a block-structured, directed graph.

## E.2 TPU TILES

In this dataset introduced by Phothilimthana et al. (2023), each DAG represents a computation within a machine learning workload, such as a training epoch or an inference step. Each of the 6301 datapoints includes a computational graph, a specific compilation configuration, and the execution time of the graph when compiled with that configuration. The dataset features different model architectures, such as ResNets, EfficientNets, Masked R-CNNs, and Transformers, with graphs of up to ∼ 400 nodes.

Table 4: TPU Tiles dataset statistics.

| Dataset | Min. nodes | Max. nodes | Avg. nodes | Min. edges | Max. edges | Avg. edges | # Graphs |
|---|---|---|---|---|---|---|---|
| Train | 2 | 394 | 41 | 1 | 711 | 43 | 5040 |
| Validation | 2 | 113 | 41 | 1 | 123 | 43 | 630 |
| Test | 2 | 154 | 41 | 1 | 249 | 44 | 631 |

We adopt the processing and splits from Li et al. (2025). Details on dataset statistics for the three different splits can be found in Table 4.

## E.3 VISUAL GENOME

The Visual Genome dataset (Krishna et al., 2017) aims at bridging computer vision and natural language understanding by providing richly annotated images. It comprises over 100,000 images annotated with object labels, attributes, relationships, which allows to capture the presence of objects but also their attributes and interactions. It has become a standard benchmark for tasks such as scene graph generation, visual question answering, and grounded language understanding.

To build the directed graph dataset, we set 3 types of nodes: objects, attributes, and relationships, with the actual node class being the text label. Then, we link them via directed edges according to the visual rules provided in the original data and taking into account that:

1. **Objects** have outgoing edges to attributes and relationships and incoming edges from relationships.
2. **Relationships** have outgoing edges to objects and incoming edges from objects.
3. **Attributes** have no outgoing edges and incoming edges from objects.

Once we built the digraphs, we select a relevant subset from this raw dataset by keeping graphs with 20 to 40 nodes, and with a minimum of 25 edges per graph. Then, for each graph, we only consider objects, attributes, and relationships that are in the top-20 in terms of prevalence in the full dataset, ending with 60 node classes. We randomly split the graphs into 203 train, 52 validation and 63 test graphs, with detailed statistics of each split available in Table 5. The total number of cyclic graph in the dataset is 119 (∼37.5% of the digraphs).

Table 5: Visual Genome dataset statistics.

| Dataset | Min. nodes | Max. nodes | Avg. nodes | Min. edges | Max. edges | Avg. edges | # Graphs |
|---|---|---|---|---|---|---|---|
| Global | 21 | 40 | 34 | 25 | 47 | 28 | 317 |
| Train | 21 | 40 | 33 | 25 | 47 | 28 | 203 |
| Validation | 24 | 40 | 34 | 25 | 40 | 28 | 51 |
| Test | 21 | 40 | 34 | 25 | 36 | 28 | 63 |

The selected top-20 objects, relationships, and attributes are:

- **Objects:** ['window', 'tree', 'man', 'shirt', 'wall', 'person', 'building', 'ground', 'sign', 'light', 'sky', 'head', 'leaf', 'leg', 'hand', 'pole', 'grass', 'hair', 'car', 'cloud']

- **Relationships:** ['on', 'has', 'in', 'of', 'wearing', 'with', 'behind',
  'holding', 'on top of', 'on a', 'near', 'next to', 'has a', 'on',
  'under', 'by', 'of a', 'wears', 'above', 'sitting on']
- **Attributes:** ['white', 'black', 'blue', 'green', 'red', 'brown',
  'yellow', 'small', 'large', 'wooden', 'gray', 'silver', 'metal',
  'orange', 'grey', 'tall', 'long', 'dark', 'pink', 'clear']

## F    FURTHER DETAILS ON EVALUATION METRICS

In this section, we detail the statistical procedures used to evaluate the graphs generated by our model.
We detail how we compute the validity metrics for the different datasets, as well as how we make the
validity and uniqueness computation more efficient.

### F.1    VALIDITY METRICS

**Erdős-Renyi**    Given a graph $G = (V, E)$ with $n = |V|$ nodes and $m = |E|$ edges, we want
to determine whether the graph likely originates from an Erdős–Rényi model $G(n, p)$ with edge
probability $p$ using a Wald test (Held & Sabanés Bové, 2013). For that, we follow the following steps
distinguishing the cases in which the graphs are DAGs or not:

1. **Empirical edge probability:** we compute the number of possible edges

$$m_{\max} = \begin{cases} \frac{n(n-1)}{2} & \text{if the graph acyclic} \\ n(n-1) & \text{if the graph is not acyclic} \end{cases}$$

   and then estimate the empirical probability of edge presence $\hat{p} = \frac{m}{m_{\max}}$.

2. **Wald test statistic:** the Wald statistic tests the null hypothesis $H_0 : \hat{p} = p$, where $p$ is the
   expected edge probability:

$$W = \frac{(\hat{p} - p)^2}{\hat{p}(1 - \hat{p}) + \varepsilon}$$

   and where we add a small regularization $\varepsilon = 10^{-6}$ to prevent division by zero.

3. $p$**-value computation**: assuming the null hypothesis, the Wald statistic $W$ asymptotically
   follows a Chi-squared distribution with 1 degree of freedom:

$$p\text{-value} = 1 - F_{\chi^2}(W; \, df = 1)$$

   where $F_{\chi^2}$ is the cumulative distribution function (CDF) of the Chi-squared distribution.

In the rare limit case where graphs contain only one node, we consider that the graph does not follow
the distribution.

**Stochastic Block Model**    To assess whether a graph $G = (V, E)$ conforms to a Stochastic Block
Model (SBM), we recover the block structure and probability parameters. The test then computes
Wald statistics for intra- and inter-block edge probabilities.

1. **Block assignment via model inference:** Given the adjacency matrix $A \in \{0, 1\}^{n \times n}$ of a
   graph $G$, we use an algorithm (Peixoto, 2014) to infer the stochastic block model (block
   assignment) from a given network:

$$\text{Infer } z : V \rightarrow \{1, \ldots, B\} \quad \text{using } \min \mathcal{L}(G, z)$$

   where $B$ is the number of non-empty blocks found and $\mathcal{L}$ is the description length. The
   model can be further refined using Markov-Chain Monte Carlo (MCMC) for $T$ timesteps.

2. **Estimating intra- and inter-block probabilities:** let $N_i$ be number of nodes in block $i$
   and $E_{ij}$ the number of edges between block $i$ and block $j$ recovered by the algorithm in the
   previous step. Then the estimated probabilities for directed SBM graphs are:

$$\hat{p}_{\text{intra},i} = \frac{E_{ii}}{N_i(N_i - 1) + \varepsilon}$$

$$\hat{p}_{\text{inter},ij} = \frac{E_{ij}}{N_i N_j + \varepsilon}, \quad i \neq j$$

where $\varepsilon = 10^{-6}$ is a small regularization constant.

3. **Wald test statistic:** we compare the empirical estimates to expected values $p_{\text{intra}}$ and $p_{\text{inter}}$ using a Wald statistic:

$$W_{ii} = \frac{(\hat{p}_{\text{intra},i} - p_{\text{intra}})^2}{\hat{p}_{\text{intra},i}(1 - \hat{p}_{\text{intra},i}) + \varepsilon}$$

$$W_{ij} = \frac{(\hat{p}_{\text{inter},ij} - p_{\text{inter}})^2}{\hat{p}_{\text{inter},ij}(1 - \hat{p}_{\text{inter},ij}) + \varepsilon}, \quad i \neq j$$

4. $p$-**value computation**: assuming the null hypothesis (i.e., estimated probabilities match expected ones), each Wald statistic follows a Chi-squared distribution with 1 degree of freedom:

$$p_{ij} = 1 - F_{\chi^2}(W_{ij}; df = 1),$$

and therefore the overall p-value can be computed as the mean of all $p_{ij}$

$$p\text{-value} = \frac{1}{B^2} \sum_{i=1}^{B} \sum_{j=1}^{B} p_{ij}$$

**Price's model**  To assess whether a graph follows a Price's model, we use a nonparametric two-sample Kolmogorov–Smirnov (KS) test:

1. **Computing degree distribution:** let $D = \{d_1, d_2, \ldots, d_n\}$ be the degree sequence of the nodes in $G$. To reduce noise and ensure a more stable distribution, ignore nodes with degree $\leq 1$ and end with the distribution $D' = \{d_i \in D \mid d_i > 1\}$.

2. **Synthetic graph generation:** we construct a synthetic graph $G_{\text{BA}}$ using the Barabási–Albert model with the same number of nodes: $G_{\text{BA}} \sim \text{BA}(n, m)$, where $n = |V|$ is the number of nodes and $m = 6$ is the number of edges each new node adds during attachment, fixed to the same value as in our data generation. The degree sequence of the synthetic graph is $D_{\text{BA}}$.

3. **Kolmogorov-Smirnov Two-Sample test:** we perform a two-sample Kolmogorov–Smirnov (KS) test to compare the empirical distribution functions of $D'$ and $D_{\text{BA}}$

$$p\text{-value} = 1 - \text{KS}_{2\ sampled}(D', D_{\text{BA}})$$

which evaluates the null hypothesis: $H_0 : D' \sim D_{\text{BA}}$.

**TPU Tiles**  For the TPU Tiles dataset, we report the percentage of valid Directed Acyclic Graphs (DAGs) as our primary validity metric, reflecting our focus on accurately reconstructing acyclic computational graphs.

**Visual Genome**  As seen in Appendix E.3, by construction, object nodes can have outgoing edges to relationship or attribute nodes but only receive incoming edges from relationship nodes. Relationship nodes only have incoming edges from object nodes and send outgoing edges to objects. Finally, attributes can only receive edges from object nodes and do not have any outgoing edges. Therefore, to measure digraph validity we evaluate that the generated graphs verify these constraints.

### F.2  UNIQUENESS AND NOVELTY

To evaluate the quality and novelty of generated graphs, we employ the usual metrics based on graph isomorphism. In particular we measure the fraction of isomorphic graphs from the sampled set to the train set (uniqueness) and the fraction of graphs from the sampled set that are not isomorphic to any other in the same sampled set (novelty).

However, since exact graph isomorphism testing can be computationally expensive, particularly for large or dense graphs such as the one in the TPU Tiles dataset, we incorporate a timeout mechanism. Specifically, if an isomorphism check does not complete within 5 seconds, it is treated as a timeout and the graph is conservatively assumed to be non-isomorphic.

In Table 10, we see that in three of the model configurations the computations timed out. We report the average percentage of graphs that could not be tested, which was 2.5% for DIGRESS, 0.8% for DEFOG, and 2% for DIRECTO RRWP.

### F.3 MAXIMUM MEAN DISCREPANCY METRICS

For the different metrics, we adapt previous work from Martinkus et al. (2022); Law et al. (2025). We propose to measure both out-degree and in-degree distributions, as well as clustering, spectre and walvelet. To measure the *clustering* coefficient, we compute the distribution of directed local clustering coefficients using `networkx` function `clustering()`, which supports digraphs. For the spectral features (*spectre* and *wavelet*), we derive the spectral features from the directed Laplacian, as described in Section 4.

In addition, the *orbit* metric that was also computed in Martinkus et al. (2022) has not been adapted to the directed setting as it relies on the Orbit Counting Algorithm (ORCA) which counts graphlets in networks but is only implemented in the undirected setting. Therefore, adapting this metric to the directed setting remains an open challenge.

### F.4 JOINT NODE-EDGE DISTRIBUTIONAL METRICS

Downstream evaluations in graph generative models usually assume conditional graph generation and alignment with auxiliary inputs. Moreover, standard evaluations in domains such as Scene Graph Generation (Li et al., 2024; Lorenz et al., 2024) assume access to input images to compare generated graphs against ground truth images using retrieval metrics such as Precision@K and Recall@K. These require alignment with specific input conditions, which is not possible in our formulation as there are no ground truth images for the generated graphs. Instead, we propose metrics to directly evaluate generative performance by assessing joint node-edge distributional coverage, an essential aspect for labeled graphs such as NAS and scene graphs.

- **Node type distribution MMD:** To evaluate coverage of node class distributions, we compute normalized histograms over node labels for each graph and calculate the MMD score between generated and reference sets, which captures how well categorical proportions are preserved across generated graphs.

- **Triplet-based precision and recall:** Inspired by retrieval-style evaluations in Scene Graph Generation (Li et al., 2024; Lorenz et al., 2024), we extract semantic tuples from graphs. For scene graphs, we use (subject, attribute) pairs and (object, relation, object) triplets. For the TPU Tiles dataset, we extract element–element pairs. We compare the tuples from generated graphs with those in the test set and compute:

$$\text{Precision} = \frac{|\text{correctly generated tuples}|}{|\text{generated tuples}|} \quad \text{Recall} = \frac{|\text{correctly generated tuples}|}{|\text{test tuples}|}$$

- **Embedding-based distances (FID and RBF):** Following Thompson et al. (2022), we compute RBF-kernel MMD and Fréchet Inception Distance (FID) over embeddings extracted from a random GNN trunk. These embeddings encode both node labels and structural information, thus capturing *joint node-edge distributions*.

### F.5 DOWNSTREAM TASKS METRICS

In addition to the distributional alignment evaluations, we also provide a evaluations on downstream tasks for the TPU Tiles dataset. Here, we train a *conditional* generative model and follow the evaluation protocol of LayerDAG (Li et al., 2025). Specifically, we employ their ML-based surrogate cost models to approximate the computational performance of generated architectures. This ensures direct comparability with prior work.

# G  EXPERIMENTAL DETAILS

## G.1  DETAILS ON BASELINES

**Maximum Likelihood Estimation (MLE)**   We define a simple baseline by estimating empirical probabilities from the training dataset via maximum likelihood estimation. The following distributions are computed:

1. **Number of Nodes Distribution:** Let $n$ denote the number of nodes in a graph. The empirical distribution over $n$ is given by:

$$P(n) = \frac{\text{Number of training graphs with } n \text{ nodes}}{\text{Total number of training graphs}}.$$

2. **Node Class Distribution:** Let $c$ be a node class. The empirical node class distribution is:

$$P(v = c) = \frac{\text{Number of nodes of class } c}{\text{Total number of nodes in all training graphs}}.$$

3. **Edge Type Distribution Conditioned on Node Pairs:** Let $(v_i, v_j)$ be a pair of nodes such that $v_i$ has class $c_a$ and $v_j$ has class $c_b$. Let $e_{ij} \in \{1, \ldots, K\}$ denote the edge type between them (with $K$ total edge types). The empirical conditional distribution is:

$$P(e_{ij} = k \mid v_i = c_a, v_j = c_b) = \frac{\text{Number of edges of type } k \text{ between } (c_a, c_b)}{\sum_{k'=1}^{K} \text{Number of edges of type } k' \text{ between } (c_a, c_b)}.$$

These distributions are stored and later used to construct graphs by first sampling the number of nodes, then sampling node types independently, and finally sampling edge types between each node pair according to the conditional edge distribution.

**D-VAE (Zhang et al., 2019)**   This autoregressive method encodes and decodes directed acyclic graphs (DAGs) using an asynchronous message passing scheme. Unlike standard graph neural networks (GNNs), which apply simultaneous message passing across all nodes and updates them all at once, D-VAE updates nodes sequentially, allowing it to capture the computational flow within the graph rather than just its structure. During generation, the DAG is built one node at a time in topological order, with edges always directed toward newly added nodes ensuring the resulting graph remains acyclic. To enable this process, input graphs must be topologically sorted before being processed by the model.

To reproduce this baseline, we adapted the code available in their GitHub and arranged both the TPU Tiles and ER-DAG datasets to match the input format expected by the original model. We used the same hyperparameter configurations as suggested in the original paper to ensure consistency in training, which happened for 500 epochs. After generation, we converted the output DAGs back into the required format for compatibility with our evaluation metrics.

For ER-DAG, we tested two different learning rates $10^{-4}$ and $10^{-5}$, and ended using the model with $10^{-5}$, which resulted in better V.U.N. results with a training time of $\sim$3min per epoch. For TPU Tiles, due to the large size of the dataset and some of the graphs in it, it was impossible to perform a training epoch in a reasonable time, as each of them was predicted to take $\sim$12h.

**LayerDAG (Li et al., 2025)**   This autoregressive diffusion model generates DAGs by converting them into sequences of bipartite graphs, effectively enabling a layer-wise tokenization suitable for autoregressive generation. At each step, a layer-wise diffusion process captures dependencies between nodes that are not directly comparable (i.e., not connected by a path). As with other autoregressive DAG models, the input graphs must be topologically sorted to ensure correct processing.

To reproduce this baseline, we adopted the implementation from GitHub adapted the ER-DAG data to the correct format ordering the graphs topologically, as TPU Tiles was already in the correct format. We kept the hyperparameters from the default configurations, including the number of epochs for each of the elements of the model. After generation, we converted the output DAGs back into the required format for compatibility with our evaluation metrics.

**DiGress (Vignac et al., 2023a)**   To benchmark against DIGRESS, we consider its directed version by simply removing the symmetrization operations. This corresponds to DIRECTO-DD without dual attention and using the Laplacian positional encoding.

**DeFoG (Qin et al., 2025a)**   To benchmark against DEFOG, we consider its directed version by again removing the symmetrization operations. In this case this corresponds to DIRECTO without dual attention and using the RRWP Positional encoding (without PPR).

### G.2   TRAINING SETUP

This section details the training hyperparameters used across all experiments presented in the main text. Unless otherwise specified, the values reported below were used uniformly across all positional encoding variants and ablation settings. In particular, we report the choices for the training, for the positional encodings employed, and for the sampling.

**Training**   The training hyperparameters used in our experiments are summarized in Table 6. These settings were consistently applied across all training runs for both DIRECTO and DIRECTO-DD (with discrete diffusion). We trained each model for up to 10,000 epochs, stopping the runs based on validation performance to account for variations in convergence behavior across datasets.

Table 6: Training and model hyperparameters used in all experiments.

| Hyperparameter | Value |
|---|---|
| **Training Settings** | |
| Learning rate | 0.0002 |
| Weight decay | $1 \times 10^{-12}$ |
| Optimizer | AdamW |
| **Model Settings** | |
| Initial distribution | Marginal |
| Train distortion | Identity |
| Number of diffusion steps | 500 |
| Noise schedule | Cosine |
| Number of layers | 5 |
| Hidden MLP dimensions | $X$: 256, $E$: 128, $y$: 128 |
| Transformer hidden dimensions | $d_x$: 256, $d_e$: 64, $d_y$: 64, $n_{head}$: 8 |
| Feedforward dims | $\dim_{ff_X}$: 256, $\dim_{ff_E}$: 128, $\dim_{ff_y}$: 128 |
| Training loss weights ($\lambda_{\text{train}}$) | [5, 0] |

**Positional encodings**   For RRWP, we set the walk length parameter to $K = 20$. For MagLap with $Q = 5$, we used equidistant potentials $q_5 = (0, 0.1, 0.2, 0.3, 0.4)$. For $Q = 10$, we selected again 10 equidistant potentials $q_{10} = (0.01, \ldots, 0.1)$. In all cases, we then kept the $k = 10$ first eigenvalues and eigenvectors, padding with null values whenever the graphs had less than 10 nodes.

**Conditional generation**   To train with classifier-free guidance conditioning, we first conducted a hyperparameter search to determine the guidance strength with values from 0 to 4, in particular $\omega_{cond} \in (0, 0.25, 0.5, 1, 1.5, 2, 3, 4)$. We ultimately set the parameter to $\omega_{cond} = 0.25$.

**Sampling**   To perform sampling optimization, we conducted a targeted search over the three key hyperparameters: the time distortion function, the target guidance factor ($\omega$), and the stochasticity coefficient ($\eta$). Each hyperparameter was varied independently, while the others were held at their default values (identity distortion and $\omega = \eta = 0$). For computational efficiency, searches were performed using 100 sampling steps for each configuration with five independent sampling runs to ensure robustness. An exception was made for the TPU Tiles dataset, where only a single sampling run was performed per configuration due to the large size of the dataset and some individual graphs.

Based on the outcome of this optimization, we selected the best-performing configurations for each combination of dataset and positional encoding, balancing the trade-off between V.U.N and ratio. The optimal configurations of the results reported in the main paper are summarized in Table 7. For the final evaluation, we performed five sampling runs per configuration using 1000 sampling steps. In all cases, the number of generated graphs matched the size of the test set (40 for synthetic datasets, 63

for Visual Genome), except for TPU Tiles, where we again limited the number of sampled digraphs to 40 due to computational constraints.

Table 7: Optimal sampling hyperparameters for each dataset and positional encoding variant.

| | ER-DAG | | | SBM | | | TPU Tiles | | | Visual Genome | | |
|---|---|---|---|---|---|---|---|---|---|---|---|---|
| **Encoding** | Distortion | $\omega$ | $\eta$ | Distortion | $\omega$ | $\eta$ | Distortion | $\omega$ | $\eta$ | Distortion | $\omega$ | $\eta$ |
| Baseline | Polydec | 0.3 | 10 | Revcos | 0 | 0 | Polydec | 0 | 0 | Cos | 0.1 | 0 |
| RRWP | Polydec | 0.1 | 25 | Polyinc | 0.1 | 50 | Polyinc | 0.3 | 10 | Polydec | 0.3 | 10 |
| MagLap | Polydec | 0 | 100 | Polydec | 0.01 | 50 | Polydec | 0.5 | 100 | Polydec | 0.1 | 50 |

Finally, for the ablations on dual attention and positional encodings, we did not perform sampling optimization on $\eta$ and $\omega$ to avoid the consequent computational cost, but searched for the optimal time distortion. For the scalability experiments, we also did not perform sampling optimization but chose the relevant optimal configuration (from Table 7). For DIRECTO-DD we performed 500 sampling steps in all the different model configurations.

## G.3    RESOURCES AND RUNTIME

All experiments were conducted on a single NVIDIA A100-SXM4-80GB GPU. Table 8 summarizes the runtime of the training and sampling stages across the different datasets. These timings reflect the training time until best performance and the average sampling time per sample, for all the configurations reported in the main results for DIRECTO.

Table 8: Training and sampling runtimes, and VRAM usage, across datasets and models.

| Model | # Train Graphs | Train Time (h) | # Sampled | Sample Time (min/sample) | VRAM (GB) |
|---|---|---|---|---|---|
| **ER-DAG** | | | | | |
| MLE | 128 | 0.001 | 40 | 0.02 | 0.15 |
| D-VAE | 128 | 33.8 | 40 | 1.2 | 9.07 |
| LayerDAG | 128 | 0.2 | 40 | 0.1 | 1.21 |
| DiGress | 128 | 22 | 40 | 0.2 | 5.21 |
| DeFoG | 128 | 13.6 | 40 | 0.1 | 5.71 |
| Directo-DD (RRWP) | 128 | 14.5 | 40 | 0.2 | 5.21 |
| Directo-DD (MagLap $Q$=1) | 128 | 18.2 | 40 | 0.4 | 5.21 |
| Directo-DD (MagLap $Q$=5) | 128 | 20 | 40 | 1.2 | 5.24 |
| Directo-DD (MagLap $Q$=10) | 128 | 26.5 | 40 | 2.5 | 5.30 |
| Directo (RRWP) | 128 | 13.5 | 40 | 0.3 | 5.72 |
| Directo (MagLap $Q$=1) | 128 | 14.4 | 40 | 0.5 | 5.72 |
| Directo (MagLap $Q$=5) | 128 | 16 | 40 | 0.8 | 5.75 |
| Directo (MagLap $Q$=10) | 128 | 25 | 40 | 1.5 | 5.71 |
| **SBM** | | | | | |
| MLE | 128 | 0.002 | 40 | 0.01 | 0.15 |
| DiGress | 128 | 15 | 40 | 0.2 | 26.81 |
| DeFoG | 128 | 23.7 | 40 | 0.3 | 28.46 |
| Directo-DD (RRWP) | 128 | 21.3 | 40 | 0.4 | 26.83 |
| Directo-DD (MagLap $Q$=1) | 128 | 20.4 | 40 | 1.3 | 26.81 |
| Directo-DD (MagLap $Q$=5) | 128 | 21 | 40 | 1.8 | 26.85 |
| Directo-DD (MagLap $Q$=10) | 128 | 20 | 40 | 3.1 | 26.81 |
| Directo (RRWP) | 128 | 13.7 | 40 | 0.5 | 28.34 |
| Directo (MagLap $Q$=1) | 128 | 16 | 40 | 2.1 | 28.45 |
| Directo (MagLap $Q$=5) | 128 | 19 | 40 | 5.1 | 28.49 |
| Directo (MagLap $Q$=10) | 128 | 20 | 40 | 6.0 | 28.46 |
| **TPU Tiles** | | | | | |
| MLE | 5040 | 0.26 | 40 | 0.01 | 0.15 |
| D-VAE | 5040 | OOM | OOM | OOM | OOM |
| LayerDAG | 5040 | 1 | 40 | 0.5 | 1.75 |
| DiGress | 5040 | 26.3 | 40 | 1.9 | 30.56 |
| DeFoG | 5040 | 20.3 | 40 | 1.1 | 30.58 |
| Directo-DD (RRWP) | 5040 | 24.3 | 40 | 2.2 | 30.62 |
| Directo-DD (MagLap $Q$=1) | 5040 | 25 | 40 | 2.4 | 30.56 |
| Directo-DD (MagLap $Q$=5) | 5040 | 31 | 40 | 2.9 | 30.60 |
| Directo (RRWP) | 5040 | 21 | 40 | 1.1 | 30.60 |
| Directo (MagLap $Q$=1) | 5040 | 30 | 40 | 2.8 | 30.58 |
| Directo (MagLap $Q$=5) | 5040 | 30 | 40 | 3.2 | 30.61 |
| **Visual Genome** | | | | | |
| MLE | 203 | 0.02 | 63 | 0.01 | 0.15 |
| DiGress | 203 | 11 | 63 | 1.0 | 11.03 |
| DeFoG | 203 | 13.6 | 63 | 0.9 | 11.22 |
| Directo-DD (RRWP) | 203 | 8 | 63 | 1.0 | 11.25 |
| Directo-DD (MagLap $Q$=1) | 203 | 9 | 63 | 1.4 | 11.03 |
| Directo-DD (MagLap $Q$=5) | 203 | 12 | 63 | 1.5 | 11.06 |
| Directo (RRWP) | 203 | 9 | 63 | 1.1 | 11.23 |
| Directo (MagLap $Q$=1) | 203 | 8 | 63 | 1.2 | 11.22 |
| Directo (MagLap $Q$=5) | 203 | 10 | 63 | 1.3 | 11.25 |

# H ADDITIONAL RESULTS

In this section, we present additional results that offer deeper insights into our model's performance and design choices. Specifically, we analyze the impact of the dual attention mechanism and the choice of positional encoding, compare performance on ER and DAG accuracy metrics, examine how dataset size influences generation quality, provide detailed results for conditional generation, and evaluate the effect of the sampling optimization step in discrete flow matching.

## H.1 EXTENDED RESULTS FOR SYNTHETIC DATASETS

Table 9 reports the performance of DIRECTO across different configurations on synthetic datasets. We observe that the choice of architecture and positional encodings leads to noticeable differences in the distributional alignment metrics (MMD), highlighting the strong performance of DIRECTO. At the same time, uniqueness and novelty remain consistently at 100% across all models, confirming that the V.U.N. ratio is driven uniquely by the validity metric.

Table 9: Directed graph generation performance on synthetic graphs for different configurations of DIRECTO. Results are presented as mean $\pm$ standard deviation across five sampling runs. We considered $Q = 10$ for the MagLap variants.

| Model | Out-degree ↓ | In-degree ↓ | Clustering ↓ | Spectre ↓ | Wavelet ↓ | Ratio ↓ | Valid ↑ | Unique ↑ | Novel ↑ | V.U.N. ↑ |
|---|---|---|---|---|---|---|---|---|---|---|
| | | | | Erdős-Renyi Directed Acyclic Graph (ER-DAG) | | | | | | |
| Training set | 0.0113 | 0.0103 | 0.0355 | 0.0038 | 0.0024 | 1.0 | 99.2 | 100 | 0.0 | 0.0 |
| MLE | $0.0083 \pm 0.0004$ | $0.0089 \pm 0.0003$ | $0.1318 \pm 0.0077$ | $0.0823 \pm 0.0013$ | $0.1162 \pm 0.0018$ | $15.1 \pm 0.2$ | $0.0 \pm 0.0$ | $100 \pm 0.0$ | $100 \pm 0.0$ | $0.0 \pm 0.0$ |
| D-VAE | $0.6158 \pm 0.0160$ | $0.6246 \pm 0.0103$ | $1.0509 \pm 0.00304$ | $0.6160 \pm 0.0315$ | $0.5432 \pm 0.0389$ | $106.6 \pm 5.4$ | $0.0 \pm 0.0$ | $100 \pm 0.0$ | $100 \pm 0.0$ | $0.0 \pm 0.0$ |
| LAYERDAG | $0.0750 \pm 0.0697$ | $0.1773 \pm 0.0722$ | $0.1842 \pm 0.0463$ | $0.0159 \pm 0.0120$ | $0.0218 \pm 0.0160$ | $4.2 \pm 3.2$ | $21.5 \pm 2.7$ | $99.9 \pm 0.0$ | $100 \pm 0.0$ | $21.5 \pm 2.7$ |
| DIGRESS | $0.0138 \pm 0.0035$ | $0.0143 \pm 0.0050$ | $0.1074 \pm 0.0090$ | $0.0073 \pm 0.0017$ | $0.0042 \pm 0.0011$ | $1.9 \pm 0.3$ | $34.0 \pm 4.1$ | $100 \pm 0.0$ | $100 \pm 0.0$ | $34.0 \pm 4.1$ |
| DEFOG | $0.0407 \pm 0.0008$ | $0.0041 \pm 0.0011$ | $0.0460 \pm 0.0052$ | $0.0061 \pm 0.0008$ | $0.0008 \pm 0.0003$ | $1.6 \pm 0.2$ | $75 \pm 2.2$ | $100 \pm 0.0$ | $100 \pm 0.0$ | $75.0 \pm 2.2$ |
| DIRECTO-DD RRWP | $0.0142 \pm 0.0040$ | $0.0129 \pm 0.0034$ | $0.0408 \pm 0.0053$ | $0.0055 \pm 0.0010$ | $0.0048 \pm 0.0014$ | $\underline{1.4} \pm 0.3$ | $79.0 \pm 3.7$ | $100 \pm 0.0$ | $100 \pm 0.0$ | $79.0 \pm 3.7$ |
| DIRECTO-DD MagLap | $0.0145 \pm 0.0040$ | $0.0134 \pm 0.0033$ | $0.0582 \pm 0.0083$ | $0.0063 \pm 0.0015$ | $0.0034 \pm 0.0011$ | $1.5 \pm 0.2$ | $85.0 \pm 9.2$ | $100 \pm 0.0$ | $100 \pm 0.0$ | $85.0 \pm 9.2$ |
| DIRECTO RRWP | $0.0107 \pm 0.0019$ | $0.0104 \pm 0.0012$ | $0.1209 \pm 0.0194$ | $0.0054 \pm 0.0005$ | $0.0043 \pm 0.0008$ | $1.7 \pm 0.1$ | $94.0 \pm 1.0$ | $100 \pm 0.0$ | $100 \pm 0.0$ | $\mathbf{94.0} \pm 1.0$ |
| DIRECTO MagLap | $0.0117 \pm 0.0014$ | $0.0110 \pm 0.0015$ | $0.0711 \pm 0.0120$ | $0.0055 \pm 0.0016$ | $0.0026 \pm 0.0004$ | $\mathbf{1.3} \pm 0.2$ | $92.0 \pm 3.7$ | $100 \pm 0.0$ | $100 \pm 0.0$ | $\underline{92.0} \pm 3.7$ |
| | | | | Stochastic Block Model (SBM) | | | | | | |
| Training set | 0.0031 | 0.0031 | 0.0274 | 0.0027 | 0.0011 | 1.0 | 97.7 | 100 | 0.0 | 0.0 |
| MLE | $0.0020 \pm 0.0002$ | $0.0020 \pm 0.0002$ | $0.1973 \pm 0.0162$ | $0.0087 \pm 0.0003$ | $0.0508 \pm 0.0009$ | $11.6 \pm 0.2$ | $0.0 \pm 0.0$ | $100 \pm 0.0$ | $100 \pm 0.0$ | $0.0 \pm 0.0$ |
| DIGRESS | $0.0037 \pm 0.0018$ | $0.0038 \pm 0.0018$ | $0.0722 \pm 0.0098$ | $0.0045 \pm 0.0004$ | $0.0142 \pm 0.0039$ | $3.9 \pm 0.9$ | $41.5 \pm 5.1$ | $100 \pm 0.0$ | $100 \pm 0.0$ | $41.5 \pm 5.1$ |
| DEFOG | $0.0026 \pm 0.0019$ | $0.0022 \pm 0.0016$ | $0.0813 \pm 0.0058$ | $0.0048 \pm 0.0002$ | $0.0110 \pm 0.0019$ | $4.3 \pm 0.8$ | $37 \pm 6.6$ | $100 \pm 0.0$ | $100 \pm 0.0$ | $37.0 \pm 6.6$ |
| DIRECTO-DD RRWP | $0.0036 \pm 0.0019$ | $0.0036 \pm 0.0018$ | $0.0592 \pm 0.0027$ | $0.0038 \pm 0.0007$ | $0.0027 \pm 0.0009$ | $\underline{1.7} \pm 0.4$ | $81.5 \pm 3.2$ | $100 \pm 0.0$ | $100 \pm 0.0$ | $81.5 \pm 3.2$ |
| DIRECTO-DD MagLap | $0.0037 \pm 0.0019$ | $0.0038 \pm 0.0018$ | $0.0450 \pm 0.0043$ | $0.0038 \pm 0.0006$ | $0.0021 \pm 0.0009$ | $\mathbf{1.5} \pm 0.4$ | $95.5 \pm 3.7$ | $100 \pm 0.0$ | $100 \pm 0.0$ | $95.5 \pm 3.7$ |
| DIRECTO RRWP | $0.0031 \pm 0.0014$ | $0.0028 \pm 0.0013$ | $0.0594 \pm 0.0027$ | $0.0035 \pm 0.0004$ | $0.0027 \pm 0.0009$ | $1.8 \pm 0.5$ | $99.5 \pm 1.0$ | $100 \pm 0.0$ | $100 \pm 0.0$ | $\mathbf{99.5} \pm 1.0$ |
| DIRECTO MagLap | $0.0039 \pm 0.0012$ | $0.0038 \pm 0.0010$ | $0.0654 \pm 0.0052$ | $0.0038 \pm 0.0003$ | $0.0039 \pm 0.0008$ | $2.0 \pm 0.3$ | $96.5 \pm 2.5$ | $100 \pm 0.0$ | $100 \pm 0.0$ | $\underline{96.5} \pm 2.5$ |

## H.2 EXTENDED RESULTS FOR REAL-WORLD DATASETS

Table 10 reports the performance of DIRECTO across different configurations on real-world datasets. While some configurations show evidence of partial memorization, particularly in node and edge distributions, this does not negatively affect the overall generative quality: V.U.N. remains high, and DIRECTO consistently achieves the best or second-best performance across the majority of metrics. These results highlight that, despite occasional replication of training structures, the model maintains strong distributional alignment and generates diverse, valid, and novel graphs.

Table 11 reports the node-label distributional alignment metrics for the two real-world datasets considered. The results highlight that DIRECTO consistently manages to capture node-label distributions accurately. On TPU Tiles, DIRECTO achieve the lowest node-type MMD, indicating strong alignment with the training set, while also maintaining high precision and competitive recall. On Visual Genome, although DIRECTO exhibits slightly lower node-type MMD compared to DeFoG, it attains the highest precision, recall, and RBF MMD, reflecting its ability to capture the underlying distribution. Overall, these results indicate that DIRECTO effectively balances reproducing node label distributions while capturing richer graph structure, with the different positional encoding strategies providing complementary strengths in distributional alignment.

It is relevant to note that the low precision achieved in the training sets is due to the fact that the ground truth set is much larger than the generated set, and therefore there are many pairs and triplets present in the ground truth dataset that have not been generated.

Table 10: Directed graph generation performance on real-world graphs for different configurations of DIRECTO. Results are presented as mean ± standard deviation across five sampling runs. We considered $Q = 5$ for the MagLap variants. OOT indicates that the model could not be run within a reasonable timeframe. $^{(*)}$ indicates that isomorphism tests occasionally timed out, due to the large size of some graphs in this dataset; such cases were excluded from the uniqueness computation.

| Model | Out-degree ↓ | In-degree ↓ | Clustering ↓ | Spectre ↓ | Wavelet ↓ | Ratio ↓ | Valid ↑ | Unique ↑ | Novel ↑ | V.U.N. ↑ |
|---|---|---|---|---|---|---|---|---|---|---|
| | | | | | TPU Tiles | | | | | |
| Training set | 0.0003 | 0.0003 | 0.0007 | 0.0006 | 0.0002 | 1.0 | 100 | 100 | 0.0 | 0.0 |
| MLE | 0.0354 ± 0.0005 | 0.0878 ± 0.0014 | 0.0141 ± 0.0006 | 0.0689 ± 0.0013 | 0.0407 ± 0.0004 | 149.8 ± 0.7 | 24.7 ± 0.0 | 99.9 ± 0.0 | 100 ± 0.0 | 24.7 ± 0.0 |
| D-VAE | OOT | OOT | OOT | OOT | OOT | OOT | OOT | OOT | OOT | OOT |
| LAYERDAG | 0.1933 ± 0.0905 | 0.2225 ± 0.0395 | 0.1512 ± 0.0522 | 0.0501 ± 0.0206 | 0.0765 ± 0.0251 | 413.6 ± 70.1 | 100 ± 0.0 | 99.5 ± 1.0 | 98.5 ± 3.0 | **98.5** ± 3.0 |
| DIGRESS | 0.0084 ± 0.0009 | 0.0726 ± 0.0019 | 0.0020 ± 0.0006 | 0.0033 ± 0.0003 | 0.0018 ± 0.0003 | 57.5 ± 1.7 | 86.1 ± 2.3 | 87.1$^{(*)}$ ± 5.8 | 99.4 ± 0.6 | 70.9 ± 3.4 |
| DEFOG | 0.0099 ± 0.0012 | 0.0794 ± 0.0023 | 0.0042 ± 0.0027 | 0.0040 ± 0.0005 | 0.0017 ± 0.0003 | 63.7 ± 2.6 | 86.3 ± 2.2 | 87.7$^{(*)}$ ± 5.4 | 93.3 ± 2.5 | 72.0 ± 2.4 |
| DIRECTO-DD RRWP | 0.0093 ± 0.0010 | 0.0767 ± 0.0035 | 0.0015 ± 0.0013 | 0.0045 ± 0.0007 | 0.0018 ± 0.0005 | 61.0 ± 2.9 | 86.5 ± 1.9 | 90.3 ± 0.8 | 99.6 ± 0.5 | 76.8 ± 1.9 |
| DIRECTO-DD MagLap | 0.0115 ± 0.0035 | 0.0703 ± 0.0033 | 0.0103 ± 0.0021 | 0.0076 ± 0.0007 | 0.0043 ± 0.0014 | 64.3 ± 5.3 | 86.5 ± 5.1 | 90.5 ± 3.3 | 100 ± 0.0 | 77.0 ± 7.0 |
| DIRECTO RRWP | 0.0133 ± 0.0025 | 0.0859 ± 0.0072 | 0.0136 ± 0.0075 | 0.0086 ± 0.0010 | 0.0038 ± 0.0009 | 75.4 ± 8.1 | 97.0 ± 1.0 | 81.8$^{(*)}$ ± 4.5 | 96.5 ± 1.2 | 77.0 ± 2.9 |
| DIRECTO MagLap | 0.0039 ± 0.0017 | 0.0376 ± 0.0051 | 0.0211 ± 0.0117 | 0.0126 ± 0.0022 | 0.0062 ± 0.0009 | **44.0** ± 7.1 | 90.5 ± 3.3 | 90.5 ± 4.6 | 97.5 ± 3.2 | 80.5 ± 4.6 |
| | | | | | Visual Genome | | | | | |
| Training set | 0.0018 | 0.0030 | 0.0000 | 0.0072 | 0.0036 | 1.0 | 100 | 100 | 0.0 | 0.0 |
| MLE | 0.0607 ± 0.0036 | 0.0474 ± 0.0023 | 0.5342 ± 0.0830 | 0.0535 ± 0.0015 | 0.0399 ± 0.0017 | 17.0 ± 0.6 | 0 ± 0.0 | 100 ± 0.0 | 100 ± 0.0 | 0.0 ± 0.0 |
| DIGRESS | 0.0654 ± 0.0044 | 0.0389 ± 0.0032 | 0.0008 ± 0.0008 | 0.0416 ± 0.0017 | 0.0225 ± 0.0007 | 17.0 ± 0.6 | 0.3 ± 0.6 | 100 ± 0.0 | 100 ± 0.0 | 0.3 ± 0.6 |
| DEFOG | 0.0447 ± 0.0044 | 0.0396 ± 0.0028 | 0.0002 ± 0.0002 | 0.0122 ± 0.0020 | 0.0089 ± 0.0011 | 10.6 ± 0.8 | 39.6 ± 2.8 | 100 ± 0.0 | 100 ± 0.0 | 39.6 ± 2.8 |
| DIRECTO-DD RRWP | 0.0424 ± 0.0043 | 0.0413 ± 0.0037 | 0.0000 ± 0.0000 | 0.0132 ± 0.0053 | 0.0074 ± 0.0000 | 15.3 ± 0.8 | 72.7 ± 3.9 | 100 ± 0.0 | 100 ± 0.0 | 72.7 ± 3.9 |
| DIRECTO-DD MagLap | 0.0245 ± 0.0022 | 0.0359 ± 0.0036 | 0.0000 ± 0.0000 | 0.0163 ± 0.0026 | 0.0086 ± 0.0011 | 7.6 ± 0.7 | 86.5 ± 5.1 | 100 ± 0.0 | 100 ± 0.0 | 61.9 ± 4.4 |
| DIRECTO RRWP | 0.0494 ± 0.0021 | 0.0425 ± 0.0029 | 0.0000 ± 0.0000 | 0.0226 ± 0.0075 | 0.0228 ± 0.0058 | 12.8 ± 0.6 | 86.0 ± 4.5 | 98.4 ± 1.7 | 99.3 ± 0.7 | **83.8** ± 4.3 |
| DIRECTO MagLap | 0.0180 ± 0.0024 | 0.0302 ± 0.0040 | 0.0000 ± 0.0000 | 0.0142 ± 0.0021 | 0.0099 ± 0.0013 | **6.2** ± 0.5 | 67.6 ± 3.6 | 99.7 ± 0.6 | 99.7 ± 0.6 | 67.0 ± 4.3 |

Table 11: Directed graph generation performance on node-label distributional alignment metrics for the two real-world datasets. Results are presented as mean ± standard deviation across five different sampling runs.

| Model | Node type (MMD) ↓ | Precision ↑ | Recall ↑ | FID ↓ | RBF (MMD) ↓ |
|---|---|---|---|---|---|
| | | | TPU Tiles | | |
| Training set | 0.0001 | 0.64 | 0.99 | 0.0854 | 0.0021 |
| MLE | 0.4573 ± 0.0070 | 0.10 ± 0.01 | 0.22 ± 0.01 | 105776.8 ± 22676.3 | 1.0392 ± 0.0326 |
| D-VAE | OOT | OOT | OOT | OOT | OOT |
| LAYERDAG | 0.0130 ± 0.0077 | 0.31 ± 0.03 | 0.24 ± 0.10 | 2864.1 ± 1715.6 | 1.0208 ± 0.0225 |
| DIGRESS | 0.0037 ± 0.0033 | 0.95 ± 0.05 | 0.46 ± 0.05 | 887.7 ± 313.8 | 0.0969 ± 0.0331 |
| DEFOG | 0.0031 ± 0.0015 | 0.90 ± 0.03 | 0.45 ± 0.05 | 159.5 ± 108.9 | 0.0579 ± 0.0150 |
| DIRECTO-DD RRWP | 0.0028 ± 0.0008 | 0.92 ± 0.03 | **0.48** ± 0.02 | 336.6 ± 163.9 | 0.0587 ± 0.0225 |
| DIRECTO-DD MagLap | 0.0051 ± 0.0030 | 0.91 ± 0.03 | 0.42 ± 0.03 | 482.2 ± 124.3 | 0.0792 ± 0.0266 |
| DIRECTO RRWP | **0.0018** ± 0.0004 | **0.97** ± 0.01 | 0.44 ± 0.03 | 293.0 ± 110.1 | 0.0438 ± 0.0179 |
| DIRECTO MagLap | 0.0086 ± 0.0014 | 0.96 ± 0.02 | 0.42 ± 0.02 | **96.0** ± 64.0 | **0.0421** ± 0.0080 |
| | | | Visual Genome | | |
| Training set | 0.0072 | 0.34 | 0.81 | 1.2 | 0.0214 |
| MLE | 0.0807 ± 0.0030 | 0.25 ± 0.03 | 0.13 ± 0.01 | 48941.0 ± 19586.5 | 0.6179 ± 0.0253 |
| DIGRESS | 0.0206 ± 0.0034 | 0.46 ± 0.04 | 0.35 ± 0.04 | 42.6 ± 1.3 | 0.2321 ± 0.0276 |
| DEFOG | **0.0181** ± 0.0030 | 0.51 ± 0.02 | 0.39 ± 0.02 | 19.9 ± 5.6 | 0.0845 ± 0.0233 |
| DIRECTO-DD RRWP | 0.0231 ± 0.0082 | 0.48 ± 0.02 | **0.48** ± 0.02 | 4.8 ± 1.2 | 0.0391 ± 0.0047 |
| DIRECTO-DD MagLap | 0.0288 ± 0.0069 | 0.50 ± 0.02 | 0.46 ± 0.02 | 8.4 ± 2.5 | 0.0420 ± 0.0056 |
| DIRECTO RRWP | 0.0272 ± 0.0056 | 0.53 ± 0.02 | 0.45 ± 0.02 | 4.7 ± 2.1 | **0.0383** ± 0.0049 |
| DIRECTO MagLap | 0.0299 ± 0.0047 | **0.58** ± 0.02 | 0.46 ± 0.01 | 9.3 ± 2.6 | 0.0507 ± 0.0118 |

## H.3 THE ROLE OF DUAL ATTENTION

To assess the impact of the dual-attention mechanism in our model, we conduct an ablation study in which we remove the cross-attention between edge features and their transposes. This mechanism is designed to capture both source-to-target and target-to-source interactions, which are critical for modeling directional dependencies in directed graphs. By disabling dual attention, we isolate its contribution to generation quality, particularly in terms of validity, expressiveness, and generalization. We compare the full model with its ablated variant across model combinations to quantify the importance of this component.

**DIRECTO** Table 16 presents detailed ablation results highlighting the impact of the dual attention mechanism in our method. The results are presented for two datasets: ER-DAG and SBM. In particular, we observe that, across both datasets, the inclusion of the dual attention mechanism consistently improves the validity of the generated graphs and reduces structural discrepancies as measured by clustering, spectral, and wavelet distances. In the ER-DAG setting, models with dual

attention (e.g., RRWP-Dual and MagLap-Dual) significantly outperform their non-dual counterparts in validity, achieving up to 94% validity with RRWP-Dual and 91% with MagLap-Dual, compared to just 67.5% and 74%, respectively. These models also demonstrate better structural fidelity, particularly in clustering and spectral metrics. A similar trend is observed in the SBM dataset, where dual attention boosts validity (e.g., RRWP-Dual: 87%, MagLap-Dual: 77%). These results demonstrate how incorporating bidirectional information flow into the attention mechanism contributes to improved model performance and provides insight into the effectiveness of this architectural component.

Table 12: Directed Graph Generation performance across different transformer architectures using discrete flow matching. MagLap considers $Q = 1$.

| Dataset | Model | Out-degree ↓ | In-degree ↓ | Clustering ↓ | Spectre ↓ | Wavelet ↓ | Ratio ↓ | Valid ↑ | Unique ↑ | Novel ↑ | V.U.N. ↑ |
|---|---|---|---|---|---|---|---|---|---|---|---|
| | No PE | 0.0078 ± 0.0008 | 0.0078 ± 0.0010 | 0.1293 ± 0.0024 | 0.0735 ± 0.0023 | 0.0874 ± 0.0030 | 12.2 ± 0.4 | 0.0 ± 0.0 | 100 ± 0.0 | 100 ± 0.0 | 0.0 ± 0.0 |
| | No PE - Dual | 0.0109 ± 0.0012 | 0.0110 ± 0.0012 | 0.0807 ± 0.0147 | 0.0073 ± 0.0007 | 0.0215 ± 0.0017 | 3.0 ± 0.2 | 47.0 ± 12.1 | 100 ± 0.0 | 100 ± 0.0 | 47.0 ± 12.1 |
| ER-DAG | RRWP | 0.0109 ± 0.0019 | 0.0108 ± 0.0017 | 0.0540 ± 0.0112 | 0.0061 ± 0.0008 | 0.0030 ± 0.0005 | 1.3 ± 0.1 | 67.5 ± 1.6 | 100 ± 0.0 | 100 ± 0.0 | 67.5 ± 1.6 |
| | RRWP - Dual | 0.0107 ± 0.0019 | 0.0104 ± 0.0012 | 0.1209 ± 0.0194 | 0.0054 ± 0.0005 | 0.0043 ± 0.0008 | 1.7 ± 0.1 | 94.0 ± 1.0 | 100 ± 0.0 | 100 ± 0.0 | 94.0 ± 1.0 |
| | MagLap | 0.0113 ± 0.0024 | 0.0110 ± 0.0015 | 0.1196 ± 0.0107 | 0.0061 ± 0.0011 | 0.0029 ± 0.0006 | 1.7 ± 0.2 | 74.0 ± 4.4 | 100 ± 0.0 | 100 ± 0.0 | 74.0 ± 4.4 |
| | MagLap - Dual | 0.0110 ± 0.0017 | 0.0116 ± 0.0021 | 0.0509 ± 0.0017 | 0.0062 ± 0.0014 | 0.0027 ± 0.0005 | 1.3 ± 0.2 | 91.0 ± 2.5 | 100 ± 0.0 | 100 ± 0.0 | 91.0 ± 2.5 |
| | No PE | 0.0035 ± 0.0015 | 0.0031 ± 0.0015 | 0.1782 ± 0.0117 | 0.0044 ± 0.0006 | 0.0655 ± 0.0058 | 20.6 ± 2.0 | 0.0 ± 0.0 | 100 ± 0.0 | 100 ± 0.0 | 0.0 ± 0.0 |
| | No PE - Dual | 0.0038 ± 0.0011 | 0.0037 ± 0.0010 | 0.0638 ± 0.0059 | 0.0037 ± 0.0003 | 0.0127 ± 0.0018 | 3.5 ± 0.4 | 48.5 ± 6.4 | 100 ± 0.0 | 100 ± 0.0 | 48.5 ± 6.4 |
| SBM | RRWP | 0.0026 ± 0.0019 | 0.0022 ± 0.0016 | 0.0813 ± 0.0058 | 0.0048 ± 0.0002 | 0.0110 ± 0.0019 | 4.3 ± 0.8 | 37 ± 6.6 | 100 ± 0.0 | 100 ± 0.0 | 37.0 ± 6.6 |
| | RRWP - Dual | 0.0031 ± 0.0014 | 0.0028 ± 0.0013 | 0.0594 ± 0.0027 | 0.0035 ± 0.0004 | 0.0027 ± 0.0009 | 1.8 ± 0.5 | 99.5 ± 1.0 | 100 ± 0.0 | 100 ± 0.0 | 99.5 ± 1.0 |
| | MagLap | 0.0032 ± 0.0014 | 0.0028 ± 0.0013 | 0.1229 ± 0.0097 | 0.0068 ± 0.0010 | 0.0198 ± 0.0016 | 7.3 ± 0.7 | 31.5 ± 2.0 | 100 ± 0.0 | 100 ± 0.0 | 31.5 ± 2.0 |
| | MagLap - Dual | 0.0039 ± 0.0011 | 0.0036 ± 0.0010 | 0.0653 ± 0.0026 | 0.0033 ± 0.0003 | 0.0038 ± 0.0010 | 1.9 ± 0.3 | 77.0 ± 7.6 | 100 ± 0.0 | 100 ± 0.0 | 77.0 ± 7.6 |

**DIRECTO-DD** Table 13 shows a second ablation assessing the contribution of the dual attention mechanism within the discrete diffusion framework. In this case, we see a similar pattern to that observed under discrete flow matching: the incorporation of dual attention leads to consistent gains in graph validity across both ER-DAG and SBM datasets. Models augmented with dual attention not only produce more valid graphs but also exhibit improved alignment with structural statistics such as clustering, spectre, and wavelet.

Table 13: Directed Graph Generation performance across different transformer architectures using discrete diffusion. MagLap considers $Q = 1$.

| Dataset | Model | Out-degree ↓ | In-degree ↓ | Clustering ↓ | Spectre ↓ | Wavelet ↓ | Ratio ↓ | Valid ↑ | Unique ↑ | Novel ↑ | V.U.N. ↑ |
|---|---|---|---|---|---|---|---|---|---|---|---|
| | No PE | 0.0094 ± 0.0024 | 0.0095 ± 0.0023 | 0.1221 ± 0.0122 | 0.0814 ± 0.0030 | 0.1094 ± 0.0066 | 14.4 ± 0.8 | 0.0 ± 0.0 | 100 ± 0.0 | 100 ± 0.0 | 0.0 ± 0.0 |
| | No PE - Dual | 0.0149 ± 0.0037 | 0.0138 ± 0.0035 | 0.1256 ± 0.0088 | 0.0089 ± 0.0018 | 0.0106 ± 0.0014 | 2.6 ± 0.3 | 53.0 ± 2.9 | 100 ± 0.0 | 100 ± 0.0 | 53.0 ± 2.9 |
| ER-DAG | RRWP | 0.0147 ± 0.0038 | 0.0139 ± 0.0038 | 0.0985 ± 0.0154 | 0.0067 ± 0.0009 | 0.0038 ± 0.0009 | 2.1 ± 0.3 | 42.0 ± 1.0 | 100 ± 0.0 | 100 ± 0.0 | 42.0 ± 1.0 |
| | RRWP - Dual | 0.0142 ± 0.0040 | 0.0129 ± 0.0034 | 0.0408 ± 0.0053 | 0.0055 ± 0.0010 | 0.0048 ± 0.0014 | 1.4 ± 0.3 | 79.0 ± 3.7 | 100 ± 0.0 | 100 ± 0.0 | 79.0 ± 3.7 |
| | MagLap | 0.0144 ± 0.0040 | 0.0135 ± 0.0038 | 0.0792 ± 0.0126 | 0.0071 ± 0.0012 | 0.0036 ± 0.0011 | 1.6 ± 0.3 | 59.0 ± 6.8 | 100 ± 0.0 | 100 ± 0.0 | 59.0 ± 6.8 |
| | MagLap - Dual | 0.0142 ± 0.0044 | 0.0135 ± 0.0034 | 0.0438 ± 0.0071 | 0.0057 ± 0.0013 | 0.0046 ± 0.0010 | 1.4 ± 0.3 | 69.0 ± 6.4 | 100 ± 0.0 | 100 ± 0.0 | 69.0 ± 6.4 |
| | No PE | 0.0038 ± 0.0021 | 0.0040 ± 0.0020 | 0.1912 ± 0.0177 | 0.0120 ± 0.0019 | 0.0975 ± 0.0081 | 20.5 ± 2.0 | 0.0 ± 0.0 | 100 ± 0.0 | 100 ± 0.0 | 0.0 ± 0.0 |
| | No PE - Dual | 0.0037 ± 0.0020 | 0.0039 ± 0.0019 | 0.0721 ± 0.0059 | 0.0055 ± 0.0007 | 0.0403 ± 0.0067 | 8.8 ± 1.3 | 35.0 ± 9.1 | 100 ± 0.0 | 100 ± 0.0 | 35.0 ± 9.1 |
| SBM | RRWP | 0.0038 ± 0.0019 | 0.0038 ± 0.0018 | 0.1085 ± 0.0205 | 0.0088 ± 0.0009 | 0.0359 ± 0.0043 | 8.5 ± 0.3 | 4.0 ± 2.0 | 100 ± 0.0 | 100 ± 0.0 | 4.0 ± 2.0 |
| | RRWP - Dual | 0.0036 ± 0.0019 | 0.0036 ± 0.0018 | 0.0592 ± 0.0027 | 0.0038 ± 0.0007 | 0.0027 ± 0.0009 | 1.7 ± 0.4 | 81.5 ± 3.2 | 100 ± 0.0 | 100 ± 0.0 | 81.5 ± 3.2 |
| | MagLap | 0.0038 ± 0.0019 | 0.0039 ± 0.0019 | 0.0186 ± 0.0046 | 0.0043 ± 0.0006 | 0.0543 ± 0.0072 | 11.3 ± 1.6 | 10.0 ± 4.2 | 100 ± 0.0 | 100 ± 0.0 | 10.0 ± 4.2 |
| | MagLap - Dual | 0.0035 ± 0.0018 | 0.0036 ± 0.0018 | 0.0670 ± 0.0032 | 0.0045 ± 0.0008 | 0.0033 ± 0.0013 | 1.9 ± 0.4 | 66.5 ± 4.1 | 100 ± 0.0 | 100 ± 0.0 | 66.5 ± 4.1 |

## H.4 THE ROLE OF POSITIONAL ENCODINGS

To evaluate the importance of positional encodings in our model, we perform an ablation study comparing different options. Directed graphs lack a canonical node ordering, making positional information crucial for capturing structural context. We test several variants, including no positional encoding, encodings that do not take into accounts directed information, and then different directed positional encodings. This analysis isolates how each encoding contributes to the model's ability to generate valid and semantically meaningful graphs. We report performance across synthetic datasets for both discrete diffusion and flow matching.

**DIRECTO** Table 14 presents an ablation study analyzing the impact of various positional encoding (PE) strategies on directed graph generation performance under the discrete flow matching framework. This evaluation highlights the critical role of positional information in enabling models to capture the asymmetric and hierarchical structure inherent to directed graphs.

Overall, the inclusion of any form of positional encoding significantly improves generation validity compared to the baseline with no PE, as expected. Among the methods tested, directed encodings like RRWP and magnetic Laplacian-based variants demonstrate particularly strong performance. These

encodings consistently enhance graph validity and reduce MMD ratio in both datasets. Notably, the MagLap variant with multiple potentials ($Q = 10$) achieves the best results in terms of validity and structural fidelity, although resulting in a higher computational cost.

Table 14: Directed Graph Generation performance across different positional encodings using discrete flow matching.

| Dataset | Model | Out-degree ↓ | In-degree ↓ | Clustering ↓ | Spectre ↓ | Wavelet ↓ | Ratio ↓ | Valid ↑ | Unique ↑ | Novel ↑ | V.U.N. ↑ |
|---|---|---|---|---|---|---|---|---|---|---|---|
| ER-DAG | No PE | 0.0109 ± 0.0012 | 0.0110 ± 0.0012 | 0.0807 ± 0.0147 | 0.0073 ± 0.0007 | 0.0215 ± 0.0017 | 3.0 ± 0.2 | 47.0 ± 12.1 | 100 ± 0.0 | 100 ± 0.0 | 47.0 ± 12.1 |
| | Lap | 0.0119 ± 0.0020 | 0.0115 ± 0.0025 | 0.1272 ± 0.0224 | 0.0047 ± 0.0009 | 0.0048 ± 0.0005 | 1.8 ± 0.0 | 84.0 ± 4.9 | 100 ± 0.0 | 100 ± 0.0 | 84.0 ± 4.9 |
| | DirLap | 0.0048 ± 0.0010 | 0.0046 ± 0.0012 | 0.0509 ± 0.0053 | 0.0041 ± 0.0008 | 0.0006 ± 0.0002 | 1.6 ± 0.2 | 88.0 ± 4.0 | 100 ± 0.0 | 100 ± 0.0 | 88.0 ± 4.0 |
| | RRWP | 0.0107 ± 0.0019 | 0.0104 ± 0.0012 | 0.1209 ± 0.0194 | 0.0054 ± 0.0005 | 0.0043 ± 0.0008 | 1.7 ± 0.1 | 94.0 ± 1.0 | 100 ± 0.0 | 100 ± 0.0 | 94.0 ± 1.0 |
| | MagLap | 0.0110 ± 0.0017 | 0.0116 ± 0.0021 | 0.0509 ± 0.0017 | 0.0062 ± 0.0014 | 0.0027 ± 0.0005 | 1.3 ± 0.2 | 91.0 ± 2.5 | 100 ± 0.0 | 100 ± 0.0 | 91.0 ± 2.5 |
| | MagLap ($Q = 5$) | 0.0112 ± 0.0015 | 0.0107 ± 0.0022 | 0.0527 ± 0.0059 | 0.0056 ± 0.0009 | 0.0032 ± 0.0006 | 1.3 ± 0.2 | 91.5 ± 2.5 | 100 ± 0.0 | 100 ± 0.0 | 91.5 ± 2.5 |
| | MagLap ($Q = 10$) | 0.0117 ± 0.0014 | 0.0110 ± 0.0015 | 0.0711 ± 0.0120 | 0.0055 ± 0.0016 | 0.0026 ± 0.0004 | 1.3 ± 0.2 | 92.0 ± 3.7 | 100 ± 0.0 | 100 ± 0.0 | 92.0 ± 3.7 |
| SBM | No PE | 0.0038 ± 0.0011 | 0.0037 ± 0.0010 | 0.0638 ± 0.0059 | 0.0037 ± 0.0003 | 0.0127 ± 0.0018 | 3.5 ± 0.4 | 48.5 ± 6.4 | 100 ± 0.0 | 100 ± 0.0 | 48.5 ± 6.4 |
| | Lap | 0.0040 ± 0.0012 | 0.0038 ± 0.0011 | 0.0552 ± 0.0042 | 0.0048 ± 0.000g | 0.0044 ± 0.0008 | 2.1 ± 0.3 | 71.5 ± 4.1 | 100 ± 0.0 | 100 ± 0.0 | 71.5 ± 4.1 |
| | DirLap | 0.0061 ± 0.0048 | 0.0057 ± 0.0046 | 0.0544 ± 0.0041 | 0.0035 ± 0.0005 | 0.0024 ± 0.0015 | 2.2 ± 1.3 | 83.0 ± 3.7 | 100 ± 0.0 | 100 ± 0.0 | 83.0 ± 3.7 |
| | RRWP | 0.0031 ± 0.0018 | 0.0028 ± 0.0013 | 0.0594 ± 0.0027 | 0.0035 ± 0.0004 | 0.0027 ± 0.0009 | 1.8 ± 0.5 | 99.5 ± 1.0 | 100 ± 0.0 | 100 ± 0.0 | 99.5 ± 1.0 |
| | MagLap | 0.0039 ± 0.0011 | 0.0036 ± 0.0010 | 0.0653 ± 0.0026 | 0.0033 ± 0.0003 | 0.0038 ± 0.0010 | 1.9 ± 0.3 | 77.0 ± 7.6 | 100 ± 0.0 | 100 ± 0.0 | 77.0 ± 7.6 |
| | MagLap ($Q = 5$) | 0.0032 ± 0.0013 | 0.0028 ± 0.0012 | 0.0620 ± 0.0020 | 0.0035 ± 0.0005 | 0.0034 ± 0.0012 | 2.1 ± 0.3 | 95.5 ± 2.0 | 100 ± 0.0 | 100 ± 0.0 | 95.5 ± 2.0 |
| | MagLap ($Q = 10$) | 0.0039 ± 0.0012 | 0.0038 ± 0.0010 | 0.0654 ± 0.0052 | 0.0038 ± 0.0003 | 0.0039 ± 0.0008 | 2.0 ± 0.3 | 96.5 ± 2.5 | 100 ± 0.0 | 100 ± 0.0 | 96.5 ± 2.5 |

Interestingly, although classical Laplacian encodings (Lap and DirLap) also play a role in increasing the V.U.N. and overall ratio, they obtain worse results than the variants that explicitly encode position and directionality. This emphasizes that, for directed generation tasks, the choice of positional encoding is not merely a design detail, but a relevant architectural decision that affects model success.

**DIRECTO-DD**  Table 15 presents an ablation study assessing the influence of various positional encodings within the discrete diffusion framework across four synthetic datasets: SBM, ER-DAG, ER, and Price's model.

Table 15: Directed Graph Generation performance across different positional encodings using discrete diffusion.

| Dataset | Model | Out-degree ↓ | In-degree ↓ | Clustering ↓ | Spectre ↓ | Wavelet ↓ | Ratio ↓ | Valid ↑ | Unique ↑ | Novel ↑ | V.U.N. ↑ |
|---|---|---|---|---|---|---|---|---|---|---|---|
| ER-DAG | No PE | 0.0149 ± 0.0037 | 0.0138 ± 0.0035 | 0.1256 ± 0.0088 | 0.0089 ± 0.0018 | 0.0106 ± 0.0014 | 2.6 ± 0.3 | 53.0 ± 2.9 | 100 ± 0.0 | 100 ± 0.0 | 53.0 ± 2.9 |
| | Lap | 0.0138 ± 0.0038 | 0.0144 ± 0.0042 | 0.0462 ± 0.0071 | 0.0055 ± 0.0011 | 0.0052 ± 0.0015 | 1.5 ± 0.3 | 67.0 ± 5.3 | 100 ± 0.0 | 100 ± 0.0 | 67.0 ± 5.3 |
| | RRWP | 0.0142 ± 0.0040 | 0.0129 ± 0.0034 | 0.0408 ± 0.0053 | 0.0055 ± 0.0010 | 0.0048 ± 0.0014 | 1.4 ± 0.3 | 79.0 ± 3.7 | 100 ± 0.0 | 100 ± 0.0 | 79.0 ± 3.7 |
| | MagLap | 0.0142 ± 0.0044 | 0.0135 ± 0.0034 | 0.0438 ± 0.0071 | 0.0057 ± 0.0013 | 0.0046 ± 0.0010 | 1.4 ± 0.3 | 69.0 ± 6.4 | 100 ± 0.0 | 100 ± 0.0 | 69.0 ± 6.4 |
| | MagLap ($Q = 5$) | 0.0135 ± 0.0032 | 0.0140 ± 0.0045 | 0.0596 ± 0.0081 | 0.0061 ± 0.0007 | 0.0037 ± 0.0014 | 1.4 ± 0.2 | 78.5 ± 2.5 | 100 ± 0.0 | 100 ± 0.0 | 78.5 ± 2.5 |
| | MagLap ($Q = 10$) | 0.0145 ± 0.0040 | 0.0134 ± 0.0033 | 0.582 ± 0.0083 | 0.0063 ± 0.0015 | 0.0034 ± 0.0011 | 1.5 ± 0.2 | 85.0 ± 9.2 | 100 ± 0.0 | 100 ± 0.0 | 85.0 ± 9.2 |
| SBM | No PE | 0.0037 ± 0.0020 | 0.0039 ± 0.0019 | 0.0721 ± 0.0059 | 0.0055 ± 0.0007 | 0.0403 ± 0.0067 | 8.8 ± 1.3 | 35.0 ± 9.1 | 100 ± 0.0 | 100 ± 0.0 | 35.0 ± 9.1 |
| | Lap | 0.0037 ± 0.0020 | 0.0038 ± 0.0017 | 0.0531 ± 0.0042 | 0.0038 ± 0.0006 | 0.0106 ± 0.0009 | 1.4 ± 0.4 | 81.0 ± 3.7 | 100 ± 0.0 | 100 ± 0.0 | 81.0 ± 3.7 |
| | RRWP | 0.0036 ± 0.0019 | 0.0036 ± 0.0018 | 0.0592 ± 0.0027 | 0.0038 ± 0.0007 | 0.0027 ± 0.0009 | 1.7 ± 0.4 | 81.5 ± 3.2 | 100 ± 0.0 | 100 ± 0.0 | 81.5 ± 3.2 |
| | MagLap | 0.0035 ± 0.0018 | 0.0036 ± 0.0018 | 0.0670 ± 0.0032 | 0.0045 ± 0.0006 | 0.0033 ± 0.0013 | 1.9 ± 0.4 | 66.5 ± 4.1 | 100 ± 0.0 | 100 ± 0.0 | 66.5 ± 4.1 |
| | MagLap ($Q = 5$) | 0.0039 ± 0.0021 | 0.0038 ± 0.0018 | 0.0504 ± 0.0011 | 0.0047 ± 0.0006 | 0.0065 ± 0.0018 | 2.4 ± 0.4 | 92.0 ± 1.9 | 100 ± 0.0 | 100 ± 0.0 | 92.0 ± 1.9 |
| | MagLap ($Q = 10$) | 0.0037 ± 0.0019 | 0.0038 ± 0.0018 | 0.0450 ± 0.0037 | 0.0038 ± 0.0006 | 0.0021 ± 0.0009 | 1.5 ± 0.4 | 95.5 ± 3.7 | 100 ± 0.0 | 100 ± 0.0 | 95.5 ± 3.7 |
| ER | No PE | 0.0039 ± 0.0022 | 0.0038 ± 0.0023 | 0.0486 ± 0.0039 | 0.0039 ± 0.0009 | 0.0030 ± 0.0019 | 2.8 ± 1.4 | 100 ± 0.0 | 100 ± 0.0 | 100 ± 0.0 | 100 ± 0.0 |
| | Lap | 0.0022 ± 0.0014 | 0.0021 ± 0.0014 | 0.0479 ± 0.0058 | 0.0037 ± 0.0013 | 0.0017 ± 0.0010 | 1.8 ± 0.9 | 99.5 ± 1.0 | 100 ± 0.0 | 100 ± 0.0 | 99.5 ± 1.0 |
| | RRWP | 0.0021 ± 0.0013 | 0.0021 ± 0.0013 | 0.0505 ± 0.0047 | 0.0033 ± 0.0013 | 0.0018 ± 0.0010 | 1.8 ± 0.8 | 99.0 ± 1.2 | 100 ± 0.0 | 100 ± 0.0 | 99.5 ± 1.2 |
| | MagLap | 0.0021 ± 0.0013 | 0.0021 ± 0.0013 | 0.0515 ± 0.0055 | 0.0029 ± 0.0004 | 0.0019 ± 0.0011 | 1.8 ± 0.8 | 99.0 ± 1.2 | 100 ± 0.0 | 100 ± 0.0 | 99.0 ± 1.2 |
| | MagLap ($Q = 5$) | 0.0022 ± 0.0014 | 0.0021 ± 0.0014 | 0.0456 ± 0.0068 | 0.0029 ± 0.0006 | 0.0017 ± 0.0010 | 1.8 ± 0.9 | 100 ± 0.0 | 100 ± 0.0 | 100 ± 0.0 | 100 ± 0.0 |
| Price | No PE | 0.0149 ± 0.0020 | 0.0011 ± 0.0002 | 0.0524 ± 0.0030 | 0.0041 ± 0.0040 | 0.0017 ± 0.0000 | 2.9 ± 1.6 | 92.5 ± 1.6 | 100 ± 0.0 | 100 ± 0.0 | 92.5 ± 1.6 |
| | Lap | 0.0219 ± 0.0030 | 0.0010 ± 0.0002 | 0.0725 ± 0.0016 | 0.0476 ± 0.0034 | 0.0124 ± 0.0020 | 16.2 ± 2.0 | 99.0 ± 1.2 | 100 ± 0.0 | 100 ± 0.0 | 99.0 ± 1.2 |
| | RRWP | 0.0231 ± 0.0034 | 0.0010 ± 0.0002 | 0.0837 ± 0.0022 | 0.0619 ± 0.0048 | 0.0123 ± 0.0027 | 17.0 ± 2.8 | 99.0 ± 1.2 | 100 ± 0.0 | 100 ± 0.0 | 99.0 ± 1.2 |
| | MagLap | 0.0232 ± 0.0013 | 0.0012 ± 0.0001 | 0.0752 ± 0.0028 | 0.0531 ± 0.0059 | 0.0126 ± 0.0019 | 16.8 ± 2.1 | 97.5 ± 2.2 | 100 ± 0.0 | 100 ± 0.0 | 97.5 ± 2.2 |
| | MagLap ($Q = 5$) | 0.0191 ± 0.0031 | 0.0010 ± 0.0001 | 0.0653 ± 0.0118 | 0.0239 ± 0.0023 | 0.0044 ± 0.0016 | 6.8 ± 1.7 | 99.5 ± 1.0 | 100 ± 0.0 | 100 ± 0.0 | 99.5 ± 1.0 |

We observe that that ER and Price graphs, due to their simplicity and highly regular generative processes, pose a relatively easy modeling challenge. As a result, performance is consistently high across all variants, including the baseline with no positional encoding. In contrast, the SBM and ER-DAG datasets present more intricate structural patterns, which amplify the benefits of informed positional encodings. In these cases, clear performance gaps emerge between the baseline, non-directed encodings (e.g., Lap), and directed-aware methods such as RRWP and MagLapPE.

These differences highlight the importance of encoding directionality to faithfully capture the distribution of more structurally diverse directed graphs. For this reason, in the main paper we focus our synthetic evaluations on SBM and ER-DAG, where the modeling challenge is sufficiently rich to reveal the comparative strengths of different modeling strategies.

## H.5  ABLATION OF OTHER MODEL COMPONENTS

To evaluate the importance of other parts of our proposed architecture, we have performed an ablation on the role of FiLM layers to enrich the attention with edge information, the use of the softmax in

the aggregation of the source and target attention maps, and the gated residual connection for the node update. The results in Table support the addition of these model components, since it results in improved model performance.

Table 16: Ablation results for different model components. All trainings use dual attention and RRWP as the positional encoding.

| Dataset | Model | Ratio $\downarrow$ | Valid $\uparrow$ | Unique $\uparrow$ | Novelty $\uparrow$ | V.U.N. $\uparrow$ |
|---|---|---|---|---|---|---|
| ER-DAG | Original | **1.7** $\pm$ **0.1** | 94 $\pm$ 1.0 | 100 $\pm$ 0.0 | 100 $\pm$ 0.0 | **94** $\pm$ **1.0** |
| | No FILM | 6.9 $\pm$ 0.6 | 4 $\pm$ 3.0 | 100 $\pm$ 0.0 | 100 $\pm$ 0.0 | 4 $\pm$ 3.0 |
| | No softmax | 1.7 $\pm$ 0.2 | 74.5 $\pm$ 1.9 | 100 $\pm$ 0.0 | 100 $\pm$ 0.0 | 74.5 $\pm$ 1.9 |
| | No FILM + No softmax | 6.3 $\pm$ 0.2 | 0.0 $\pm$ 0.0 | 100 $\pm$ 0.0 | 100 $\pm$ 0.0 | 0.0 $\pm$ 0.0 |
| | No gated residual connection | 2.5 $\pm$ 0.2 | 66.5 $\pm$ 6.0 | 100 $\pm$ 0.0 | 100 $\pm$ 0.0 | 66.5 $\pm$ 6.0 |
| SBM | Original | **1.8** $\pm$ **0.5** | 99.5 $\pm$ 3.7 | 100 $\pm$ 0.0 | 100 $\pm$ 0.0 | **99.5** $\pm$ **3.7** |
| | No FILM | 15.8 $\pm$ 3.3 | 0.0 $\pm$ 0.0 | 100 $\pm$ 0.0 | 100 $\pm$ 0.0 | 0.0 $\pm$ 0.0 |
| | No softmax | 2.4 $\pm$ 1.4 | 89.0 $\pm$ 3.7 | 100 $\pm$ 0.0 | 100 $\pm$ 0.0 | 89.0 $\pm$ 3.7 |
| | No FILM + No softmax | 16.7 $\pm$ 2.8 | 0.0 $\pm$ 0.0 | 100 $\pm$ 0.0 | 100 $\pm$ 0.0 | 0.0 $\pm$ 0.0 |
| | No gated residual connection | 2.7 $\pm$ 1.4 | 91.5 $\pm$ 1.7 | 100 $\pm$ 0.0 | 100 $\pm$ 0.0 | 91.5 $\pm$ 1.7 |

## H.6 ER VS DAG PERFORMANCE

To evaluate the structural quality of generated synthetic directed acyclic graphs, we defined a composite validity metric that captures two key properties: (i) the percentage of generated graphs that are Directed Acyclic Graphs (DAGs), and (ii) the percentage that follow the target structural distribution (e.g. Erdős-Renyi). The validity score is computed as the proportion of generated graphs that satisfy both criteria. For completeness, we also report each component individually to disentangle the contributions of acyclicity and distributional alignment.

Table 17: ER vs DAG accuracy

| Model | % DAG | % ER | Valid |
|---|---|---|---|
| Training set | 100 | 99.2 | 99.2 |
| MLE | 0.0 $\pm$ 0.0 | 97.0 $\pm$ 2.1 | 0.0 $\pm$ 0.0 |
| D-VAE | 100 $\pm$ 0.0 | 0.0 $\pm$ 0.0 | 0.0 $\pm$ 0.0 |
| LAYERDAG | 100 $\pm$ 0.0 | 21.5 $\pm$ 2.7 | 21.5 $\pm$ 2.7 |
| DIGRESS | 35.0 $\pm$ 4.2 | 98.5 $\pm$ 2.0 | 34.0 $\pm$ 4.1 |
| DEFOG | 21.5 $\pm$ 2.7 | 100 $\pm$ 0.0 | 21.5 $\pm$ 2.7 |
| DIRECTO-DD RRWP | 79.0 $\pm$ 3.7 | 100 $\pm$ 0.0 | 79.0 $\pm$ 3.7 |
| DIRECTO-DD MagLap ($Q = 10$) | 86.0 $\pm$ 9.4 | 99 $\pm$ 1.2 | 85.0 $\pm$ 10.2 |
| DIRECTO RRWP | 94.0 $\pm$ 1.0 | 100 $\pm$ 0.0 | 94.0 $\pm$ 1.0 |
| DIRECTO MagLap ($Q = 10$) | 99.5 $\pm$ 1.0 | 92.5 $\pm$ 2.7 | 92.0 $\pm$ 3.7 |

As shown in Table 17, baselines such as MLE and LAYERDAG exhibit a stark imbalance: while LAYERDAG guarantees acyclicity, both fail to model the target ER distribution, resulting in low overall validity. On the other hand, MLE closely aligns with the distribution but fails to generate acyclic graphs.

In contrast, diffusion- and flow matching–based methods paired with dual attention and relevant directed positional encodings achieve strong performance on both fronts, demonstrating their ability to generate structurally valid DAGs that also align with the underlying data distribution. This highlights the limitations of purely autoregressive or heuristic DAG-specific approaches when used in isolation.

## H.7 SCALABILITY EXPERIMENTS

In this section, we investigate the scalability of our method with respect to three key axes: dataset size, parameter efficiency, and graph size. These experiments are designed to disentangle the contributions of data availability, architectural choices, and input complexity to the overall generative performance, providing a clearer understanding of the trade-offs between accuracy, efficiency, and computational cost in our framework.

**Effect of dataset size** To evaluate the effect on the number of graphs available at training time, we conduct an ablation study using the synthetic ER-DAG dataset in two configurations: the standard

size and a variant with 10× more training, test, and validation samples. This setup allows us to assess whether the method encodings benefit from increased data. We compare the different positional encodings for discrete diffusion using dual attention, with results in Table 18.

Table 18: Effect of the dataset size (using DIRECTO-DD).

| Dataset | Model | Out-degree ↓ | In-degree ↓ | Clustering ↓ | Spectre ↓ | Wavelet ↓ | Ratio ↓ | Valid ↑ | Unique ↑ | Novel ↑ | V.U.N. ↑ |
|---|---|---|---|---|---|---|---|---|---|---|---|
| Standard | Train set | 0.0113 | 0.0103 | 0.0355 | 0.0038 | 0.0024 | 1.0 | 99.2 | 100 | 0.0 | 0.0 |
| | Lap | 0.0138 ± 0.0038 | 0.0144 ± 0.0042 | 0.0462 ± 0.0071 | 0.0055 ± 0.0011 | 0.0052 ± 0.0015 | 1.5 ± 0.3 | 67.0 ± 5.3 | 100 ± 0.0 | 100 ± 0.0 | 67.0 ± 5.3 |
| | RRWP | 0.0142 ± 0.0040 | 0.0129 ± 0.0034 | 0.0408 ± 0.0053 | 0.0055 ± 0.0010 | 0.0048 ± 0.0014 | 1.4 ± 0.3 | 79.0 ± 3.7 | 100 ± 0.0 | 100 ± 0.0 | 79.0 ± 3.7 |
| | MagLap | 0.0142 ± 0.0044 | 0.0135 ± 0.0034 | 0.0438 ± 0.0071 | 0.0057 ± 0.0013 | 0.0046 ± 0.0010 | 1.4 ± 0.3 | 69.0 ± 6.4 | 100 ± 0.0 | 100 ± 0.0 | 69.0 ± 6.4 |
| | MagLap ($Q = 5$) | 0.0135 ± 0.0032 | 0.0140 ± 0.0045 | 0.0596 ± 0.0081 | 0.0061 ± 0.0007 | 0.0037 ± 0.0017 | 1.4 ± 0.2 | 78.5 ± 2.5 | 100 ± 0.0 | 100 ± 0.0 | 78.5 ± 2.5 |
| x 10 | Train set | 0.0002 | 0.0003 | 0.0021 | 0.0004 | 0.0000 | 1.0 | 99.9 | 100 | 0.0 | 0.0 |
| | Lap | 0.00035 ± 0.0007 | 0.0032 ± 0.0008 | 0.0261 ± 0.0025 | 0.0024 ± 0.0003 | 0.0006 ± 0.0001 | 10.5 ± 1.3 | 85.5 ± 2.4 | 100 ± 0.0 | 100 ± 0.0 | 85.5 ± 2.4 |
| | RRWP | 0.0030 ± 0.0008 | 0.0031 ± 0.0006 | 0.0277 ± 0.0075 | 0.0024 ± 0.0004 | 0.0006 ± 0.0002 | 10.1 ± 2.2 | 94.0 ± 4.4 | 100 ± 0.0 | 100 ± 0.0 | 94.0 ± 4.4 |
| | MagLap | 0.0030 ± 0.0006 | 0.0033 ± 0.0005 | 0.0305 ± 0.0051 | 0.0025 ± 0.0005 | 0.0007 ± 0.0003 | 14.5 ± 2.4 | 89.5 ± 3.7 | 100 ± 0.0 | 100 ± 0.0 | 89.5 ± 3.7 |
| | MagLap ($Q = 5$) | 0.0031 ± 0.0008 | 0.0028 ± 0.0006 | 0.0237 ± 0.0028 | 0.0022 ± 0.0006 | 0.0008 ± 0.0001 | 9.6 ± 1.6 | 93.5 ± 2.5 | 100 ± 0.0 | 100 ± 0.0 | 93.5 ± 2.5 |

Overall, we see that increasing the dataset size significantly improves generation performance across all positional encoding variants, confirming that additional training data helps models better capture structural patterns. However, this performance gain comes at the cost of substantially longer training times and greater computational demand (∼12h for the standard dataset vs ∼32h for the bigger version). These trade-offs motivate our focus on architectural innovations such as employing discrete flow matching as more efficient alternatives to scaling purely through data and compute.

**Effect of model size (parameter efficiency)**   We next evaluate parameter efficiency by contrasting our dual-attention mechanism with an alternative approach that scales model capacity by doubling the number of parameters. Dual attention increases the parameter space by introducing two adjacency-based attention heads, each specialized for handling edge directionality. To ablate for model size, we compare against a baseline in which the architecture is widened to match the parameter count, but without dual-attention.

Table 19: Directed Graph Generation performance for the dual attention architecture versus doubling the depth of the network without using the dual attention mechanism.

| Dataset | Model | Out-degree ↓ | In-degree ↓ | Clustering ↓ | Spectre ↓ | Wavelet ↓ | Ratio ↓ | Valid ↑ | Unique ↑ | Novel ↑ | V.U.N. ↑ |
|---|---|---|---|---|---|---|---|---|---|---|---|
| ER-DAG | RRWP - Double | 0.0129 ± 0.0020 | 0.0130 ± 0.0015 | 0.1741 ± 0.0200 | 0.0107 ± 0.0007 | 0.0020 ± 0.0005 | 4.9 ± 0.2 | 72.0 ± 9.8 | 100 ± 0.0 | 100 ± 0.0 | 72.0 ± 9.8 |
| | RRWP - Dual | 0.0107 ± 0.0019 | 0.0104 ± 0.0012 | 0.1209 ± 0.0194 | 0.0054 ± 0.0005 | 0.0043 ± 0.0008 | **1.7 ± 0.1** | 94.0 ± 1.0 | 100 ± 0.0 | 100 ± 0.0 | **94.0 ± 1.0** |
| | MagLap - Double | 0.0131 ± 0.0019 | 0.0118 ± 0.0022 | 0.1237 ± 0.0217 | 0.0095 ± 0.0030 | 0.0020 ± 0.0008 | 4.3 ± 0.7 | 80.0 ± 6.3 | 100 ± 0.0 | 100 ± 0.0 | 80.0 ± 6.3 |
| | MagLap - Dual | 0.0110 ± 0.0017 | 0.0116 ± 0.0021 | 0.0509 ± 0.0017 | 0.0062 ± 0.0014 | 0.0027 ± 0.0005 | **1.3 ± 0.2** | 91.0 ± 2.5 | 100 ± 0.0 | 100 ± 0.0 | **91.0 ± 2.5** |
| SBM | RRWP - Double | 0.0085 ± 0.0025 | 0.0081 ± 0.0026 | 0.1755 ± 0.0135 | 0.0078 ± 0.0009 | 0.0816 ± 0.0075 | 26.3 ± 2.5 | 0.0 ± 0.0 | 100 ± 0.0 | 100 ± 0.0 | 0.0 ± 0.0 |
| | RRWP - Dual | 0.0031 ± 0.0014 | 0.0028 ± 0.0013 | 0.0594 ± 0.0027 | 0.0035 ± 0.0004 | 0.0027 ± 0.0009 | **1.8 ± 0.5** | 99.5 ± 1.0 | 100 ± 0.0 | 100 ± 0.0 | **99.5 ± 1.0** |
| | MagLap - Double | 0.0086 ± 0.0023 | 0.0083 ± 0.0024 | 0.1878 ± 0.0221 | 0.0152 ± 0.0044 | 0.0376 ± 0.0037 | 14.2 ± 1.5 | 8.0 ± 7.5 | 100 ± 0.0 | 100 ± 0.0 | 8.0 ± 7.5 |
| | MagLap - Dual | 0.0039 ± 0.0011 | 0.0036 ± 0.0010 | 0.0653 ± 0.0026 | 0.0033 ± 0.0003 | 0.0038 ± 0.0010 | **1.9 ± 0.3** | 77.0 ± 7.6 | 100 ± 0.0 | 100 ± 0.0 | **77.0 ± 7.6** |

As shown in Table 19, this baseline fails to recover the correct edge distribution, particularly in more challenging regimes such as SBM graphs. In contrast, the dual-attention model achieves substantially higher fidelity, demonstrating that our architecture leverages parameters more effectively by embedding structural inductive bias rather than merely increasing model scale.

**Effect of node size**   We finally investigate how the method scales with increasing graph size by conducting experiments on ER-DAG datasets ranging from 80–150 up to 200–250 nodes (see Table 20 for dataset statistics). Due to the poor scalability of MultMagLap encodings, we restrict this ablation to RRWP and MagLap positional encodings, with the largest graphs (200–250 nodes) evaluated only under RRWP. Training and sampling times, together with generation quality, are reported in Table 21. Importantly, the computational overhead introduced by dual attention remains bounded by a constant factor of two and does not affect the overall asymptotic complexity, which matches that of standard graph transformers.

Table 20: Dataset statistics for the synthetic experiments with larger graph size (nodes and edges).

| Min. nodes | Max. nodes | Avg. nodes | Min. edges | Max. edges | Avg. edges |
|---|---|---|---|---|---|
| 80 | 150 | 117 | 1866 | 6699 | 4170 |
| 150 | 200 | 173 | 6643 | 11851 | 9046 |
| 200 | 250 | 225 | 11982 | 18645 | 15148 |

In terms of positional encodings, MagLap exhibits clear scalability limitations, both in runtime and performance, whereas RRWP features scale considerably better, consistent with prior findings (e.g., Appendix G.4 of DeFoG (Qin et al., 2025a)). Regarding generative performance, DIRECTO demonstrates strong distributional fidelity across node sizes, as reflected in the Ratio and distributional validity scores. However, ensuring strict acyclicity becomes increasingly challenging at scale: a single misplaced edge can violate the constraint entirely, and the risk grows quadratically with the number of nodes. This highlights the inherent difficulty of enforcing global constraints in large directed graphs, motivating future work on more robust inductive biases for acyclicity preservation.

Table 21: Scaling performance of DIRECTO on ER-DAG datasets of increasing node size.

| Model | Out-degree ↓ | In-degree ↓ | Clustering ↓ | Spectre ↓ | Wavelet ↓ | Ratio ↓ | Valid ↑ | Unique ↑ | Novel ↑ | V.U.N. ↑ | Train (h) | Min / sample |
|---|---|---|---|---|---|---|---|---|---|---|---|---|
| 80-150 (RRWP) | $0.0045 \pm 0.0001$ | $0.0044 \pm 0.0008$ | $0.3233 \pm 0.0676$ | $0.0010 \pm 0.0003$ | $0.0003 \pm 0.0001$ | $3.7 \pm 0.5$ | $53.5 \pm 3.0$ | $100 \pm 0.0$ | $100 \pm 0.0$ | $53.5 \pm 3.0$ | 22 | 0.285 |
| 80-150 (MagLap) | $0.0040 \pm 0.0005$ | $0.0044 \pm 0.0005$ | $0.3382 \pm 0.0361$ | $0.0015 \pm 0.0002$ | $0.0007 \pm 0.0002$ | $4.2 \pm 0.3$ | $45.0 \pm 7.1$ | $100 \pm 0.0$ | $100 \pm 0.0$ | $45.0 \pm 7.1$ | 26 | 0.940 |
| 150-200 (RRWP) | $0.0034 \pm 0.0002$ | $0.0031 \pm 0.0002$ | $0.1570 \pm 0.0496$ | $0.003 \pm 0.0000$ | $0.0000 \pm 0.0000$ | $1.7 \pm 0.3$ | $60.0 \pm 5.9$ | $100 \pm 0.0$ | $100 \pm 0.0$ | $60.0 \pm 5.9$ | 28 | 0.555 |
| 150-200 (MagLap) | $0.0031 \pm 0.0005$ | $0.0031 \pm 0.0004$ | $0.6691 \pm 0.0558$ | $0.0004 \pm 0.0000$ | $0.0001 \pm 0.0001$ | $3.2 \pm 0.4$ | $34.5 \pm 1.9$ | $100 \pm 0.0$ | $100 \pm 0.0$ | $34.5 \pm 1.9$ | 38 | 1.365 |
| 200-250 (RRWP) | $0.0034 \pm 0.0002$ | $0.0033 \pm 0.0003$ | $0.0411 \pm 0.0178$ | $0.0003 \pm 0.0000$ | $0.0000 \pm 0.0000$ | $1.9 \pm 0.3$ | $2.0 \pm 2.4$ | $100 \pm 0.0$ | $100 \pm 0.0$ | $2.0 \pm 2.4$ | 40 | 0.645 |

## H.8 CONDITIONAL GENERATION

Table 22 shows the results of the experiments with classifier-free guidance conditional generation. We followed the setup of Li et al. (2025), training on the TPU Tiles dataset with the provided conditional information and evaluating with their available suite and surrogate models.

Table 22: Averaged conditional generation results across 3 runs. Baseline results were obtained from Li et al. (2025), where Spearman correlation was not reported.

| Model | Pearson | Spearman | MAE |
|---|---|---|---|
| Real graphs | $0.75 \pm 0.01$ | $0.83 \pm 0.01$ | $0.96 \pm 0.00$ |
| D-VAE | $0.50 \pm 0.01$ | N/A | $1.4 \pm 0.00$ |
| GRAPHRNN | $0.62 \pm 0.02$ | N/A | $1.3 \pm 0.00$ |
| GRAPHPNAS | $0.24 \pm 0.10$ | N/A | $2.1 \pm 0.06$ |
| ONESHOTDAG | $0.56 \pm 0.02$ | N/A | $1.4 \pm 0.01$ |
| LAYERDAG (T=1) | $0.37 \pm 0.11$ | N/A | $2.0 \pm 0.04$ |
| LAYERDAG | $0.65 \pm 0.01$ | N/A | $1.2 \pm 0.01$ |
| DIRECTO | $0.59 \pm 0.01$ | $0.63 \pm 0.03$ | $1.3 \pm 0.01$ |

Unlike methods explicitly designed for conditional DAG generation, our framework is general and primarily optimized for unconditional digraph generation. In contrast, autoregressive models such as GraphRNN and LayerDAG (especially the latter, where conditional generation is the default training setup) are naturally more suited for this task and thus represent strong baselines. Moreover, our evaluation follows their official suite, which is tailored to their own method and may not fully align with our formulation.

Despite this, DIRECTO achieves competitive performance. In Pearson correlation, DIRECTO ranks just behind LayerDAG and GraphRNN, while performing on par with them in mean absolute error (MAE). Notably, it also outperforms LAYERDAG's one-shot variant ONESHOTDAG. We further report Spearman correlation, which is implemented in the evaluation suite but was not included in prior work. These results show that even though conditional generation is not our main focus, DIRECTO achieves a strong balance across all metrics, demonstrating that our framework can readily extend to conditional tasks without specialized architectural modifications.

Finally, our results were obtained using classifier-free guidance, a straightforward extension that may not be optimal for conditional training in our setting. We expect that more dedicated strategies for conditioning could further improve performance. Overall, these findings highlight the versatility of DIRECTO: while unconditional generation remains our primary focus, its ability to adapt effectively to conditional generation underscores the promise of diffusion-based formulations for directed graph generation.

## H.9 IMPACT OF SAMPLING OPTIMIZATION USING DISCRETE FLOW MATCHING

To better understand the influence of the three sampling hyperparameters discussed in Appendix C: time distortion, stochasticity coefficient ($\eta$), and target guidance factor ($\omega$) on generative performance, we analyze their effect on the V.U.N. and the structural ratio. Specifically, we present performance curves for the four primary datasets reported in the main results table: SBM, RRWP, TPU Tiles, and Visual Genome. For each dataset, we evaluate the impact of these parameters under two positional encoding strategies: RRWP and MagLap. We report the results for 100 sampling steps and 5 different sampling runs (except for TPU Tiles where only 1 run was performed due to computational constraints).

The results for the synthetic datasets in Figure 4 illustrate that the sampling hyperparameters influence the generative performance. We observe that different values can lead to variations in both V.U.N. and the structural ratio. However, a higher V.U.N. does not necessarily correspond to a lower ratio. While certain configurations tend to perform well across both RRWP and MagLap positional encodings, the improvements are generally modest, suggesting that optimal settings may vary slightly depending on the dataset and positional encoding strategy used.

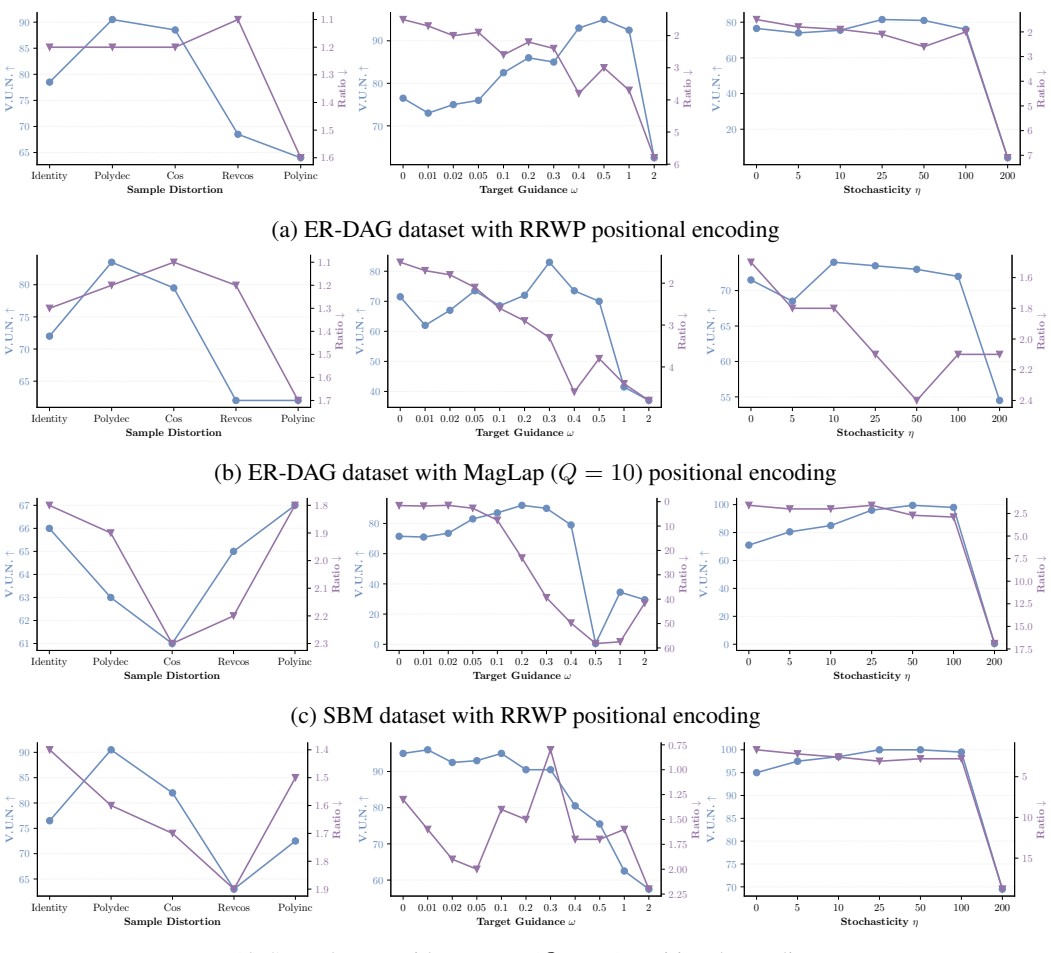

(a) ER-DAG dataset with RRWP positional encoding

(b) ER-DAG dataset with MagLap ($Q = 10$) positional encoding

(c) SBM dataset with RRWP positional encoding

(d) SBM dataset with MagLap ($Q = 10$) positional encoding

Figure 4: Sampling optimization curves for the synthetic datasets with 100 sampling steps and 5 sampling runs. We represent V.U.N. (blue) and MMD ratio (purple) and optimize for best trade-off for each of the three parameters individually.

A similar pattern is observed in the results for the real-world datasets (TPU Tiles and Visual Genome), as shown in Figure 5. The sampling hyperparameters continue to affect generation performance, and there is no clear one-to-one relationship between V.U.N. and ratio. As with the synthetic

datasets, some hyperparameter combinations show slightly more consistent behavior across positional encodings, but overall, the best hyperparameters combination remain dependent on the dataset and positional encoding used.

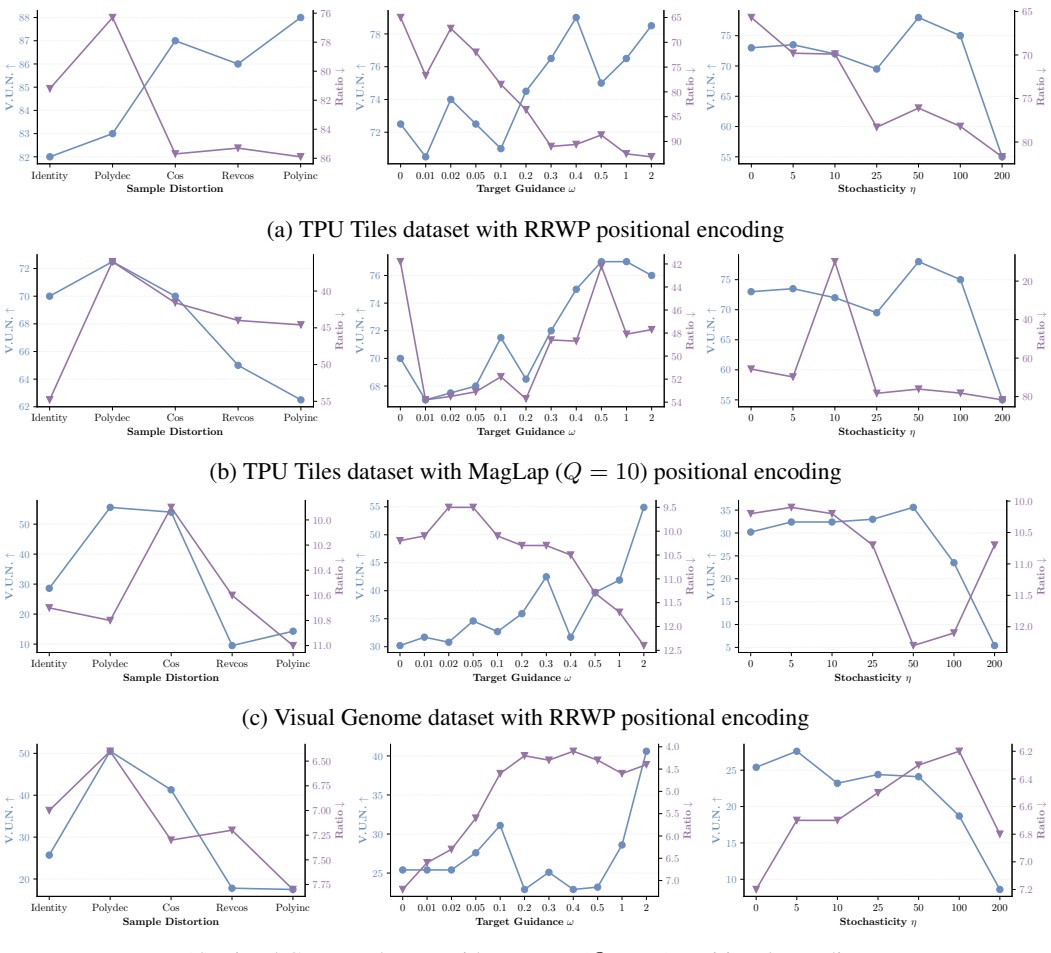

Figure 5: Sampling optimization curves for the synthetic datasets with 100 sampling steps and 5 sampling runs. We represent V.U.N. (blue) and MMD ratio (purple) and optimize for best trade-off for each of the three parameters individually.

# I ITERATIVE REFINEMENT METHODS FOR GRAPH GENERATION

## I.1 GRAPH ITERATIVE REFINEMENT METHODS

Within the realm of undirected graph generation, discrete state-space iterative refinement approaches (Vignac et al., 2023a; Xu et al., 2024; Siraudin et al., 2024; Qin et al., 2025a) have emerged as a powerful framework for capturing the intricate dependencies that govern target graph distributions. These methods achieve state-of-the-art performance by naturally aligning with the structure of graphs: they operate directly in discrete state spaces, matching the inherent discreteness of adjacency matrices, and respect node permutation equivariances due to their one-shot prediction formulation. These characteristics make iterative refinement approaches particularly well-suited for the task of directed graph generation.

These methods comprise two main processes: a noising process and a denoising process. For consistency, we define $t = 0$ as corresponding to fully noised graphs and $t = 1$ to clean graphs. The

noising process runs from $t = 1$ to $t = 0$, progressively corrupting the original graphs, while the denoising process learns how to reverse this trajectory, running from $t = 0$ to $t = 1$.

Denoising is performed using a neural network parametrized by $\theta$,

$$\boldsymbol{p}_{1|t}^{\theta}(\cdot|G_t) = \left( \left( p_{1|t}^{\theta,(n)}(x_1^{(n)} \mid G_t) \right)_{1 \leq n \leq N}, \left( p_{1|t}^{\theta,(i,j)}(e_1^{(i,j)} \mid G_t) \right)_{1 \leq i \neq j \leq N} \right), \quad (39)$$

which predicts categorical distributions over nodes and edges given a noised graph $G_t$. In practice, $\mathbf{p}_{1|t}^{\theta}(\cdot|G_t)$ is parameterized using a graph transformer such as the one from Dwivedi & Bresson (2021), which allows for expressive modeling of complex graph structures. The network is trained using a cross-entropy loss applied independently to each node and each edge:

$$\mathcal{L} = \mathbb{E}_{t,G_1,G_t} \left[ -\sum_n \log \left( p_{1|t}^{\theta,(n)} \left( x_1^{(n)} \mid G_t \right) \right) - \lambda \sum_{i \neq j} \log \left( p_{1|t}^{\theta,(i,j)} \left( e_1^{(i,j)} \mid G_t \right) \right) \right], \quad (40)$$

where the expectation is taken over time $t$ sampled from a predefined distribution over $[0, 1]$ (e.g., uniform); $G_1 \sim p_1(G_1)$ is a clean graph from the dataset; and $G_t \sim p_t(G_t|G_1)$ is its noised version at time $t$. The hyperparameter $\lambda \in \mathbb{R}^+$ controls the relative weighting between node and edge reconstruction losses. Once the denoising network is trained, different graph iterative refinement methods vary in how they leverage the predicted $\mathbf{p}_{1|t}^{\theta}(\cdot|G_t)$ to progressively recover the clean graphs.

## I.2 DISCRETE FLOW MATCHING FOR GRAPH GENERATION

Among graph iterative refinement methods, Discrete Flow Matching (DFM) (Campbell et al., 2024; Gat et al., 2024) has recently emerged as a particularly powerful framework. By decoupling the training and sampling processes, DFM enables a broader and more flexible design space, which has been shown to improve generative performance. This framework has proven particularly effective for undirected graph generation (Qin et al., 2025a).

The noising process in DFM is defined as a linear interpolation between the data distribution and a pre-specified noise distribution. This interpolation is applied independently to each variable, corresponding to each node and each edge in the case of graphs:

$$p_{t|1}^X(x_t^{(n)} \mid x_1^{(n)}) = t\,\delta(x_t^{(n)}, x_1^{(n)}) + (1 - t)\,p_{\text{noise}}^X(x_t^{(n)}), \quad (41)$$

where $\delta(\cdot, \cdot)$ is the Kronecker delta and $p_{\text{noise}}^X$ is a reference distribution over node categories. A similar construction is used for edges. Therefore, the complete noising process corresponds to independently noising each node and edge through $p_{t|1}^X$ and $p_{t|1}^E$, respectively.

To reverse this process, DFM models denoising as a Continuous-Time Markov Chain (CTMC). Starting from an initial distribution $\boldsymbol{p}_0$, the generative process evolves according to:

$$\boldsymbol{p}_{t+\Delta t|t}(G_{t+\Delta t} \mid G_t) = \boldsymbol{\delta}(G_t, G_{t+\Delta t}) + \boldsymbol{R}_t(G_t, G_{t+\Delta t})\,dt, \quad (42)$$

where $\boldsymbol{R}_t$ is the CTMC rate matrix. In practice, this update is approximated over a finite interval $\Delta t$, in an Euler method step, with the rate matrix estimated from the network predictions $\mathbf{p}_{1|t}^{\theta}(\cdot \mid G_t)$ via:

$$\boldsymbol{R}_t^{\theta}(G_t, G_{t+\Delta t}) = \sum_n \delta(G_t^{\backslash(n)}, G_{t+\Delta t}^{\backslash(n)})\, \mathbb{E}_{p_{1|t}^{\theta,(n)}(x_1^{(n)}|G_t)} \left[ R_t^{(n)}(x_t^{(n)}, x_{t+\Delta t}^{(n)} \mid x_1^{(n)}) \right] \quad (43)$$

$$+ \sum_{i \neq j} \delta(G_t^{\backslash(i,j)}, G_{t+\Delta t}^{\backslash(i,j)})\, \mathbb{E}_{p_{1|t}^{\theta,(i,j)}(e_1^{(i,j)}|G_t)} \left[ R_t^{(i,j)}(e_t^{(i,j)}, e_{t+\Delta t}^{(i,j)} \mid e_1^{(i,j)}) \right], \quad (44)$$

where the $G^{\backslash(d)}$ denote the full graph $G$ except variable $d$ (node or edge). The Knonecker deltas ensure that each variable-specific rate matrix, $R_t^{(n)}$ or $R_t^{(i,j)}$, is applied independently at each node and edge, respectively. Finally, each node-specific rate matrix is defined as:

$$R_t(x_t^{(n)}, x_{t+\Delta t}^{(n)} \mid x_1^{(n)}) = \frac{\text{ReLU} \left[ \partial_t p_{t|1}(x_{t+\Delta t}^{(n)} \mid x_1^{(n)}) - \partial_t p_{t|1}(x_t^{(n)} \mid x_1^{(n)}) \right]}{X_t^{>0}\, p_{t|1}(x_t^{(n)} \mid x_1^{(n)})}, \quad (45)$$

for each node $x^{(n)}$, where $X_t^{>0} = \left| \left\{ x_t^{(n)} : p_{t|1}(x_t^{(n)} \mid x_1^{(n)}) > 0 \right\} \right|$. An analogous definition applies for edge rate matrices.

### I.3 DISCRETE DIFFUSION FOR GRAPH GENERATION

Among discrete diffusion-based models for undirected graphs, DIGRESS (Vignac et al., 2023a), grounded in the structured discrete diffusion framework (Austin et al., 2021a), has been particularly influential. This approach mostly differs from the DFM-based formulation described in Appendix I.2, as it operates in discrete time, with the denoising process modeled as a Discrete-Time Markov Chain (DTMC), in contrast to the continuous-time formulation used in DFM.

The forward, or noising, process is modeled as a Markov noise process $q$, which generates a sequence of progressively noised graphs $G_t$, for $t = 1, \ldots, T$. At each timestep, node and edge tensors are perturbed using categorical transition matrices $[\mathbf{Q}_t^X]^{(i,j)} = q(x_t = j \mid x_{t-1} = i)$ and $[\mathbf{Q}_t^E]^{(i,j)} = q(e_t = j \mid e_{t-1} = i)$, respectively. This process induces structural changes such as edge addition or deletion, and edits to node and edge categories. The transition probabilities can be summarized as:

$$q(G_t \mid G_{t-1}) = \left(\mathbf{X}_{t-1}\mathbf{Q}_t^X, \mathbf{E}_{t-1}\mathbf{Q}_t^E\right) \quad \text{and} \quad q(G_t \mid G) = \left(\mathbf{X}\prod_{i=1}^t \mathbf{Q}_i^X, \mathbf{E}\prod_{i=1}^t \mathbf{Q}_i^E\right). \quad (46)$$

In practice, these noise matrices are implemented based on the *marginal* noise model, defined as:

$$\mathbf{Q}_t^X = \alpha^t\mathbf{I} + (1 - \alpha^t)\mathbf{1}_X\mathbf{m}_X^\top \quad \text{and} \quad \mathbf{Q}_t^E = \alpha^t\mathbf{I} + (1 - \alpha^t)\mathbf{1}_E\mathbf{m}_E^\top,$$

where $\alpha^t$ transitions from 1 to 0 with $t$ according to the popular cosine scheduling (Nichol & Dhariwal, 2021). The vectors $\mathbf{1}_X \in \{1\}^X$ and $\mathbf{1}_E \in \{1\}^{E+1}$ are filled with ones, and $\mathbf{m}_X \in \Delta^X$ and $\mathbf{m}_E \in \Delta^{E+1}$ are vectors filled with the marginal node and edge distributions, respectively[2].

This noising process can be rewritten using the analogous notation to the one used for DFM in Equation (47) as:

$$p_{t'|1}^X(x_{t'}^{(n)} \mid x_1^{(n)}) = \bar{\alpha}(t')\,\delta(x_{t'}^{(n)}, x_1^{(n)}) + (1 - \bar{\alpha}(t'))\,p_{\text{noise}}^X(x_{t'}^{(n)}), \quad (47)$$

with $t' = 1 - \frac{t}{T}$ and $\bar{\alpha}(t') = \cos\left(\frac{\pi}{2}\frac{1-t'+s}{1+s}\right)^2$, with a small $s$. Importantly, this function is only evaluated in the discrete values of time considered for the DTMC associated to the diffusion model. For the remaining of this section, we consider the transformed variable $t'$ as the reference time variable.

In the reverse, or denoising, process, a clean graph is progressively built leveraging the denoising neural network predictions $p_{1|t'}^\theta(\cdot|G_t')$, analogously to the DFM setting. In particular, the model begins from a noise sample $G_0$ and iteratively predicts the clean graph $G_1$ by modeling node and edge distributions conditioned on the full graph structure. The reverse transition is defined as:

$$p_{t'+\Delta t'|t'}^\theta(G_{t'+\Delta t'} \mid G_{t'}) = \prod_{i=1}^N p_{t'+\Delta t'|t'}^\theta(x_{t'+\Delta t'}^{(n)} \mid G_{t'}) \prod_{1 \le i < j \le N} p_{t'+\Delta t'|t'}^\theta(e_{t'+\Delta t'}^{(i,j)} \mid G_{t'}). \quad (48)$$

with each of the node and edge denoising terms are computer through the following marginalization:

$$\boldsymbol{p}_{t'+\Delta t'|t'}^\theta(x_{t'+\Delta t'}^{(n)} \mid G_{t'}) = \sum_{x_1^{(n)}\in\{1,\ldots,X\}} \boldsymbol{p}_{t'+\Delta t'|1,t'}(x_{t'+\Delta t'}^{(n)}|x_1^{(n)}, G_{t'})\, p_{1|t'}^{\theta,(n)}(x_1^{(n)}|G_{t'}), \quad (49)$$

and similarly for edges. To compute the missing posterior term in Equation (49), $p_{t'+\Delta t'|1,t'}(x_{t'+\Delta t'}^{(n)}|x_1^{(n)}, G_{t'})$, we equate it to the posterior term of the forward process:

$$\boldsymbol{p}_{t'+\Delta t'|1,t'}(x_{t'+\Delta t'}^{(n)}|x_1^{(n)}, G_{t'}) = \begin{cases} \dfrac{\mathbf{x}_{t'}^{(n)}(\mathbf{Q}_{t'}^X)^\top \odot \mathbf{x}_1^{(n)}\bar{\mathbf{Q}}_{t'+\Delta t'}^X}{\mathbf{x}_{t'}^{(n)}\bar{\mathbf{Q}}_{t'}^X\mathbf{x}_1^{(n)}} & \text{if } q(x_{t'}^{(n)}|x_1^{(n)}) > 0, \\ 0 & \text{otherwise,} \end{cases} \quad (50)$$

where $\mathbf{x}_{t'}^{(n)}$ and $\mathbf{x}_1^{(n)}$ denote the vectorized versions of $x_{t'}^{(n)}$ and $x_1^{(n)}$, respectively. Crucially, and contrarily to DFM, the sampling strategy with discrete diffusion is fixed at training time, which restricts the design space of this generative framework.

---

[2]We denote the probability simplex of the state-space of cardinality $Z$ by $\Delta^Z$

## J    VISUALIZATIONS

In this section, we present visualizations of the digraphs generated with DIRECTO for the different synthetic and real-world datasets and combinations of positional encodings reported in the main paper. To allow for a fair comparison between original and generated digraphs, we additionally report digraphs from the train splits of the original datasets.

Original digraphs for the SBM dataset are shown in Figure 6. Figure 7 and Figure 8 show examples of digraphs from the Stochastic Block Model (SBM), where clear community structures are visible, consistent with the underlying block partitioning. These visualizations help illustrate how the positional encodings preserve modularity and inter-cluster connectivity. Similarly, Figure 9 shows original digraphs from the ER-DAG dataset, with Figure 10 and Figure 11 displaying generated digraphs from the Erdős–Rényi (ER-DAG) model. In this case, digraphs are plotted with nodes in topological order, to highlight the acyclicity.

For the real-world datasets, in particular for the TPU Tiles dataset, which consists of directed acyclic graphs (DAGs) representing hardware execution plans, we can see the original DAGs in Figure 12 From the visualizations in Figure 13 and Figure 14 we see that the model is able to capture and generalize across structural patterns that emerge in computational workloads. In a similar way as before, digraphs are represented with nodes in topological order to highlight acyclicity.

Finally, for the Visual Genome dataset, we represent the three different node types: objects (blue), relationships (red) and attributes (green), alongside their respective node label. Figure 15 shows the original graphs, while Figure 16 and Figure 17 report the generations. They show how the model effectively captures the semantic and relational structure across these node categories, learning meaningful patterns that align with human understanding (i.e. *a person wears a yellow shirt*, or *a white cloud in the sky*).

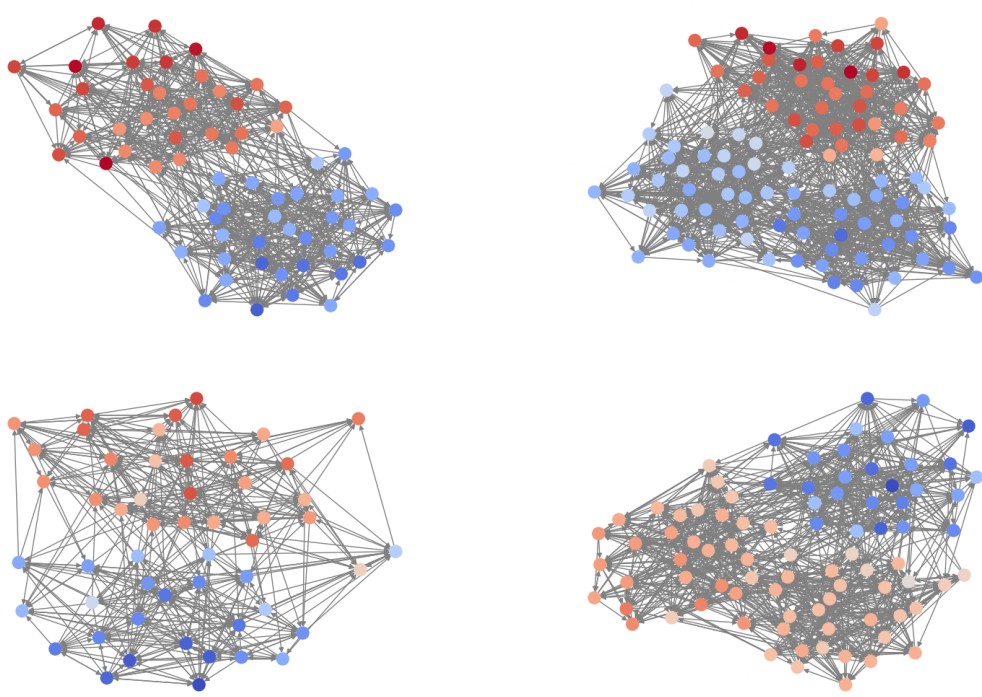

Figure 6: Visualizations of original synthetic digraphs from the train splits of the SBM dataset.

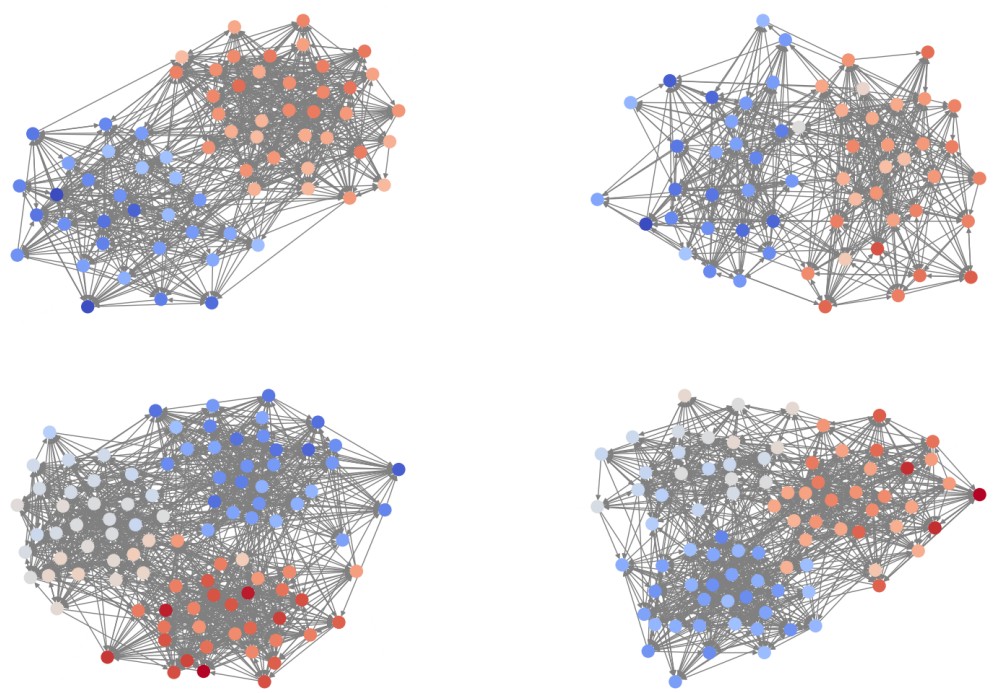

Figure 7: Visualizations of four generated digraphs for the SBM dataset with RRWP positional encoding.

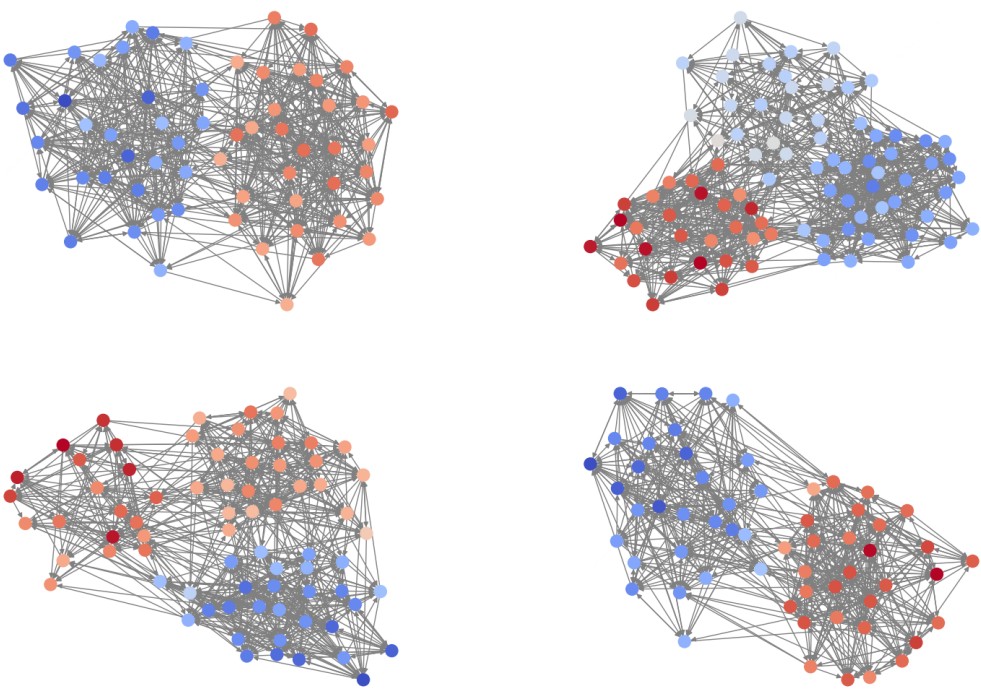

Figure 8: Visualizations of four generated digraphs for the SBM dataset with MagLap ($Q = 10$) positional encoding.

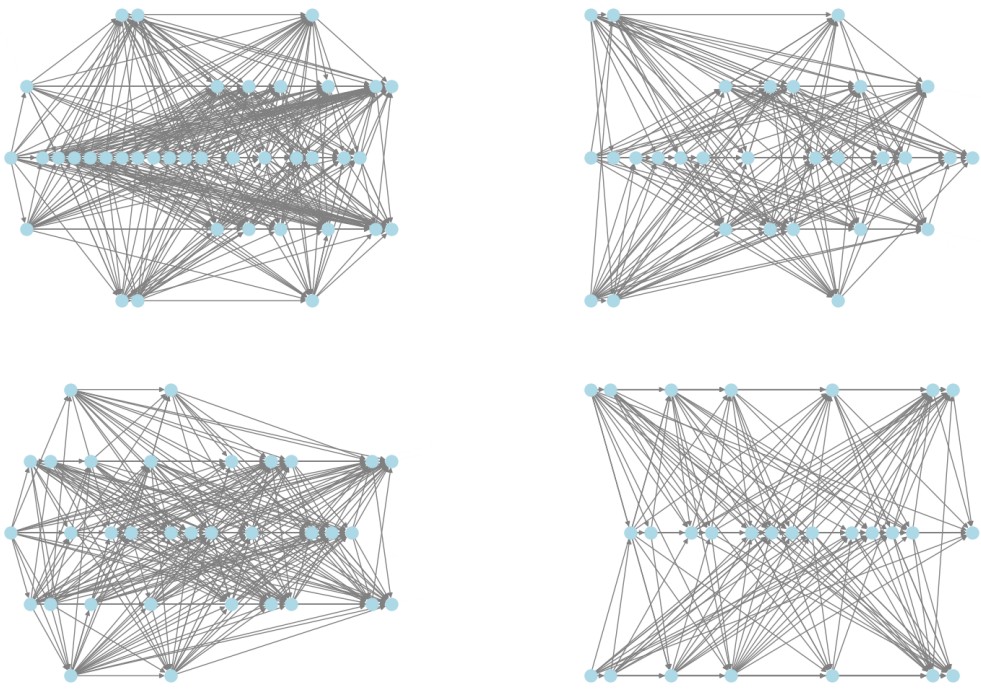

Figure 9: Visualizations of original synthetic digraphs from the train splits of the ER-DAG datasets. Nodes are in topological ordering to highlight the acyclic structure.

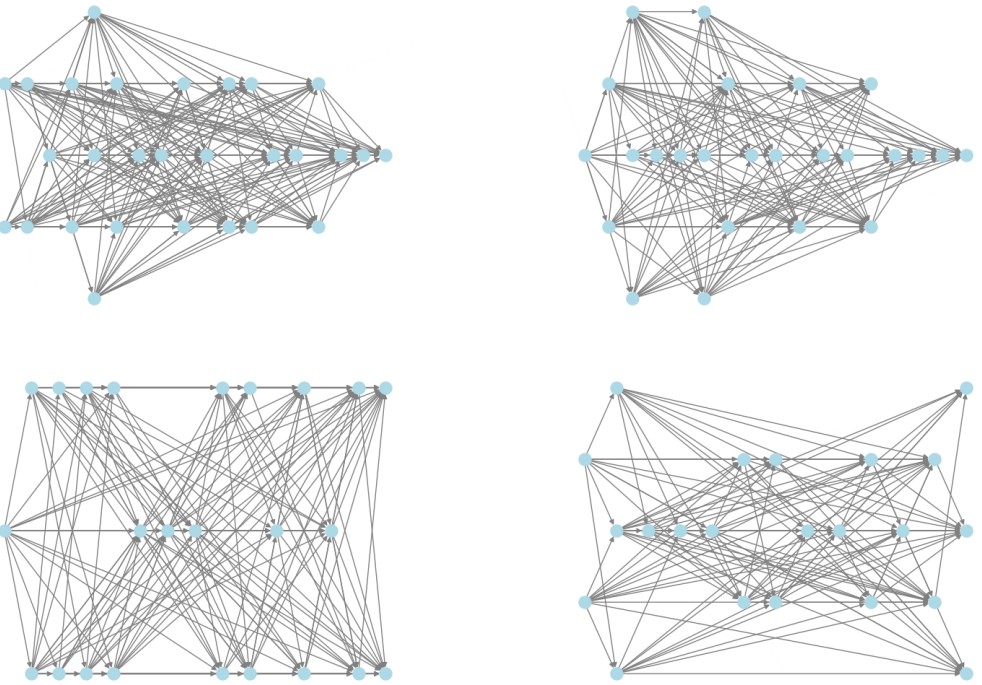

Figure 10: Visualizations of four generated digraphs for the ER-DAG dataset with RRWP positional encoding. Nodes are in topological ordering to highlight the acyclic structure.

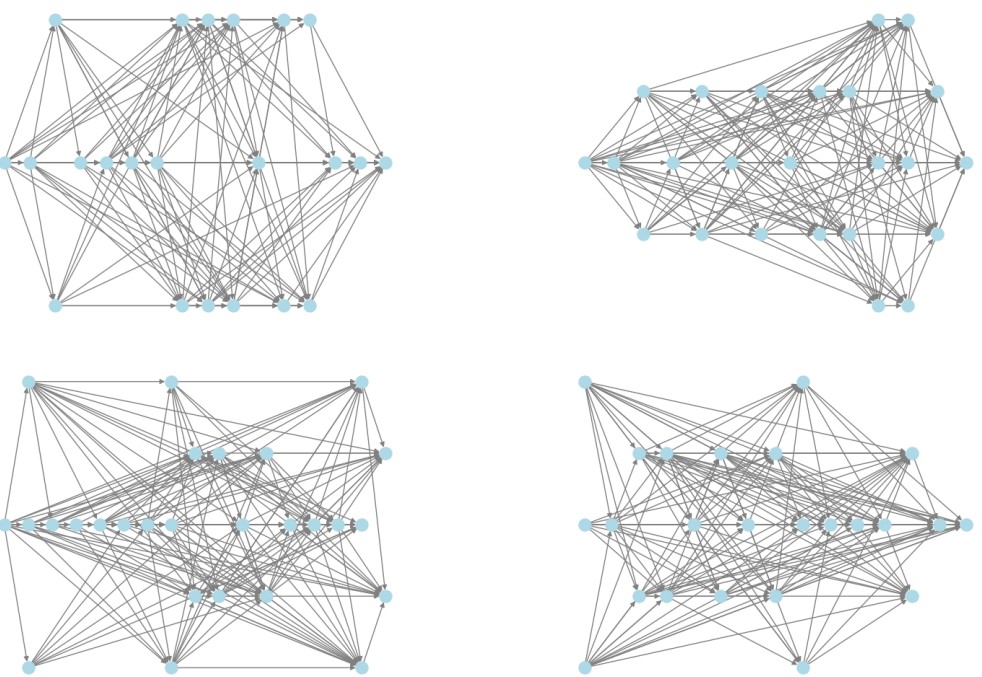

Figure 11: Visualizations of four generated digraphs for the ER-DAG dataset with MagLap ($Q = 10$) positional encoding. Nodes are in topological ordering to highlight the acyclic structure.

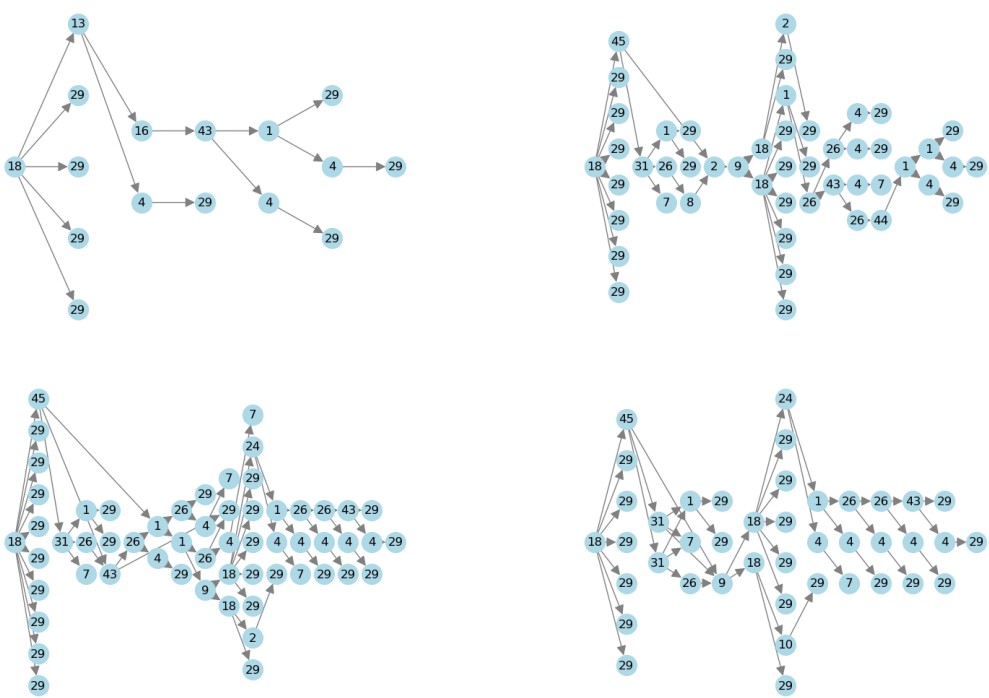

Figure 12: Visualizations of original real-world digraphs from the train splits of the TPU Tiles dataset. Nodes are in topological ordering to highlight the acyclic structure.

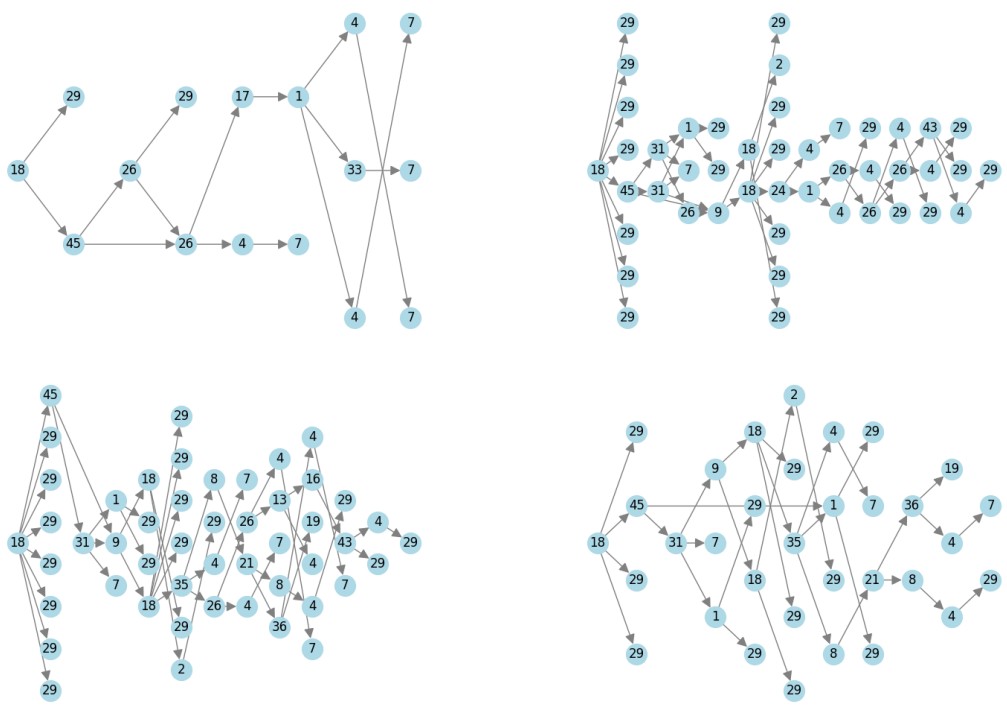

Figure 13: Visualizations of four generated dgraphs for the TPU Tiles dataset with RRWP positional encoding. Nodes are in topological ordering to highlight the acyclic structure.

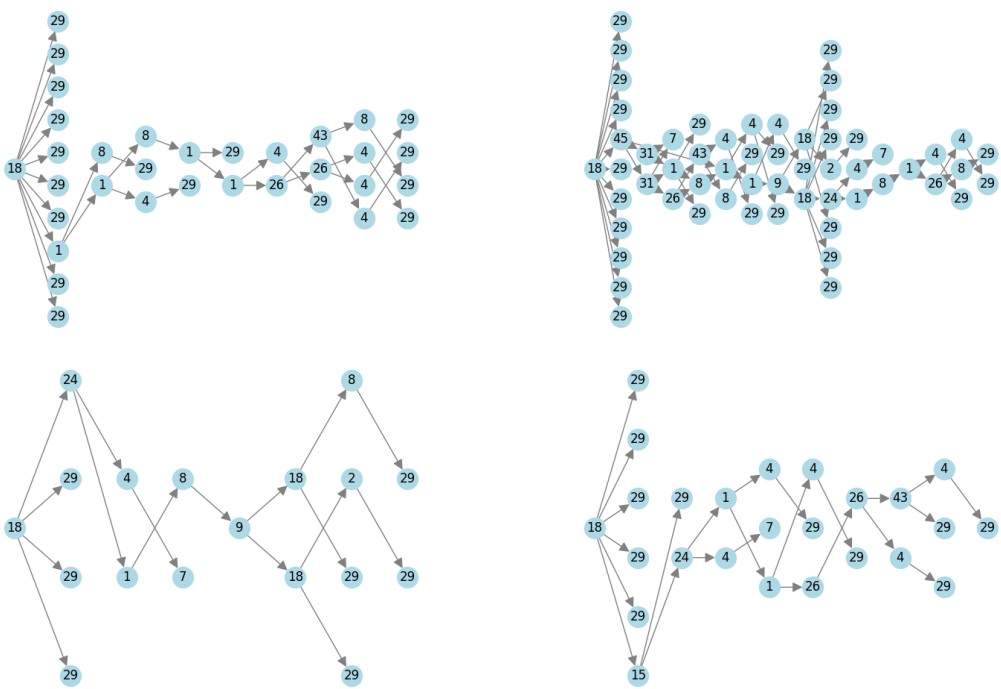

Figure 14: Visualizations of four generated digraphs for the TPU Tiles dataset with MagLap ($Q = 5$) positional encoding. Nodes are in topological ordering to highlight the acyclic structure.

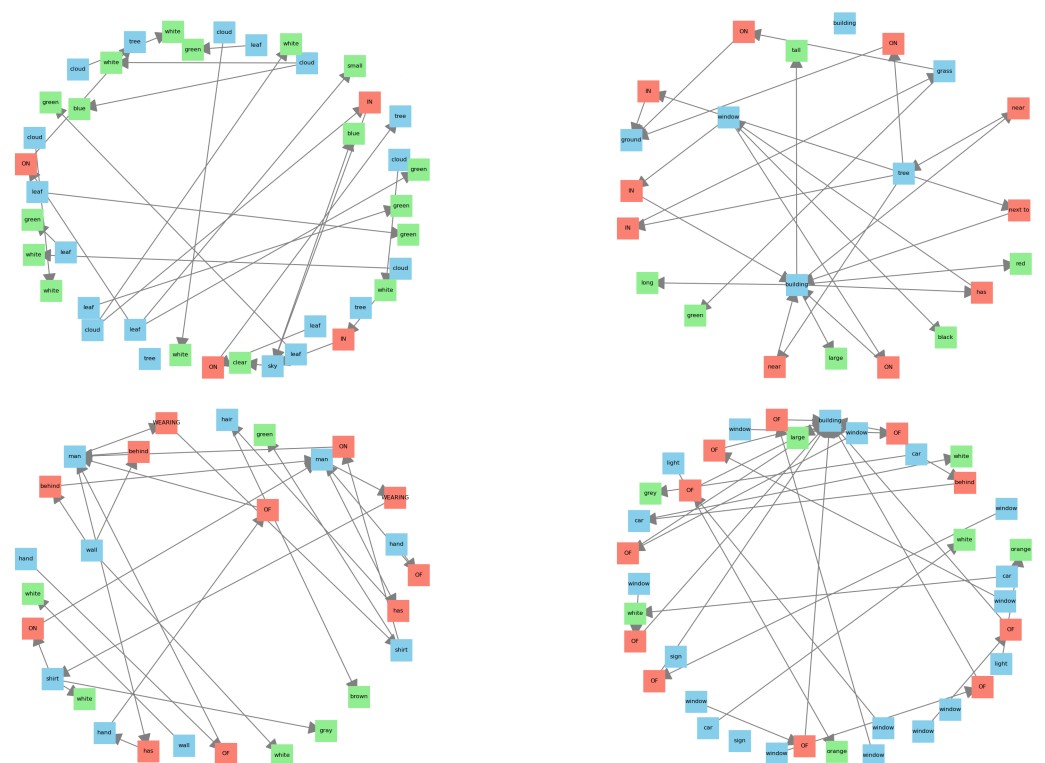

Figure 15: Visualizations of original real-world digraphs from the train splits of the Visual Genome dataset. Nodes represent objects (blue), relationships (red), and attributes (green).

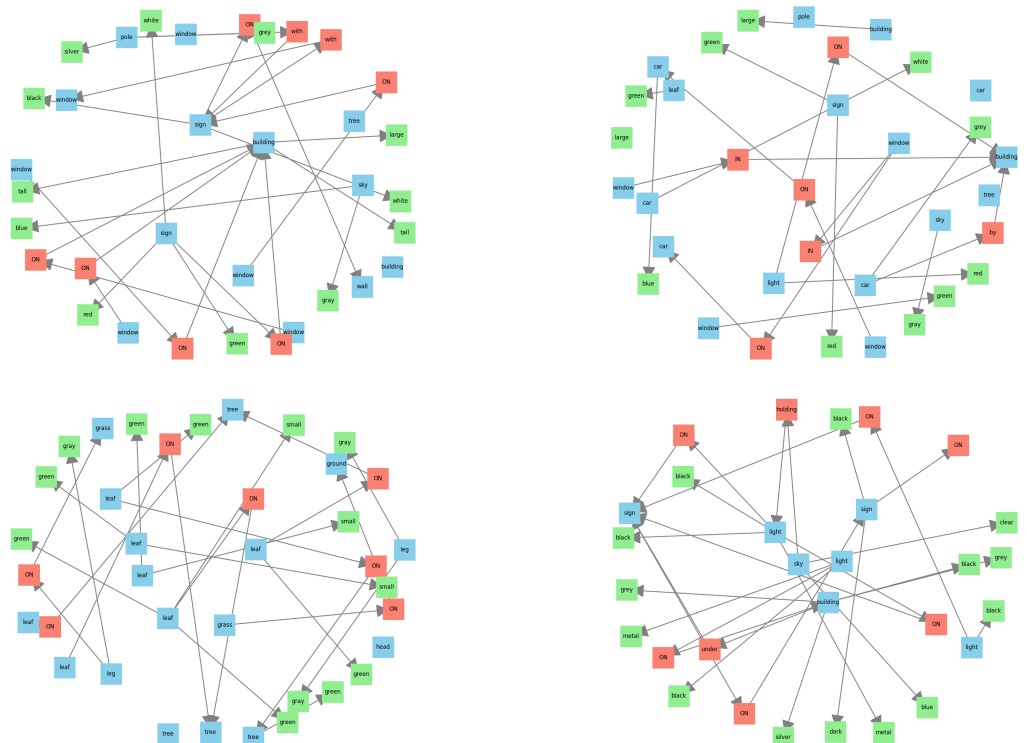

Figure 16: Visualizations of four generated digraphs for the Visual Genome dataset with RRWP positional encoding. Nodes represent objects (blue), relationships (red), and attributes (green).

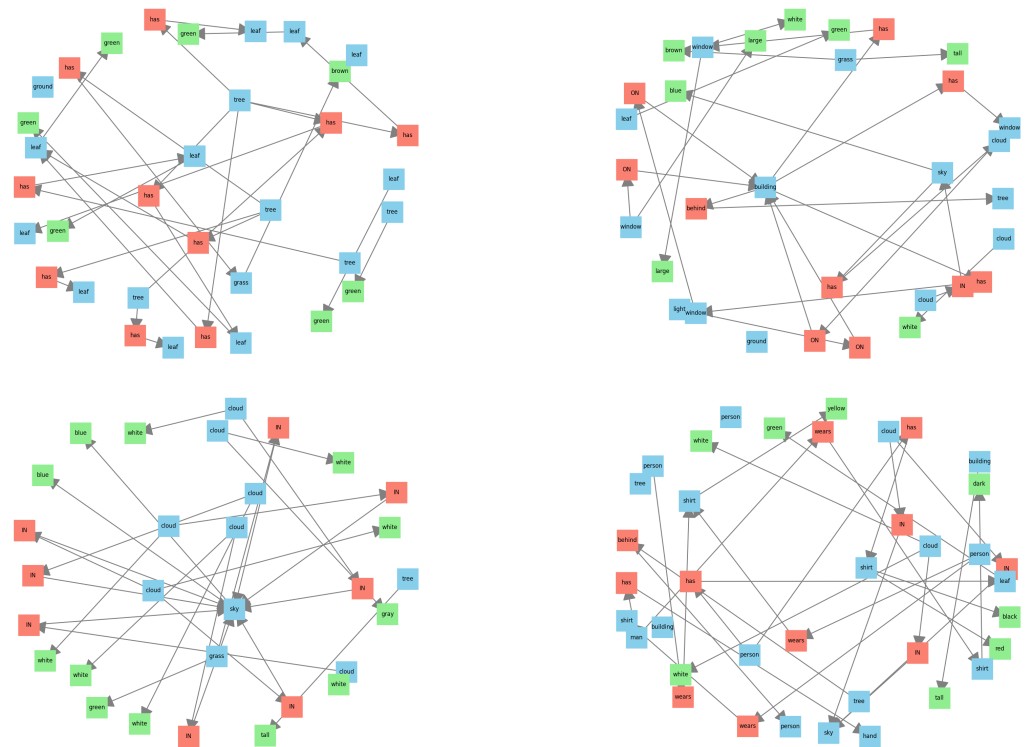

Figure 17: Visualizations of generated digraphs for the Visual Genome dataset with MagLap ($Q = 5$) positional encoding. Nodes represent objects (blue), relationships (red), and attributes (green).

## K    LIMITATIONS AND FUTURE WORK

Although DIRECTO shows strong performance in the generation of directed graphs, it suffers from inheren lmitiations, that open several avenues for future work. These are mostly addressed at further enhancing its scalability, controllability, and applicability.

**Scalability to Large Graphs and Datasets:**    Directed graphs present a combinatorial explosion in the space of possible edges due to asymmetry, posing challenges in both training and sampling efficiency.  Future work could explore *sparsity-aware attention mechanisms* (Qin et al., 2025b) to reduce quadratic memory and computational complexity. *Hierarchical* or *latent diffusion approaches* (Bergmeister et al., 2024; Jang et al., 2024; Yang et al., 2024) may allow the model to efficiently generate large-scale graphs by operating at multiple levels of granularity or in a lower-dimensional latent space. Investigating adaptive sampling schedules and dataset subsampling strategies could also help manage computational costs without sacrificing model quality.

**Conditional Graph Generation:**    Many real-world applications require generating graphs conditioned on specific properties or context, such as node features, initial subgraphs, or global graph characteristics.  DIRECTO already supports *classifier-free* conditional generation, which allows flexible conditioning without requiring additional networks. However, this approach may not be optimal for all tasks, and exploring *alternative conditional strategies* (such as learned conditional embeddings or auxiliary networks) could further improve performance. Such extensions would enable targeted graph design, e.g., generating traffic networks with predefined entry/exit nodes, molecular graphs with given scaffolds, or DAGs adhering to specific causal structures

**Explicit Structural Constraints:**    While DIRECTO can implicitly learn structural constraints such as acyclicity, certain domains require *strict adherence* to properties like DAG structure, planarity, or node degree limits. Integrating ideas from methods such as ConStrucy (Madeira et al., 2024) or PRODIGY (Sharma et al., 2024) could enforce these constraints explicitly, enhancing control over

generated graphs. Future research could also explore *soft vs. hard constraint integration*, allowing a trade-off between flexibility and domain-specific validity.

**Integration with Downstream Tasks:** Beyond purely generative evaluation, integrating DIRECTO with downstream tasks such as causal inference, traffic simulation, or molecular optimization—could highlight practical benefits and guide model improvements. Extending the metrics in the benchmark to further reflect *task-specific objectives* could be relevant to quantify real-world impact.

**Interpretability and Controllability:** Understanding which components (e.g., dual attention vs. directional positional encodings) drive specific structural properties in the generated graphs is an open research question. Future work could investigate *interpretable latent representations* and mechanisms to control generation in a predictable manner.

