# OpenReview forum: "Generating Directed Graphs with Dual Attention and Asymmetric Encoding"
_ICLR.cc/2026/Conference — ICLR 2026 Poster_

### Official Review · Reviewer_gox2 · 2025-10-28

**Soundness:** 3
**Presentation:** 3
**Contribution:** 3
**Rating:** 6
**Confidence:** 3

**Summary:**

DIRECTO is a discrete-state, iterative-refinement generator specifically designed for directed graphs. It introduces two key enhancements: (1) direction-aware positional encodings, and (2) a dual-attention transformer that explicitly combines source-to-target and target-to-source channels. The training process employs discrete flow matching, and the authors also present a discrete-diffusion variant. The generated graphs are evaluated on various benchmarks, including synthetic directed/DAG distributions, TPU compute DAGs, and Visual Genome scene graphs. The evaluation metrics focus on validity, uniqueness, and novelty, as well as a normalized MMD Ratio and label-aware distances on real-world data.

**Strengths:**

Treating directionality as a first-class citizen (both in PEs and attention) is long overdue. Dual attention is a clean, architecture-level bias that addresses asymmetry.

The comparison between “dual vs double depth” demonstrates a genuine architectural advantage, not merely an increase in capacity.

Evaluating typed constraints on Visual Genome and acyclicity on DAGs is the appropriate approach. The V.U.N. effectively penalizes memorization.

The same directional concepts also apply to discrete diffusion (DIRECTO-DD), indicating that they are not exclusive to DFM-specific techniques.

**Weaknesses:**

1. Directed attention doubles attention maps, while MagLap/Multi-q PEs are expensive. Additionally, CTMC sampling requires numerous steps to achieve quality. The paper claims decent scaling, but we need to compare it to strong autoregressive digraph models on larger graphs, considering wall-clock time and VRAM usage versus.

2. The benchmarks are relatively small and close to the training support. We need out-of-distribution (OOD) tests, such as altered degree exponents, flipped community asymmetry, or different label marginals. This will help us understand how brittle dual attention is when arrow statistics shift.

**Questions:**

How much of the performance improvement is attributed to (a) splitting the roles (S/T) versus (b) employing the aggregation trick (concat two maps and then apply one softmax)? Please provide an ablation study that distinguishes between (i) role-splitting alone, (ii) role-splitting with FiLM edge modulation, and (iii) role-splitting with a unified softmax.

Were the sampling steps, time, and VRAM used in each model (including DIRECTO-DD) consistent? Please include a fairness table.

The authors consider every ordered pair as a categorical “edge (including absent).” How is the graph size N determined at the generation time, and do the metrics account for N?

---

> ### Author Response · Authors · 2025-11-21
> **Response to the reviewer**
>
> We thank the reviewer for their thoughtful and encouraging assessment and appreciate their recognition of the motivation behind dual attention, of our ablation strategy, the relevance of our evaluations, and the portability of our directional design across both flow-matching and diffusion frameworks. Below, we address the remaining questions and comments.
>
> **W1: Scalability considerations and comparison with other AR digraph models** - Our claims about scalability are made specifically with the intention of comparing to the components and frameworks typically adopted in graph iterative refinement methods. In particular, we highlight the higher efficiency of discrete flow matching compared to discrete diffusion-based approaches. Moreover, while dual attention doubles the attention maps, it yields better performance than simply doubling the parameters of a standard network, as shown in the scalability ablation (Section 5.2). We also identify the RRWP positional encodings as the most scalable among those investigated. We have revised the manuscript to make this clearer.
>
> We agree with the reviewer that scalability to very large graphs is an important research direction. As acknowledged in the paper, the improved efficiency of DIRECTO does not fully eliminate the inherent challenges of scaling graph generation to such sizes. Nevertheless, while DIRECTO cannot yet handle graphs with thousands of nodes, it achieves strong generative performance across all graph scales involved in the applications explored in the paper. Furthermore, the method is fully compatible with sparse training and sampling frameworks such as SparseDiff [1], which would substantially improve scalability. Exploring such sparse implementations is a natural extension of our work that we envision and explicitly mention as future work.
>
> Regarding comparisons with autoregressive approaches, we benchmark DIRECTO against, to the best of our knowledge, the only two existing autoregressive models specifically designed for directed graph (in particular DAG) generation, D-VAE and LayerDAG. To broaden the evaluation, we also adapt two strong undirected diffusion and flow-based models to the directed setting. Across all baselines, DIRECTO achieves consistently stronger generative performance, and we have added a VRAM and wall-clock comparison in our answer to Question 2. We would be happy to consider any additional methods the reviewer deems relevant for an even more meaningful comparison.
>
> **W2: OOD generalization** - Unconditional graph generation aims to faithfully model the target data distribution rather than handle deliberate distribution shifts. While OOD evaluation is an interesting research direction, it falls outside the primary scope of this work and is not standard practice in unconditional graph generative modeling [2-5]. Additionally, for directed graphs, OOD shifts such as altered degree exponents or flipped community asymmetry would fundamentally redefine the target distribution rather than test generalization ability, making such evaluations less meaningful for an unconditional flow-based model like DIRECTO.
>
> Moreover, DIRECTO is evaluated across diverse settings, including multiple synthetic datasets and two real-world applications (neural architecture search and scene understanding), and achieves strong performance across all these regimes, indicating robust behavior under a variety of graph structures. For tasks where distribution shifts at inference time are meaningful, such as conditional or controllable generation, we include experiments in the appendix showing that DIRECTO is amenable to off-the-shelf conditional generation approaches, allowing it to steer generation toward different structural regimes based on external labels or signals. DIRECTO achieves solid performance in these settings, and we view richer conditional control mechanisms as a promising direction for future extensions of the framework. We therefore agree that studying robustness under structured distribution shifts is an interesting avenue for future work.
>
> **Q1: Further ablations** - We provide the results of an additional ablation study to highlight the relevance of using FiLM layers for edge modulation and the softmax in the aggregation trick. FiLM layers are used to modulate node features based on edge information and to additionally modulate edge and node features based on global information. The former allows edges to directly influence attention, which is especially relevant when edges encode direction, while the latter provides global context during generation. Additionally, the softmax in the aggregation trick forms a stable convex combination of source and target attention maps, preventing scale mismatches between the two sets of logits.

---

> ### Author Response · Authors · 2025-11-21
> **Response to the reviewer (continued)**
>
> Following the reviewer’s suggestions, we perform three ablations:
> - Removing the FiLM layer that incorporates edge features into the node representations for the attention maps (role-splitting + softmax).
> - Removing the softmax in the aggregation trick (i.e. just raw concatenation of the two ST and TS attention maps, role-splitting + FiLM).
> - Removing both of this components at the same time (role-splitting alone).
>
> The results for the ablations are presented below:
>
> Table 1. Ablation results for role-splitting, the aggregation trick, and the use of FiLM layers. All models were trained with RRWP positional encoding.
> |Dataset|Model|Ratio ↓|Valid ↑|Unique ↑|Novelty ↑|V.U.N. ↑|
> |-|-|-|-|-|-|-|
> |ER-DAG|Role-splitting + FiLM + softmax|**1.7±0.1**|94±1.0|100±0.0|100±0.0|**94±1.0**|
> ||Role-splitting alone|6.3±0.2|0.0±0.0|100±0.0|100±0.0|0.0±0.0|
> ||Role-splitting + FiLM|1.7±0.2|74.5±1.9|100±0.0|100±0.0|74.5±1.9|
> ||Role-splitting + softmax|6.9±0.6|4±3.0|100±0.0|100±0.0|4±3.0|
> |SBM|Role-splitting + FiLM + softmax|**1.8±0.5**|99.5±3.7|100±0.0|100±0.0|**99.5±3.7**|
> ||Role-splitting alone|16.7±2.8|0.0±0.0|100±0.0|100±0.0|0.0±0.0|
> ||Role-splitting + FiLM|2.4±1.4|89.0±3.7|100±0.0|100±0.0|89.0±3.7|
> ||Role-splitting + softmax|15.8±3.3|0.0±0.0|100±0.0|100±0.0|0.0±0.0|
>
> It can be seen that FiLM layers play an important role in capturing the underlying distribution. On the other hand, the aggregation trick with softmax manages to achieve a good performance in terms of ratio, but the validity is reduced. This pattern can be seen for both ER-DAG and SBM. We thank the reviewer for bringing this point out which has allowed us to extend our ablation studies (added to Appendix H.5 in the revised manuscript) and to further demonstrate the importance of our model components.
>
> **Q2: Fairness table** - We have updated and extended the section on "resources and runtime" (Appendix G.3 in the revised manuscript) to further accommodate the information for Directo-DD and the three baseline models considered. We have also added the VRAM consumption information.
>
> Table 2. Fairness comparison across models.
> |Dataset|Model|Training graphs| Training time (h)| Graphs sampled| Sampling time (min/sample)| VRAM (GB)|
> |-|-|-|-|-|-|-|
> |ER-DAG|MLE|128|0.001|40|0.02|0.15|
> ||D-VAE|128|33.8|40|1.2|9.07|
> ||LayerDAG|128|0.2|40|0.1|1.21|
> ||DiGress|128|22|40|0.2|5.21|
> ||DeFoG|128|13.6|40|0.1|5.71|
> ||Directo-DD (RRWP)|128|14.5|40|0.2|5.21|
> ||Directo-DD (MagLap $Q=1$)|128|18.2|40|0.4|5.21|
> ||Directo-DD (MagLap $Q=5$)|128|20|40|1.2|5.24|
> ||Directo-DD (MagLap $Q=10$)|128|26.5|40|2.5|5.30|
> ||Directo (RRWP)|128|13.5|40|0.3|5.72|
> ||Directo (MagLap $Q=1$)|128|14.4|40|0.5|5.72|
> ||Directo (MagLap $Q=5$)|128|16|40|0.8|5.75|
> ||Directo (MagLap $Q=10$)|128|25|40|1.5|5.71|
> |SBM|MLE|128|0.002|40|0.01|0.15|
> ||DiGress|128|15|40|0.2|26.81|
> ||DeFoG|128|23.7|40|0.3|28.46|
> ||Directo-DD (RRWP)|128|21.3|40|0.4|26.83|
> ||Directo-DD (MagLap $Q=1$)|128|20.4|40|1.3|26.81|
> ||Directo-DD (MagLap $Q=5$)|128|21|40|1.8|26.85|
> ||Directo-DD (MagLap $Q=10$)|128|20|40|3.1|26.81|
> ||Directo (RRWP)|128|13.7|40|0.5|28.34|
> ||Directo (MagLap $Q=1$)|128|16|40|2.1|28.45|
> ||Directo (MagLap $Q=5$)|128|19|40|5.1|28.49|
> ||Directo (MagLap $Q=10$)|128|20|40|6|28.46|
> |TPU Tiles|MLE|5040|0.26|40|0.01|0.15|
> ||D-VAE|5040|OOM|OOM|OOM|OOM|
> ||LayerDAG|5040|1|40|0.5|1.75|
> ||DiGress|5040|26.3|40|1.9|30.56|
> ||DeFoG|5040|20.3|40|1.1|30.58|
> ||Directo-DD (RRWP)|5040|24.3|40|2.2|30.62|
> ||Directo-DD (MagLap $Q=1$)|5040|25|40|2.4|30.56|
> ||Directo-DD (MagLap $Q=5$)|5040|31|40|2.9|30.60|
> ||Directo (RRWP)|5040|21|40|1.1|30.60|
> ||Directo (MagLap $Q=1$)|5040|30|40|2.8|30.58|
> ||Directo (MagLap $Q=5$)|5040|30|40|3.2|30.61|
> |Visual Genome|MLE|203|0.02|63|0.01|0.15|
> ||DiGress|203|11|63|1.0|11.03|
> ||DeFoG|203|13.6|63|0.9|11.22|
> ||Directo-DD (RRWP)|203|8|63|1.0|11.25|
> ||Directo-DD (MagLap $Q=1$)|203|9|63|1.4|11.03|
> ||Directo-DD (MagLap $Q=5$)|203|12|63|1.5|11.06|
> ||Directo (RRWP)|203|9|63|1.1|11.23|
> ||Directo (MagLap $Q=1$)|203|8|63|1.2|11.22|
> ||Directo (MagLap $Q=5$)|203|10|63|1.3|11.25|
>
> We observe that our proposed models have roughly one order of magnitude more parameters than LayerDAG, which explains their longer training times. In contrast, they remain smaller and faster to train than D-VAE. Regarding sampling time, our models are slightly slower than LayerDAG but remain in the same order of magnitude, and are substantially faster than D-VAE. For the diffusion and flow-based models, we use 1000 sampling steps. Finally, across all the proposed variants, the RRWP-based models are generally the fastest in both training and sampling. We thank the reviewer for raising this important point that has helped us improve the transparency of our proposed method.

---

> ### Author Response · Authors · 2025-11-21
> **Response to the reviewer (continued)**
>
> **Q3: Graph size** - The reviewer’s observation regarding the edges is correct: in our formulation, every ordered node pair is treated as a categorical edge type, including the “no-edge” (absent) case, which is considered an additional class.
>
> The graph order, and thus its size, is determined at the beginning of the sampling procedure by drawing from the empirical marginal distribution of the number of nodes in the training set (Step 3 in Algorithm 2, Appendix D) as it is standard practice in iterative refinement methods for graph generation [2-5]. This sampled size remains fixed throughout the generation process.
>
> Moreover, while some metrics implicitly depend on graph size through the structural properties they assess, this does not confound the evaluation. For example, smaller DAGs are easier to generate because accidental cycles are less likely, and structural statistics such as degree distributions or clustering patterns also vary naturally with $N$. Since $N$ is fixed from the data distribution for both training and generation, graph size is not an uncontrolled variable and explicit size-matching metrics are not required. Finally, our evaluation setup aligns with standard practice in unconditional graph generative modeling [2-5], where the goal is to match the target distribution rather than to learn or generalize the number of nodes itself.
>
> We appreciate the reviewer’s insightful feedback and hope our response addresses their remaining concerns. We are available for any further clarification.
>
> **References**
>
> [1] Qin, Y., et al, Sparse Training of Discrete Diffusion Models for Graph Generation. arXiv 2311.02142, 2023.
>
> [2] Vignac, C., et al., DiGress: Discrete Denoising diffusion models for graph generation. ICLR, 2023.
>
> [3] Xu, Z., et al., Discrete-state Continuous-time Diffusion for Graph Generation. NeurIPS, 2024.
>
> [4] Siraudin, A., et al., Cometh: A continuous-time discrete-state graph diffusion model. TMLR, 2025.
>
> [5] Qin, Y., et al., DeFoG: Discrete Flow Matching for Graph Generation. ICML, 2025.

---

> > ### Comment · Reviewer_gox2 · 2025-11-25
> >
> > Thanks authors for the details response and additional experiments! I will increase my score.

---

### Official Review · Reviewer_RPrZ · 2025-10-31

**Soundness:** 3
**Presentation:** 3
**Contribution:** 3
**Rating:** 6
**Confidence:** 3

**Summary:**

The paper proposes Directo, a method for generating directed graphs based on discrete flow matching. Great care is given to the denoising model arch. To enhance the capability of the method to handle edge directionality, 1) a tailored attention mechanism called "dual attention" is proposed; and 2) asymmetric position encodings, such as those based on directional Laplacians, are used. Other GNN components from prior work, FiLM and PNA, are also incorporated into the arch. Besides the method, the paper also creates a benchmark for directed graph generation, with several synthetic and real-world datasets. Directo is shown to have strong performance on this benchmark relative to other methods, based on evaluation metrics adapted to the setting. Finally, further experiments are discussed that explore the impact of ablating the major model arch components, as well as the scalability of the method.

**Strengths:**

- The area of directed graph generation seems important but relatively unexplored given the many works for undirected graph generation.
- The writing is clear, grammatical, and well-organized.
- The paper introduces not only a new method, but also a benchmark for the area of directed graph generation. A code repository is included for reproducibility.
- Ablations are included to justify major new components of the model arch.

**Weaknesses:**

- The proposed arch is complex, and while there are ablation studies for some important components, this is not true of all components, e.g., use of FiLM.
- As the authors note, the proposed method can fail to maintain validity when scaling up (e.g., failing to maintain strict acyclicity beyond 200 nodes).
- Ideally the broad setting of the paper in terms of the scale of generated graphs would be clarified earlier in the paper. Graph generation papers and algorithms roughly cluster on two categories, those for 10s-100s of nodes (this paper), and others for 1000s-10000s and up.
- (nit) Some notation could be easier to read, e.g., the use of Kronecker delta in Eq 1 could be replaced with piecewise notation.

**Questions:**

- Much of the main paper describes a complex GNN arch tailored for directed graphs, which could be applied outside of the graph generation setting, e.g., to link prediction or node classification. Has this been attempted? It seems worthwhile to evaluate the arch on the more standardized benchmarks for those tasks. If the results are weak on other tasks but strong for generation, that is also an interesting finding.
- Relatedly, there is little discussion of the flow matching part in the main paper. Were there any interesting findings related to the flow matching as opposed to the arch that have not been discussed in prior work?
- Regarding the time distortion function, it is stated that "these functions are selected based on dataset-specific properties to improve fidelity and structural constraints without need for retraining." Could you please clarify the selection procedure, given that there is not a unique objective unlike in typical hyperparameter selection via cross-validation, e.g., for node classification?
- For graph generation papers in general, a core theme is the trade-off between generating diverse graphs (U.N. from V.U.N.) and graphs that match the training distribution ("ratio" in this paper). Is there some way to tune the proposed method to favor one or the other? How does the ease of such tuning compare to other methods?

---

> ### Author Response · Authors · 2025-11-21
> **Response to the reviewer**
>
> We thank the reviewer for highlighting the importance of the directed graph generation setting and for their positive evaluation of the clarity of our presentation, the contributions of our benchmark, and the thoroughness of our ablations. Below, we address the reviewer’s questions and concerns in detail.
>
> **W1: Ablation on the use of FiLM** - We agree with the reviewer that, given the architectural richness of DIRECTO, additional ablations are valuable. In particular, FiLM layers are used to modulate node features based on edge information and to additionally modulate edge and node features based on global information. The former allows edges to directly influence attention, which is especially relevant when edges encode direction, while the latter provides global context during generation. We ablate over the FiLM modulation of nodes based on edges.
>
> Other reviewers raised similar questions regarding additional architectural components, so we have integrated all relevant ablation results in this part of the response. In particular, our design also includes gated residual connections for node updates, which regulate how much new information is incorporated at each step (acting as a learned importance weight), and a softmax-based aggregation mechanism that forms a stable convex combination of the source and target attention maps, preventing scale mismatches between the two sets of logits.
>
> The results in Table 1 show that FiLM is a key component to generate both valid and correct structures. The gated residual connections and the softmax in the aggregation trick also play an important role.
>
> Table 1. Ablation results for different model components. All models were trained with RRWP positional encoding.
> |Dataset|Model|Ratio ↓|Valid ↑|Unique ↑|Novelty ↑|V.U.N. ↑|
> |-|-|-|-|-|-|-|
> |ER-DAG|Original|**1.7±0.1**|94±1.0|100±0.0|100±0.0|**94±1.0**|
> ||No FILM|6.9±0.6|4±3.0|100±0.0|100±0.0|4±3.0|
> ||No softmax|1.7±0.2|74.5±1.9|100±0.0|100±0.0|74.5±1.9|
> ||No FILM + No softmax|6.3±0.2|0.0±0.0|100±0.0|100±0.0|0.0±0.0|
> ||No gated residual connection|2.5±0.2|66.5±6.0|100±0.0|100±0.0|66.5±6.0|
> |SBM|Original|**1.8±0.5**|99.5±3.7|100±0.0|100±0.0|**99.5±3.7**|
> ||No FILM|15.8±3.3|0.0±0.0|100±0.0|100±0.0|0.0±0.0|
> ||No softmax|2.4±1.4|89.0±3.7|100±0.0|100±0.0|89.0±3.7|
> ||No FILM + No softmax|16.7±2.8|0.0±0.0|100±0.0|100±0.0|0.0±0.0|
> ||No gated residual connection|2.7±1.4|91.5±1.7|100±0.0|100±0.0|91.5±1.7|
>
> We have added this ablation to the supplementary material (Appendix H.5 of the revised version), helping justify our choice and complementing the broader set of ablations on our architecture that we already include. While DIRECTO’s architecture is indeed rich, this complexity is motivated by the challenges inherent to modelling directed graphs and is supported by our ablation results. For example, models without dual attention fail to learn meaningful directional structure (see Section 5.2), and removing other components such as the gated residual connections for the nodes, the softmax in the aggregation, or the FiLM-based feature modulation can substantially degrade performance. Taken together, these ablations show that the proposed components are necessary for achieving a strong level of performance.
>
> **W2 & W3: Scalability** - DIRECTO follows other works such as DiGress [1] or CatFlow [2], and belongs to the class of graph generative models dealing with small to medium sizes of graphs. We updated the manuscript to emphasize this explicitly in the introduction (line 76).
>
> We agree with the reviewer that maintaining validity becomes increasingly challenging as graph size grows beyond 200 nodes. Nonetheless, DIRECTO is able to handle the largest graphs on which prior directed generative models have been evaluated. In addition, DIRECTO supports training and sampling on larger graphs when such sizes are present in the dataset, as demonstrated on the TPU Tiles dataset, which includes graphs with up to 400 nodes. Across all these settings, our model maintains strong performance.
>
> **W4: Notation** - We appreciate the reviewer's suggestion. We adopted the notation from previous work that introduced Discrete Flow Matching [3, 4], and have revised the manuscript to better clarify it (line 145).
>
> **Q1: Extension to other tasks** - We appreciate the reviewer highlighting this interesting direction. While our focus is on generative modeling, the proposed transformer might indeed be applicable to discriminative tasks such as link prediction and node classification by modifying the current graph-to-graph architecture (e.g. by masking the unnecessary entries of the output) and evaluating in the appropriate benchmarks. To encourage that, we have revised our manuscript to add it to the future lines of research proposed (lines 514-519).

---

> ### Author Response · Authors · 2025-11-21
> **Response to the reviewer (continued)**
>
> **Q2: Results on Flow Matching** - Our primary goal was to extend graph generation to the directed setting. For that, we identified that the current major challenge lies primarily in the architecture and not that much in the Discrete Flow Matching setting, as demonstrated by DeFoG, which already nearly-saturates performance in the undirected benchmarks and is robust across different datasets for the size of graphs that we are handling.
>
> Therefore, we identified the expressivity of the architecture as the bottleneck for digraph generation performance and focused more on how to add a better directed prior into the model. Nonetheless, we agree that further discussion of flow matching itself is also interesting. In particular, in our experiments we typically observed trade-offs between VUN and ratio metrics depending on the chosen sampling hyperparameters, suggesting that different configurations bias the model more towards structural fidelity or towards validity, uniqueness, and novelty (we provide further details and intuitions about this in the answer to question 4). This indicates that there is an interesting space of flow-matching-specific effects worth investigating in future work.
>
> **Q3: Time distortion** - The time distortion function is selected post-training: after training the model under the default configuration, we optimize the sampling procedure by performing a grid search on different distortion functions. In particular, we test the ones proposed in prior work on flow-matching for graph generation [4] and choose the one that yields the best empirical trade-off between distributional alignment with the training set (essentially measured by the ratio) and validity and generation diversity (measured by V.U.N.), under reduced sampling steps.
>
> **Q4: Trade-off**
>
> - As the reviewer correctly points out, there is an inherent trade-off between generating diverse graphs and generating graphs that closely match the training distribution. In our case, from a practical perspective, we tune this by using the stochasticity and target guidance hyperparameters (described in detail in Appendix C) at sampling time. As it can be seen in Appendix H.9 (of the revised version), there is a visible trade-off showing how different sampling hyperparameters affect V.U.N. and the average ratio. In particular, we typically observe that higher values of stochasticity result in a high V.U.N. but high MMD ratio, and high target guidance results in low ratio but low V.U.N.. This might be explained by the fact that increasing stochasticity allows the denoising trajectory to get further from the learned flow, allowing for more exploration/diversity but reducing fidelity to the training distribution, increasing the ratio metric. On the other hand, higher target guidance helps the model align better with the original distribution at sampling time, resulting in low ratio but potentially incurring in memorization, harming generation diversity. A more principled investigation on this issue would be valuable, and we view this as a promising direction for future work.
> - The observations above indicate that DIRECTO’s sampling optimization offers a practical way to steer the model toward higher diversity or closer distributional fidelity, depending on the desired result. This follows the trend of DeFoG, which makes the tuning particularly efficient when compared with discrete-diffusion based models, since it happens only at sampling time, without the need for retraining the entire model. Additionally, notice that this sampling stage tuning requires only a small portion of the overall training time (~1h for ER-DAG, ~3h for SBM, ~3h for TPU and ~30min for Visual Genome, which represent only a small fraction of the overall training time detailed in Table 2 in the comment for Reviewer gox2). More details on how this tuning is performed efficiently in practice are available in Appendix C.
>
> We appreciate the reviewer’s insightful feedback and hope our response addresses the reviewer’s concerns. We remain available for any further clarification.
>
> **References**
>
> [1] Vignac, C., et al., DiGress: Discrete Denoising diffusion models for graph generation. ICLR, 2023.
>
> [2] Eijkelboom, F., Variational Flow Matching for Graph Generation. NeurIPS, 2024.
>
> [3] Campbell, A., et al., Generative Flows on Discrete State-Spaces: Enabling Multimodal Flows with Applications to Protein Co-Design. ICML, 2024.
>
> [4] Qin, Y., et al., DeFoG: Discrete Flow Matching for Graph Generation. ICML, 2025.

---

### Official Review · Reviewer_pVm7 · 2025-11-06

**Soundness:** 3
**Presentation:** 1
**Contribution:** 2
**Rating:** 4
**Confidence:** 3

**Summary:**

This work proposes a model for generating directed graphs. Its main contributions are in (i) the use of direction-aware positional encodings, (ii) a dual attention mechanism that again takes edge directions into account and (iii) the introduction of new benchmarks for evaluating directed graph generative models. The performance is measured across different synthetic and true graphs and seemingly outperforms other undirected GGM or specialized in generating DAG models. An ablation study shows that the dual attention mechanism is critical in the increased performance.

**Strengths:**

* The model proposed in this work succeeds in generating graphs with specific constraints (acyclic, or edges only across certain types) even if those constrains are not explicitly stated (e.g., through some regularization parameter) during the training process.
* The architecture is quite generic in that it can be used to generate directed graphs of virtually any type as edges are encoding through categorical variables, while it can also accomodate node and graph features.

**Weaknesses:**

*  The model itself, at least as presented, is not self-contained. Section 2 which provides a background on diffusion models and past work seems disconnected to Section 3 that describes the attention mechanism employed. For example, it is not specified how edges are modeled through categorical variables (e.g, is it representing one of the 4 possible classes -- edges in both directions, in one of the two (x2), or absent?), how the rate matrix is parameterized though the proposed architecture, while fig. 2 employs global features that are not part of the loss function in eq. 4.
* The complexity of the proposed model seems to make it a bit susceptible to choice of hyperparameters (see fluctuation in performance in fig 4).
* The novelty of the work is on the dual attention mechanism, but it does not extend to other parts of it (e.g., new positional encoding, diffusion process is of DFM, evaluation metrics is MMD).

**Questions:**

Q1: are global features utilized/ part of the training loss? what is the categorical modeling for edges for the datasets the model is evaluated against?
Q2: What is the benefit of MagLap encoding vs the directed laplacian of chung?
Q3: Ratio comes from averaging the MMD for different descriptors. Is it though a good practice to average MMD for different descriptors when it is unbounded and lacks scale comparison?

---

> ### Author Response · Authors · 2025-11-21
> **Response to the reviewer**
>
> We thank the reviewer for their positive assessment of our architectural design and for recognizing the model’s ability to generate graphs that respect structural constraints without explicitly imposing them. We address the raised questions and concerns below:
>
> **W1 + Q1: Model presentation** - We thank the reviewer for the detailed reading and appreciate the opportunity to further clarify our presentation.
>
> Section 2 introduces the relevant notation and flow matching for graph generation because it is the underlying generative paradigm in which our model operates, and which uses a graph transformer with graph positional encodings as the denoising neural network. However, this framework alone is not sufficient to ensure effective directed graph generation. Therefore, Section 3 details how to extend previous work on discrete flow matching for undirected graph generation to later focus on our methodological contributions, which are mainly the use of directed positional encodings and more importantly the design of a direction-aware attention mechanism. We have revised the paper to clarify the progression of ideas and improve the section transition.
>
> Regarding edge modeling, we follow the pairwise categorical formulation described in the notation: each ordered pair of distinct nodes corresponds to a categorical variable. When edges are unattributed, this reduces to two categories, “edge present’’ and “edge absent’’. When edge types are available, the categorical space expands to include all edge labels plus the “absent’’ class. We have revised the paper to make this enumeration more explicit.
> Therefore, each directed edge (namely, each entry of the adjacency matrix without any symmetrization) is treated separately and assigned one of the possible classes. This formulation allows edges $(i,j)$ and $(j,i)$ to take different classes: both may be absent (no connection between nodes $i$ and $j$), only one may exist, or both may exist, even with different classes. This fully captures the generality of directed graphs.
>
> Since we adopt the same parametrization of the rate matrix as in prior work [1, 2], we originally omitted the full details from the main text but provided a more detailed explanation in Appendix I.2 with details on the hyperparameter optimization in Appendix C. Nonetheless, to improve the progression of ideas in the paper, in the updated manuscript, we added a brief description of how the denoising GNN predictions are integrated into the design of the rate matrix and the corresponding pointer to appendix for complete formulation.
>
> Finally, as correctly noted by the reviewer, the global features are not used explicitly in the loss function and the cross-entropy loss is applied only at the node and edge level. Nevertheless, the *updated* global features at every layer are incorporated into the model through FiLM layers, where they modulate the node and edge updates, and are therefore implicitly taken into account when computing the loss. We have revised Figure 2 to make this more explicit.
>
> We appreciate the reviewer’s comments, as they helped us improve the flow of the paper and the transitions between sections. We have revised the paper to incorporate the discussions of the four paragraphs above (mainly in lines 158-184).
>
> **W2: Choice of hyperparameters** - We view robustness as one of DIRECTO’s core strengths. Using the same training strategy and sampling optimization recipes across datasets, the model consistently delivers strong performance. The variation observed in Figure 4 reflects the flexibility of discrete flow matching rather than instability, being one of the advantages of this methodology over alternative frameworks such as discrete diffusion. Since in flow matching training and sampling are decoupled, it enables efficient optimization of sampling hyperparameters without requiring full retraining. As discussed in Section H.9 (of the revised version), exploring these hyperparameters can be done with a reduced number of sampling steps (and thus modest computational cost), and still can lead to substantial performance improvements. In particular, the search takes ~1h for ER-DAG, ~3h for SBM, ~3h for TPU and ~30min for Visual Genome, which represents only a small fraction of the overall training time (see Table 2 in the comment for Reviewer gox2).
>
> **W3: Novelty** - We value the reviewer's appreciation of the dual attention mechanism proposed. While we agree that our methodology builds upon previous foundations such as DFM, we believe that adapting them meaningfully to the directed graph domain introduces substantial modeling challenges. Moreover, we are, to the best of our knowledge, the first generative method to employ directed positional encodings for generative tasks.

---

> ### Author Response · Authors · 2025-11-21
> **Response to the reviewer (continued)**
>
> We would also like to emphasize the benchmarking framework as a major contribution of our work. We believe that general-purpose digraph generation is still underexplored. By releasing a benchmark with diverse datasets (including DAGs, relevant synthetic distributions, and real-world settings) and tailored metrics (note that beyond MMD metrics, we also consider validity, uniqueness and novelty, and other joint node-edge distributional alignment metrics such as RBF and FID over GNN embeddings for a comprehensive graph generative performance evaluation), we aim to set a precedent for future work. This follows the direction advocated in recent position papers, which highlight the need for new benchmarks in graph learning, and particularly in the directed setting [3].
>
> **Q2: Directed Laplacian** - When proposing a generative model tailored to directed graphs, we first envisioned using a naive version of the Laplacian applied to digraphs, where we consider the symmetrized versions of the adjacency matrix ($\frac{A + A^\top}{2}$). Nonetheless, this resulted in poor performance. In contrast, we observed strong performance with the directed RRWP and with the Magnetic Laplacian [4] (and its Mult-MagLap extension [5]), which provide richer directional signals. In particular, for MagLap, their complex-valued eigenvalues and eigenvectors encode both magnitude and phase information, preserving orientation patterns that otherwise get lost in the real-valued Directed Laplacian.
>
> Nevertheless, we acknowledge the practical appeal of the Directed Laplacian from Chung [6] due to its readily available implementation on `networkx`, which made it convenient for metric design. We have also conducted a quick additional ablation incorporating the Directed Laplacian as a positional encoding to the ER-DAG dataset, which shows that it underperforms comparatively to the adopted alternatives.
>
> Table 1. Ablation results for the different positional encodings including the Directed Laplacian for the ER-DAG dataset.
> |Model|Ratio ↓|Valid ↑|Unique ↑|Novelty ↑|V.U.N. ↑|
> |-|-|-|-|-|-|
> No PE|3.0±0.2|47.0±12.1|100±0.0|100±0.0|47.0±12.1|
> |Laplacian|1.8±0.0|84.0±4.9|100±0.0|100±0.0|84.0±4.9|
> |Directed Laplacian|1.6±0.2|88±4.0|100±0.0|100±0.0|88±4.0|
> |Directed RRWP|1.7±0.1|94.0±1.0|100±0.0|100±0.0|94.0±1.0|
> |Magnetic Laplacian $(Q=1)$|1.3±0.2|91.0±2.5|100±0.0|100±0.0|91.0±2.5|
> |Magnetic Laplacian $(Q=5)$|1.3±0.2|91.5±2.5|100±0.0|100±0.0|91.5±2.5|
> |Magnetic Laplacian $(Q=10)$|1.3±0.2|92.0±3.7|100±0.0|100±0.0|92.0±3.7|
>
> **Q3: Average of MMD** - We do not average raw MMD values across descriptors and then compute a single ratio from this. The ratio we report is instead obtained by first computing a ratio per each MMD score, and then averaging these ratios across descriptors. This ratio, first introduced in [7], has become standard practice as metric for performance evaluation in graph generation literature (e.g., [7, 8]). In particular, the ratio of MMD descriptors makes each of the metrics more comparable in terms of scale and averaging allows to have a single, aggregated direct scalar for performance comparison. This avoids issues related to the scale incomparability of MMD itself, since each ratio is computed before aggregation. As for the unboundedness concerns, again following standard practice [7-9], the cases where the denominator of the ratio is 0 to the fourth decimal are skipped to avoid numerical instability.
>
> Note, nevertheless, that we acknowledge that this is not a perfect metric, and therefore we report each MMD descriptor before computing the ratio and the average in Appendix H and complement it with V.U.N. and further joint node-edge structural metrics (for attributed datasets). We have updated the description in the paper to clarify this more explicitly (lines 345-347).
>
> We appreciate the reviewer's insights and hope these results have positively addressed their concerns. We remain open to any further suggestions or questions that may arise.
>
> **References**
>
> [1] Campbell, A., et al., Generative Flows on Discrete State-Spaces: Enabling Multimodal Flows with Applications to Protein Co-Design. ICML, 2024.
>
> [2] Qin, Y., et al., DeFoG: Discrete Flow Matching for Graph Generation. ICML, 2025.
>
> [3] Bechler-Speicher, M., et al., Position: Graph Learning Will Lose Relevance Due To Poor Benchmarks. ICML, 2025.
>
> [4] Geisler, S., et al., Transformers Meet Directed Graphs. ICML, 2023.
>
> [5] Huang, Y., et al., What Are Good Positional Encodings for Directed Graphs? ICLR, 2025.
>
> [6] Chung, F., Laplacians and the Cheeger inequality for directed graphs. Annals of Combinatorics, 2005.
>
> [7] Martinkus, K., et al., SPECTRE: Spectral Conditioning Helps to Overcome the Expressivity Limits of One-shot Graph Generators. ICML, 2022.
>
> [8] Bergmeister, A., et al., Efficient and Scalable Graph Generation through iterative local expansion. ICLR, 2024.
>
> [9] Madeira, M., et al., Generative Modelling of Structurally Constrained Graphs. NeurIPS, 2024.

---

### Official Review · Reviewer_A2gN · 2025-11-08

**Soundness:** 3
**Presentation:** 4
**Contribution:** 3
**Rating:** 8
**Confidence:** 3

**Summary:**

The authors present DIRECTO, a new method for generating directed graphs, an area that has been underexplored in graph generative modeling. The approach builds upon the discrete flow matching framework and introduces two key architectural components: (1) a dual attention mechanism that captures both source-to-target and target-to-source representations, and (2) direction-aware positional encodings. The authors indicate that the dual attention component is more critical to performance than the positional encodings. Additionally, they contribute a new benchmark suite containing synthetic and real-world datasets, along with metrics tailored for evaluating directed graph generation quality. The paper demonstrates strong empirical results across diverse settings. While the work provides a solid foundation for directed graph generation, conditional generation remains relatively underexplored and is identified as an important direction for future work.

**Strengths:**

- Novel and Important Problem: The paper addresses a significantly underexplored area in graph generation. The focus on directed graphs is well-motivated, with clear applications in biology, transportation networks, and scene understanding.
- Comprehensive Technical Approach: The dual attention mechanism is presented as a solution to capture bidirectional dependencies inherent in directed graphs. The integration with discrete flow matching provides a principled generative framework with theoretical grounding.
- Extensive Empirical Evaluation: The paper presents thorough experiments across multiple datasets, baselines, and ablation studies. The results consistently demonstrate the effectiveness of the proposed approach.
- Benchmark Contribution: The introduction of standardized benchmarks for directed graph generation is a valuable contribution that will facilitate future research in this area. This work can provide a solid evaluation suite to later directions in the field.
- Clear Presentation: The paper is well-written with clear motivation, technical exposition, and comprehensive experimental analysis. All visualizations seem relevant and convey information clearly.
- Thorough Ablations: The systematic ablation studies effectively isolate the contributions of different components.

**Weaknesses:**

- Unclear Claims on Expressiveness: While the authors claim their method is "expressive" and "robust," these claims lack formal theoretical justification or empirical evidence. The notion of expressiveness in the context of directed graph generation needs clearer definition and supporting arguments.

- Scalability Concerns vs. Efficiency Claims: The paper mentions "efficient generation" but simultaneously acknowledges scalability limitations. This contradiction needs clarification. Table 20 shows significant performance degradation for larger graphs (200-250 nodes), which undermines efficiency claims.

- Conditional Generation Underexplored: While acknowledged as future work, the conditional generation experiments (Section H.7) are limited. Given the practical importance of conditional generation for real-world applications, this might deserve more attention.

Limited Architectural Justification: Several design choices appear ad-hoc:
- Why is independent interpolation used in the noising process (Eq. 1). It is this unclear if this is a simplifying assumption or theoretically motivated.
- The gated residual connection for node features (Eq. 23-24) lacks justification compared to standard residual connections used elsewhere.

**Questions:**

- Positional Encoding Design Choice: The authors concatenate positional encodings to node and edge features rather than using more integrated approaches common in modern transformers (e.g., addition, rotational embeddings like RoPE, or ALiBi). What is the rationale for choosing concatenation? Has the impact on parameter efficiency been considered, given that concatenation increases the input dimensionality? Were alternative integration methods explored?

- Figure 1b Clarification: What do the node colors represent in Figure 1b? This visual element should be explained or removed if purely aesthetic.

- Independent Interpolation Assumption: Why is independent interpolation used in the noising process (Eq. 1)? Is this a simplifying assumption for computational tractability, or is there theoretical justification for treating nodes and edges independently during the noising process?

- Architectural Choices: What is the motivation for the gated residual connection in the node feature update (Eq. 23-24) when standard residual connections are used elsewhere? Has ablation been performed on this component?

- Section 3.2 Reference: The last paragraph mentions ablation studies but doesn't specify where results are presented. Could you add a forward reference?

- TPU Tiles Dataset: For the TPU Tiles dataset, was the preprocessing identical to Li et al. (2025), or were there modifications? This should be clarified.

---

> ### Author Response · Authors · 2025-11-21
> **Response to the reviewer**
>
> We thank the reviewer for positively assessing our framework, empirical results, and evaluations. Below, we address the raised questions and concerns:
>
> **W1: Expressiveness** - We thanks the reviewer for raising this point. Our use of the term *expressive* was intended to convey that the model is more *direction-aware*, meaning that under identical parameter budgets, incorporating our architectural modifications leads to better capturing and modeling of asymmetric structure (Table 2 of the paper). As for robustness, we refer to DIRECTO's ability to adapt to datasets of very different nature while maintaining high performance. We have revised the manuscript to clarify these meanings (lines 256-257) and we remain open to further refine the wording to maximize clarity.
>
> **W2: Efficiency Claims** - Our reference to *efficient* generation concerns the computational efficiency advantage enabled by a discrete flow matching formulation vs. diffusion-based approaches, both in terms of training (quicker training convergence to a given generative performance) and sampling (less steps needed to attain that performance), as it can also be seen in Table 2 in the answer for Reviewer gox2 and as documented in prior work [1] (Section 5.2).
>
> However, this efficiency does not fully eliminate the inherent challenges, acknowledged in the paper, of scaling graph generation for very large graphs. We have revised the manuscript to clearly distinguish the algorithmic efficiency of the generative mechanism from its scaling behavior with graph size (lines 82-83), ensuring that these two points do not appear contradictory.
>
> **W3: Conditional Generation** - Our motivation for this work was to establish a strong foundation for directed graph generation, an underexplored area of research. This required substantial groundwork, and thus we chose the unconditional setting to maintain maximum generality. We completely agree that conditional generation is of major practical importance and, to reflect this, we included experiments using classifier-free guidance to show that DIRECTO can be extended in this direction with minimal architectural changes while achieving solid performance. Nonetheless, we believe there is room to develop richer conditioning strategies that may yield further performance improvements, and we see this as a promising avenue for future extensions.
>
> We address the **architectural justification** in the answers to the questions below.
>
> **Q1: Positional Encoding Design Choice** - In graph transformers, node features and positional encodings usually have different dimensionalities. Therefore, unlike in NLP where positional encodings are sized to match token embeddings and can be freely adjusted, the dimensionality of graph positional encodings is tied to structural computations (e.g., number of eigenvectors or random-walk components). Increasing their dimensionality is possible but comes with significant computational cost, as each additional component requires extra graph-level preprocessing and larger per-node representations. Using addition would then require projecting both $x$ and $\mathrm{PE}$ to the same dimension with extra transformations, thus defeating the purpose of using addition in the first place.
>
> To avoid this and following common practice in previous literature [1, 2], we concatenate the inputs and the PEs directly $\tilde{x} = [x \,\|\, \mathrm{PE}]$ and apply a single MLP to map them into a shared representation space. Finally, note that the computational overhead introduced by concatenation is incurred only in the first projection layer, after which the embedding dimensionality remains fixed. As a result, the added cost is negligible, especially since graph features remain small after concatenation (typically 10–50 dimensions) compared to standard transformer embedding sizes (e.g., 512–1024).
>
> In addition, our positional encoding strategy is also motivated by the goal of preserving the model’s symmetry-respecting properties with respect to graphs. In particular, during the model’s development, we experimented with positional encodings commonly used in modern transformers. However, because these encodings impose a fixed node ordering and thus break node permutation equivariance, they led the model to exhibit memorization behavior. We show this in Table 2, where the reduced novelty under RoPE encodings indicates clear memorization of the training set.
>
> Table 2. Results for the model with rotary positional encoding (RoPE) for the ER-DAG dataset.
> |Model|Ratio ↓|Valid ↑|Unique ↑|Novelty ↑|V.U.N. ↑|
> |-|-|-|-|-|-|
> |RoPE|1.8|91.7|90|20.7|15.4|
> |RoPE + Dual Attention|2.0|91.5|90|15.7|10.7|
>
> If the reviewer deems fit, we can incorporate Table 2 and a more detailed explanation of these design choices to the supplementary material of out paper.
>
> **Q2: Figure 1b Clarification** - Node colors represent different (categorical) node types. We have updated the caption of the figure to make this explicit.

---

> ### Author Response · Authors · 2025-11-21
> **Response to the reviewer (continued)**
>
> **Q3: Independent Interpolation** - Independent interpolation in the noising process follows the standard formulation in Discrete Flow Matching [3], and is also used in its (undirected) graph-specific adaptation [1]. This formulation ensures that at $t=0$ the complex dependencies present in the clean data distribution are removed, so sampling from the limit distribution amounts to sampling each node and edge independently. This independent noising formulation allows tractability of the denoising initialization, while guaranteed to remain expressive enough to capture the data distribution after full denoising (see corollary 2 of [1]).
>
> Importantly, the denoising process does not treat nodes and edges as independent, as the model progressively reinstates their dependencies over the course of the trajectory. This formulation is also employed by discrete generative models beyond flow matching such as D3PM and discrete diffusion in continuous time [4, 5], which also factorize over dimensions in the noising process. The strong empirical performance of these methods further supports the independent noising formulation. We clarify this in the updated manuscript by providing a more explicit relation to the established methodology (lines 140-142).
>
> **Q4: Gated residual connection** - Our motivation for using gated residual connections on the nodes is that they allow to regulate how much new information is incorporated at the node update, functioning as a learned importance weight.
>
> In our experiments, we found that using these connections for node features consistently improved performance compared to standard residual connections. Motivated by the reviewer’s comment, we conducted an additional ablation study to systematically verify this effect and complete our architectural analysis.
>
> Table 1. Ablation results for gated vs standard residual connection for node features. All models were trained with RRWP positional encoding.
> |Dataset|Model|Ratio ↓|Valid ↑|Unique ↑|Novelty ↑|V.U.N. ↑|
> |-|-|-|-|-|-|-|
> |ER-DAG|Without gating|2.5±0.2|66.5±6.0|100±0.0|100±0.0|66.5±6.0|
> ||With gating (original)|**1.7±0.1**|94±1.0|100±0.0|100±0.0|**94±1.0**|
> |SBM|Without gating|2.7±1.4|91.5±1.7|100±0.0|100±0.0|91.5±1.7|
> ||With gating (original)|**1.8±0.5**|99.5±3.7|100±0.0|100±0.0|**99.5±3.7**|
>
> The results confirm that the gated residual connections improve performance, supporting their inclusion in our final model. We have added this ablation to the supplementary material and thank the reviewer for prompting us to clarify this design choice.
>
> **Q5: Section 3.2 Reference** - We have added a forward reference to Section 5, where the main experiments are presented, as well as to the corresponding appendix section for detailed results (line 215). We thank the reviewer for helping improve the flow of the manuscript.
>
> **Q6: TPU Tiles preprocessing** - We confirm that the preprocessing for the TPU Tiles dataset follows exactly the procedures described in [6]. The manuscript has been updated to make this clearer when the dataset is introduced (line 321).
>
> We appreciate the highly constructive and encouraging feedback. The comments have helped us improve the clarity and completeness of our paper. We remain open to addressing any additional questions.
>
> **References**
>
> [1] Qin, Y., et al., DeFoG: Discrete Flow Matching for Graph Generation. ICML, 2025.
>
> [2] Vignac, C., et al., DiGress: Discrete Denoising diffusion models for graph generation. ICLR, 2023.
>
> [3] Campbell, A., et al., Generative Flows on Discrete State-Spaces: Enabling Multimodal Flows with Applications to Protein Co-Design. ICML, 2024.
>
> [4] Austin, J., et al., Structured Denoising Diffusion Models in Discrete State-Spaces. NeurIPS, 2021.
>
> [5] Campbell, A., et al., A Continuous Time Framework for Discrete Denoising Models. NeurIPS, 2022.
>
> [6] Li, M., et al., LayerDAG: A Layerwise Autoregressive Diffusion Model for Directed Acyclic Graph Generation. ICLR, 2025.

---

### Author Response · Authors · 2025-12-03
**Author Final Remarks**

We would like to thank the reviewers for their recognition of our work and their positive initial rating. At the moment the interaction with reviewers ended, Reviewer A2gN had reaffirmed their strong recommendation, while Reviewer gox2 had increased their score from 6 to 8 after their concerns were addressed. The remaining two reviewers had not yet participated in the discussion.

We appreciate the reviewers’ constructive feedback, which has helped further strengthen our contribution. During the rebuttal, we improved our manuscript and conducted additional experiments to enhance the quality of our submission. A detailed summary of our contributions and improvements is provided below.

## Strengths and core contributions

* **Novel and important problem** - Modeling directed graphs with direction-aware architectures addresses an underexplored challenge in graph generative modeling, particularly for DAGs and asymmetric relational data, as highlighted by *Reviewers A2gN, pVm7, and gox2*.

* **Extensive empirical evaluations and ablations** - *Reviewers A2gN, RPrZ, and gox2* appreciated our extended ablations and experimentation. In particular, they valued the breadth of our synthetic and real-world benchmarks, the selection of baselines, and the depth of our ablations spanning architectural components and positional encoding choices.

* **Technical and generic architecture** - *Reviewers A2gN, pVm7, and gox2* valued the architectural design, noting its efficiency and ability to generate graphs with specific constraints. In addition, *Reviewers pVm7 and gox2* emphasized that dual attention and direction-aware PEs provide principled inductive biases that can be applied across both discrete flow matching and diffusion, as well as potentially to other graph-generation tasks.

* **Benchmarking contributions** - The proposal of a benchmarking suite including real-world DAGs (e.g., TPU Tiles), the Visual Genome dataset, and classical synthetic families along with comparisons to specialized directed baselines was recognized as a meaningful contribution by *Reviewers A2gN and RPrZ*.

* **Clarity of the presentation** - *Reviewers A2gN and RPrZ* highlighted the clarity of our manuscript.  In particular, they noted the clear structure of the paper and its organization in terms of making the contributions easy to follow and the results straightforward to interpret.

## Reviewer's concerns and our responses

* **Expressiveness and efficiency** (Reviewers A2gN, RPrZ, and gox2) — We clarified that our efficiency claims are made relative to existing iterative-refinement approaches (notably discrete flow matching vs. discrete diffusion). We also emphasized that dual attention provides structural advantages beyond merely increasing the number of parameters of the original architecture, supported by our scalability and architectural ablations.

* **Positional Encodings** (Reviewers A2gN and pVm7) - We expanded discussion on both our PE choices and how they were incorporated into our model. In particular, we highlight the efficiency of the directed RRWP and further clarify the better performance of the Magnetic Laplacian compared to other Directed Laplacian choices, for which we also conducted an additional ablation.

* **Flow Matching justification** (Reviewers A2gN, pVm7 and RPrZ) - We clarified the specific role of discrete flow matching in DIRECTO, emphasizing how its decoupled training–sampling formulation enables more efficient generation through sampling-time optimization. We explain that these optimized trajectories contribute to improved performance and help account for the observed V.U.N.–Ratio tradeoffs.

* **Further ablations** (Reviewers A2gN, RPrZ, and gox2) - Following the reviewer suggestions, we added new experiments to further clarify the contributions of some model components (FiLM modulations, gated residual connections on the nodes, and the aggregation-softmax trick). These results show that each component plays an essential and complementary role in achieving high V.U.N. and strong structural fidelity.

* **Fairness comparison** (Reviewer gox2) - We added a comprehensive “fairness table” covering training time, sampling cost, and VRAM usage, for all baselines and all DIRECTO variants. This makes the relative efficiency of DIRECTO and DIRECTO-DD fully transparent.

We also revised the manuscript for improved clarity and presentation during the rebuttal. We believe that the new experiments and clarifications effectively address the reviewers’ concerns and further strengthen the quality of the work.

We thank the reviewers and the ACs for their time and consideration.

---

### Meta-Review · Area_Chair_j5TS · 2026-01-02

**Summary:**

The reviewers have raised the following major concerns:

(1) Expressiveness and scalability.

(2) Conditional generation (underexplored)

(3) Novelty

(4) Presentation and self-containedness

(5) Ablation study (insufficient)

(6) Small benchmarks

**Reviewer Concerns:**

Based on the rebuttal, I believe that most of the reviewers’ concerns have been sufficiently addressed. While I agree with reviewer pVm7 that the technical novelty of the model itself is somewhat limited, the problem being studied is both novel and important. As such, I believe the work has the potential to make a meaningful contribution to the advancement of research on graph generation.

**Reviewer Scores:**

Reviewer gox2 has indicated his/her willingness to increase the score.

---

### Decision · Program_Chairs · 2026-01-26

Accept (Poster)